

# Implementation of street trees in solar radiative exchange parameterization of TEB in SURFEX v8.0

Emilie Redon[a], Aude Lemonsu[a], Valéry Masson[a], Benjamin Morille[b], and Marjorie Musy[b]

[a]CNRM UMR 3589, Météo-France/CNRS, Toulouse, France, 42 avenue Gaspard Coriolis, 31057 Toulouse cedex 1, France
[b]Ecole nationale supérieure d'Architecture de Nantes, UMR 1563, quai François Mitterrand, 44262 Nantes cedex 2, France
*Correspondence to:* A. Lemonsu (aude.lemonsu@meteo.fr)

**Abstract.** The TEB model has been refined and improved in order to explicitly represent street trees and their impacts on radiative transfers: a new vegetated stratum on the vertical plane, which can superimpose to road and low vegetation has been added. This modification has obviously complexified the radiative calculations, but has been done with a concern to preserve a certain level of simplicity and to limit the number of new input parameters for TEB to the cover fraction of trees, the mean

height of trunks and trees, their specific Leaf Area Index and albedo. Indeed, the model is designed to be run over whole cities, for which it can simulate the local climatic variability related to urban landscape heterogeneity at the neighborhood scale. This means that computing times must be acceptable, and that input urban data must be available or quite easy to define. This simplified characterization of high vegetation necessarily induces some uncertainties on solar radiative exchanges, that were quantified through the comparison of TEB with a high spatial-resolution solar enlightenment model (SOLENE). On the

basis of an idealized geometry of urban canyon with various vegetation layouts, TEB is evaluated regarding the direct and diffuse solar radiation received by the elements that compose the canyon, as well as the total shortwave radiation absorbed (after multiple reflections) by these elements. The cases of vegetationless canyons lead to root mean square errors less than 15 W m$^{-2}$ and biases less than 5 W m$^{-2}$. For the cases with high vegetation, the statistical scores are degraded (with a trend of underestimation of solar absorption by trees) but remain acceptable. It is interesting to emphasize that the summertime TEB

simulations gave the best scores ($\left|bias\right|<$ 20 W m$^{-2}$) for all configurations and surfaces considered, which is precisely the most relevant season to assess the cooling effect of deciduous trees under temperate climate.

## 1 Introduction

For counteracting the adverse environmental effects that can result of continuous process of soil artificialisation due to urban expansion, numerous projects of local urban planning or design support and favor today the preservation and reintroduction of

vegetation in the city. From an environmental point of view, the natural soils and vegetation play a important role and bring significant benefits in different sectors (Nowak and Dwyer (2007); Mullaney et al. (2015)). They act at a microclimate level through the processes of soil water evaporation and of transpiration of plants (Qiu et al., 2013). These processes refresh the ambient air (Zhang et al., 2013) and mitigate the effect of Urban Heat Island (Feyisa et al. (2014); Önder and Akay (2014); Alavipanah et al. (2015)). The high vegetation, especially the street trees in dense urban neighbourhood, create shadow areas





that can locally improve the thermal comfort of pedestrians (Spangenberg et al. (2008); Shashua-Bar et al. (2011); Coutts et al. (2015); de Abreu-Harbich et al. (2015); Joshi and Joshi (2015)) and reduce the radiation penetration in buildings in summer (Abdel-Aziz et al. (2015); Akbari et al. (1997, 2001); Ko et al. (2015); Simpson (2002)). For some years now, green roofs are implemented more intensively in cities. The cooling effects they induce on air temperature at pedestrian level are lesser than in

case of trees or ground-based vegetation but they modify the energy budget at roof level (Taleghani et al., 2016). Nonetheless, they help to reduce the temperature fluctuations of structural roofs and to better insulate buildings (Hien et al. (2007); Koko-giannakis et al. (2011)). Many studies highlighted their efficiency to significant energy savings for heating and air conditioning (Jim, 2014). The implementation of pervious soils, whether at the ground or on buildings, also enables a more effective and sustainable management of rainwater, by the storage of water in the soil and the decrease of surface water runoff (Armson

et al. (2013); Yao et al. (2015); Zhang et al. (2015)). Besides micro climatic and hydrological impacts, the urban vegetation is identified as a biodiversity reservoir for fauna and flora in cities (Alvey, 2006). It also plays its part in architectural atmo-spheres, and more widely in the perception of environmental quality by population. The green spaces in urban environment are generally perceived very positively by inhabitants because they are places of wellness, detente, and user friendliness (Bertram and Rehdanz (2014); Bowen and Parry (2015)).

In order to investigate some of the physical processes related to the presence of vegetation in urban environment, e.g. for microclimate, hydrology or building energy issues, the modelling is definitely a necessary tool. It is also pretty relevant and powerful to assess greening strategies by quantifying the potential impacts, and it consequently enables to answer to some important expectations of public stakeholders and urban planners.

The TEB urban canopy model (Masson, 2000) is today applied for many studies of climate change impacts at city scale. Among the numerous strategies of adaptation to climate change, and attenuation of Urban Heat Island, the benefits of ur-ban environment greening has been investigated through more or less realistic scenarios. Kounkou-Arnaud et al. (2014) and de Munck (2013) have proposed and tested some strategies of ground-based vegetation implementation for the city of Paris

(France). They have evaluated the cooling potential of this vegetation through the evapotranspiration process during summer. They investigated and quantified the impacts in terms of decrease of air temperature inside the streets, as well as improvement of thermal comfort conditions outside and inside buildings. The works of de Munck (2013) have also addressed the issues of building energy consumption and of water resource at pluriannual and seasonal scales.

For such modelling exercises, the TEB model had been previously improved in order to explicitly represent urban vegeta-tion within the canyons and to parameterize at small-scale the radiative and energetic interactions between the built-up covers and the vegetation (Lemonsu et al., 2012). All types of vegetation are however managed as a ground-based layer: leaves can be in shade of buildings but do not create themselves shadow effects on roads or buildings. With the current developments we attempt to remedy this lack by modifying the solar radiation budget of tree-filled canyons. The tree layer can now surimpose

ground-based surfaces. In this way, interception, transmission or absorption of solar radiation by this additional non-opaque



surface are computed. These implementations will be evaluated by a comparison with simulations provided by a fine-scale model of enlightenment: SOLENE, developed at the CERMA laboratory, Nantes, France.

## 2 Urban high vegetation modelling

Until recently, very few urban climate models were able to take into account natural soils and vegetation. This fact constitutes
a significant limitation in modelling the radiative and energetic exchanges in urban environments, according to the results of the intercomparison exercise of urban models performed by Grimmond et al. (2011). Most of models that already included vegetation were based on the tiling approach which consists in dealing separately with impervious covers and natural covers with distinct parameterizations and without micro-scale interactions between vegetation and built-up elements. However, an important effort has been conducted these last years in order to improve the representation of physical processes related to the
presence of vegetation in urban climate models. Suburban and residential areas are characterized by an abundant vegetation (all types confounded) while they are predominant in cities. This feature of most of cities have motivated the concern of improving urban micro climate modelling in such areas. Krayenhoff et al. (2013) present an exhaustive literature review of existing models and their characteristics.

More especially, different approaches are today applied for implementing high vegetation and its implication in calculation of radiative and energetic exchanges. Among following single-layer models with integrated vegetation scheme, Lee and Park (2008) have been the first to deal with effects of trees in urban canyons in their vegetated urban canopy model (VUCM). The foliage layer is explicitly represented in the middle of the canyon above the ground. It is characterized by a cover fraction of the canyon, a thickness, and a vertical profile of Leaf Area Density (LAD). The trunks are assumed to be transparent as it is
often the case in vegetation models. Radiation budget is computed for shortwave radiation (direct and diffuse components) by accounting for shadowing effects of trees on buildings and ground-based surfaces and of buildings on trees, as well as multiple reflections, and for longwave radiation with one reflection. The hypothesis of lambertian reflectance is applied by considering each element of the canyon as a uniform surface (with uniform albedo, emissivity and temperature). For each of them, the view factors in relation to surrounding elements are computed at the center of the surface. The attenuation of radiation by
leaves is included in equations when short- or long-wave radiation crosses the foliage. It depends on transmissivity properties of foliage - inspired by works of Yamada (1982) for forest canopies - that follow an exponential law based on the LAD profile and a modulation factor depending on vegetation type. More recently, Young et al. (2015) have implemented a similar parameterization for street trees (TUrban) in the Met-Office - Reading Urban Surface Echange Scheme (MORUSES, Porson et al. (2010a, b)). The tree canopy is explicitly described as a rectangle located in the middle of the canyon and assumed to be
lower than building height. Additional view factors for partially occluded facets due to the presence of the tree are calculated using an analytical method from Hottel (1954). The model implemented by Ryu et al. (2015) considers two explicit and symmetric trees with circular crowns. The Monte Carlo algorithm developed by Wang (2014) is executed once at the beginning of the simulation and subsequently transformed into simple relations to determine view factors which trees are involved. Short-





and long- wave radiation exchanges but also sensible and latent heat exchanges and root water uptake processes are represented.

BEP-Tree is the first multi-layer model of urban energy exchange and flow at the neighborhood scale that includes trees and their both radiative (Krayenhoff, 2014) and dynamic (Krayenhoff et al., 2015) effects on buildings. View factors are also

computed using a Monte Carlo ray tracing approach. The clustered distribution of tree crowns is taken into account through a unique clumping factor for all the vegetation of the canyon. Tree foliage can be present both within and above the canyon.

For large eddy simulations (LES) in urban environment, a vegetated urban canopy model (VUC) has been integrated by Tavares et al. (2015) in the ARPS model. This ARPS-VUC version distinguishes the tree canopy from natural soils and ground-based

vegetation. The radiative calculations are however rather simplified in relation to dynamics. The net radiation flux decreases exponentially from the top of urban canopy to the ground depending on the canopy density which includes both tree vegetation and buildings.

For a microscale urban climate model such as ENVI-met (Bruse and Fleer, 1998), the fine meshing enables to resolve explicitly

each element which composes the urban environment by distinguishing buildings, impervious and natural ground-based surfaces, vegetation, and air. More especially for radiative calculations, short- and long-wave net budgets can be computed at each grid mesh, by accounting for shadow effects of buildings and absorption by vegetation. For this, a separate sky-view factor is calculated in the middle of each grid mesh, and absorption coefficients of radiation by vegetation (based on an exponential formulation) are calculated at each vertical level, according to the vertical profile of LAD.

The SOLENE-microclimat model (based on the SOLENE radiative model) is able to represent the evaporative, shading and insulating characteristics of green walls and roofs, modeled as surfaces (Malys, 2012). The trees are modelled as porous volumes in the airflow model and as semi-transparent crowns in the radiative model; the energy balance (evaporative, convective and radiative fluxes) being coupled to these models (Robitu et al., 2006). This balance leads to the assessment of leaves and air

temperatures.

## 3   General concept of TEB and urban trees representation

### 3.1   Description of urban areas in TEB

The TEB model describes the urban areas on the basis of the urban canyon concept. At the scale of a modelling grid point, the mean characteristics of the local environment are represented by a single urban canyon composed of a ground-based surface

bordered by two flat-roof buildings of same height. The urban environment is thus described in TEB based on four distinct elements that compose the urban canyons: $roof$, $wall$, and for the ground-based surfaces, a combination of impervious and natural covers referred to as $road$ and $garden$, respectively (Fig. 1).





A set of geometric parameters are defined to describe the canyon (Table 1): cover fractions of buildings and gardens (the remaining fraction is assumed to be the roadways), mean building height, and wall area density. Impervious covers are also characterized by radiative and thermal properties depending on construction materials: albedo and emissivity are prescribed for outdoor surface coating of roofs, walls, and roads; thermal conductivity and heat capacity are defined for the ensemble of

materials composing buildings and roadways. For each urban facet ($roof$, $walls$ and $road$) separately, the model computes a radiation budget and a surface energy balance. It also resolves an equation of temperature evolution with a single surface temperature associated to each facet.

For natural soils and vegetation, the radiative and energetic exchanges with atmosphere, as well as the hydrological and thermal

processes in the ground are parameterized with the Interaction Soil-Biosphere-Atmosphere model (ISBA) model (Noilhan and Planton, 1989). The vegetation stratum is described in the ISBA model as an aggregation of bare soil, low stratum (grass) and high stratum (trees) of vegetation. This vegetation stratum is characterized by composite descriptive and physiologic properties that are calculated starting from properties of these different types of natural covers. They include especially albedo and emissivity (depending of proportion of bare soil and vegetation), Leaf Area Index, stomatal resistance, and roughness length.

Note that some of those properties can evolve seasonally, but also in case of snowfall which modifies radiative properties. The vegetation stratum is connected with a soil column (Boone et al., 1999), to which are associated hydrological and thermal properties depending of soil texture and water content evolution.

This version of TEB is based on two important simplifications. First, there is no explicit spatial arrangement of the gardens

within the canyons. They are only represented as land cover fractions. In addition, the vegetation stratum, even if it can be composed of trees (through the definition of specific physiological properties), is always placed on the ground without vertical extent. This means that shadow effects on ground and buildings related to the presence of high vegetation are not taken into account, and that there is no vertical distribution of energy turbulent exchanges between vegetation and atmosphere.

### 3.2    General principles of solar radiation exchange parameterization in TEB

The present study describes the improvements of the radiation budget calculations in TEB by the implementation of explicit high vegetation. Consequently, this section is focused on the description of the radiative exchanges in the initial version of TEB. The parameterization of turbulent heat fluxes and of heat conduction processes, as well as the calculations of microclimate parameters within the canyon, are not presented here (see Masson (2000); Lemonsu et al. (2004); Hamdi and Masson (2008); Masson and Seity (2009); Lemonsu et al. (2012) for details).

The TEB urban canyons are assumed to be of infinite length so that there is no street intersection. The radiative calculation are consequently done on a two dimensional plan which crosses the canyon according to an axis perpendicular to the street direction. Two main options are available for radiative calculations: (1) a street orientation can be prescribed, so that the two walls of the canyon (referred to $wall\ A$ and $wall\ B$) are managed separately; or (2) the hypothesis of streets isotropic ori-





entation is applied, and in this case, walls are managed together (implying that they will have identical temperature evolutions).

The short- and long-wave radiation budgets are resolved in TEB for each element composing the urban canyon (roofs, walls, road, ground-based vegetation, and now tree canopy) with the aim of determining the energy absorbed by each element, that is used afterward to compute the surface energy budget.

For a given element, the shortwave radiation budget combines three contributions:

1. The direct solar radiation received before any reflection. This contribution depends on zenithal angle since the incident direct radiation is unidirectional, street orientation, and canyon aspect ratio.

2. The diffuse solar radiation received before any reflection. This contribution depends on the sky-view factor of the considered element since the diffuse radiation is assumed to be isotropic.

3. The total shortwave radiation received after multiple reflections within the canyon. After a first reflection on one of the elements of the canyon, initial contributions of direct and diffuse radiation are isotropic and are treated the same way. The part of radiation received by a given element then depends on the view factors of all the other elements and on their albedos that determine the reflected radiation part.

### 3.3 Inclusion of a high vegetation stratum for solar radiation calculation

To take into account the tree canopy in TEB, it is required to add a new vegetated stratum on the vertical plane, which can superimpose to road and low vegetation. This modification obviously complexifies the radiative calculations, but is done with a concern to preserve a certain level of simplicity and to limit the number of new input parameters for TEB. It is important to emphasize that the model is designed to be run over whole cities, for which it can simulate the local climatic variability related to urban landscape heterogeneities at the neighborhood scale. This means that computing times must be acceptable, and that input urban data must be available or quite easy to define.

The arrangement of tree canopy is here described using three parameters only (Fig. 1 and Table 1): its cover fraction ($\delta_t$), i.e. the proportion of canyon which is covered by the foliage stratum on the horizontal plane, as well as the mean height of trees ($h_t$), and the mean height of trunks ($h_{tk}$). Urban trees are assumed to be less tall than surrounding buildings and systematically confined inside the canyon so that they cannot provide shade for roofs. This hypothesis is in accordance with urban planning rules for street trees management, that impose some restrictions relative to the location and height of trees. In case of public parks and private gardens, the trees are usually planted far enough from buildings to avoid shadow on roofs even if these trees can be taller than buildings.

For now, the shape of the foliage and the vertical distribution of leaves are not refined. The crowns of trees are considered as parallelepipeds (namely computed as a rectangular 2D cross-section) with homogeneous foliage which is described by a





Leaf Area Index ($LAI_t$) and an albedo ($\alpha_t$). It is however possible to vary the LAI during the year, in order to simulate the seasonal cycle of deciduous trees. Note that trunks are not taken into account in radiative calculations. The tree vegetation stratum is considered as a semi-transparent element for shortwave radiation. A part of the incident radiation received by trees is transmitted through the foliage. The part of radiation which is not transmitted is either reflected or absorbed, depending on
albedo. These processes and the associated calculations are detailed hereafter.

## 4  Solar radiation absorption of vegetated street canyon surfaces

In this part, equations related to the implementation of a tree layer into the TEB model are presented. In order to calculate these terms in TEB, the first following sections describe the way the solar radiation reaches canyon surfaces. Then, absorption is obtained using separated resolvings of first absorption of total solar radiation and sum of absorbed shortwave radiation after
infinite reflections within the canyon.

### 4.1  Direct solar radiation received by each element

The foliage of trees plays a role of obstruction and attenuation of incident direct solar radiation ($S^{\Downarrow}$) for the other elements of the urban canyon. Consequently, to determine the direct solar radiation received by each element of the canyon, we need to solve first the equations related to high vegetation.

The direct solar radiation reaching high vegetation depends on buildings height ($h$), canyon aspect ratio ($h/w$), street orientation ($\theta_{can}$), zenithal and azimuth angles ($\lambda$, $\theta_{sun}$), as well as tree height ($h_t$):

$$S_t^{\Downarrow} = S^{\Downarrow} \, max \left[ 0 \; ; \; 1 - \frac{h}{w} \left( \frac{h - h_t}{h} \right) tan(\lambda) \, sin|\theta_{sun} - \theta_{can}| \right] \tag{1}$$

As explained previously, this radiation is partially transmitted through the foliage ($S_t^{\gg}$), whereas the remaining solar radiation
is reflected ($S_t^{\Uparrow}$) or absorbed ($S_t^{*}$):

$$S_t^{\Downarrow} = S_t^{\gg} + S_t^{\Uparrow} + S_t^{*} \tag{2}$$

The proportion of direct solar radiation transmitted through the foliage is estimated by a Beer Lambert law (Campbell and Norman, 1989) where the Leaf Area Index ($LAI_t$ expressed in $m^2$ of leaves per $m^2$ of ground) of tree canopy and an extinction coefficient ($k$) are involved. The extinction coefficient is fixed to 0.5, a default value corresponding to homogeneous repartition
of leaves in terms of density and orientation (in other words, a spherical leaf angle distribution):

$$S_t^{\gg} = S_t^{\Downarrow} \left( 1 - \exp^{(- k \, LAI_t)} \right) \tag{3}$$

The reflected radiation simply depends on the part of incident radiation untransmitted through the foliage and on the albedo of trees ($\alpha_t$):

$$S_t^{\Uparrow} = \alpha_t \, S_t^{\Downarrow} \, \exp^{(- k \, LAI_t)} \tag{4}$$





Finally, the incident direct radiation absorbed by trees is the residual term calculated from Eq. 2:

$$S_t^* = (1 - \alpha_t)\, S_t^{\Downarrow}\, \exp^{(-\,k\,LAI_t)} \tag{5}$$

The direct solar radiation received by the ground (indiscriminately road or garden fraction) is deduced by correcting the
incident solar radiation above canyon from the interception of radiation by high vegetation canopy (i.e., reflected and absorbed
radiation weighted by high vegetation cover fraction referred to as $\delta_t$), and then from the shading effects of buildings (according
to Lemonsu et al. (2012)). The same equations are obtained for road ($S_r^{\Downarrow}$) and garden ($S_g^{\Downarrow}$):

$$S_r^{\Downarrow} = \left(S^{\Downarrow} - \delta_t \left(S_t^{\Uparrow} + S_t^*\right)\right) max\left[0\,;\, 1 - \frac{h}{w}\, tan(\lambda)\, sin\left|\theta_{sun} - \theta_{can}\right|\right] \tag{6}$$

The direct solar radiation which is not received either by high vegetation, or by road or garden is assigned to the sunlighted
wall, whereas the opposite wall is in the shadow. By convention in TEB in the case of an oriented canyon, we define $wall\ A$ as
the most sunlit wall and $wall\ B$ as the shaded one.

$$S_{w_A}^{\Downarrow} = \left(S^{\Downarrow} - S_r^{\Downarrow} - \delta_t \left(S_t^{\Uparrow} + S_t^*\right)\right) \frac{w}{h} \qquad\qquad S_{w_B}^{\Downarrow} = 0 \tag{7}$$

Note that shading effects of high vegetation on roofs are not represented, since urban trees are less tall than buildings by
definition.

### 4.2 Diffuse solar radiation received by each element

The incoming diffuse solar radiation ($S^{\downarrow}$) is assumed to emit isotropically. Each urban surface of the canyon ($wall$, $road$, and
$garden$) receives a part of diffuse solar radiation according to the sky-view factor of the surface $\Psi_\star$ (see Appendix A) and the
mean radiative transmissivity between the sky and the given surface $\tau_{\star s}$ (see Appendix B). Note that the sky-view factor of
$wall$ is defined at mid-height of buildings; for ground-based surfaces $road$ and $garden$, a single sky-view factor $\Psi_{rs} = \Psi_{gs}$ is
defined at the center of the street (Masson (2000); Lee and Park (2008)). The following equations are obtained for $road$ (same
expression for $garden$) and for $wall$:

$$S_r^{\downarrow} = S^{\downarrow}\, \Psi_{rs}\, \tau_{rs} \tag{8}$$

$$S_w^{\downarrow} = S^{\downarrow}\, \Psi_{ws}\, \tau_{ws} \tag{9}$$

We admit that the residual flux of diffuse solar radiation which is not intercepted in the canyon by previous surfaces reaches
the tree canopy:

$$S_t^{\downarrow} = \frac{S^{\downarrow} - \left(\delta_r\, S_r^{\downarrow} + \delta_g\, S_g^{\downarrow} + \frac{2h}{w}\, S_w^{\downarrow}\right)}{\delta_t} \tag{10}$$





The fluxes of each surface are here expressed according to the total ground-based surface of the canyon, with $\delta_r$ and $\delta_g$ the cover fractions of road and garden in the canyon ($\delta_r + \delta_g = 1$), respectively.

### 4.3 First absorption of total solar radiation by each element

The first absorption of total (direct and diffuse) solar radiation $S^\star(0)$, before any reflections, is only function of the total solar radiation received by the considered element and of its albedo ($\alpha_\star$). The same expression is obtained for walls and ground-based surfaces ($\star$ which stands for $r$, $g$, $w_A$ or $w_B$):

$$S_\star^*(0) = (1 - \alpha_\star)\left(S_\star^{\Downarrow} + S_\star^{\downarrow}\right) \tag{11}$$

For the tree canopy, the part of absorbed direct solar radiation is corrected by the transmitted flux:

$$S_t^*(0) = (1 - \alpha_t)\left[\left(S_t^{\Downarrow} - S_t^{\gg}\right) + S_t^{\downarrow}\right] \tag{12}$$

$S_t^{\Downarrow}$ includes the transmitted flux (see Eq. 2) contrary to $S_t^{\downarrow}$ which is calculated as a residual flux (Eq. 10), not intercepted by other surfaces.

### 4.4 Sum of total solar radiation absorbed by each element

Our goal is to compute the total solar radiation absorption for each element $S^*(\infty)$ by taking into account an infinite number of reflections between all elements composing the urban canyon. At each reflection, the isotropic radiation intercepted by a given element (1) after reflections on one of the other elements (2) is conditioned by the view factor of (2) from (1) referred to as $\Psi_{12}$ (see Appendix A), the mean radiative transmissivity $\tau_{12}$ (see Appendix B) and the absorption is then determined according to reflective properties of (1). Using a single view factor in TEB radiation calculations is obviously a limitation for accurately representing the various contributions of canyon's surfaces to high vegetation. Additionally, we have a poor knowledge of a potential contribution of leaves with each others, within the tree's crown. To ensure a closing system, we define the total absorbed shortwave radiation by high vegetation as the remaining shortwave radiation, after accounting for absorption from the total incident solar radiation by all other elements of the canyon. This requires to calculate the part of shortwave radiation which goes out the canyon towards the sky. The terms $R_\infty$, $G_\infty$, $A_\infty$, $B_\infty$, and $T_\infty$ make reference to the sum of total solar radiation reflected by each surface, respectively, after an infinite number of reflections (see detailed resolution in Appendix C). Here is the expression of the total absorbed solar flux per surface:

$$S_s^*(\infty) = \Psi_{sr}\tau_{sr}(\delta_r R_\infty + \delta_g G_\infty) + \Psi_{sw}\tau_{sw}\frac{A_\infty + B_\infty}{2} + \Psi_{st}\delta_t T_\infty \tag{13}$$

$$S_r^*(\infty) = S_r^*(0) + (1 - \alpha_r)\left[\Psi_{rw}\tau_{rw}\frac{A_\infty + B_\infty}{2} + \Psi_{rt}\delta_t T_\infty\right] \tag{14}$$

$$S_g^*(\infty) = S_g^*(0) + (1 - \alpha_g)\left[\Psi_{rw}\tau_{rw}\frac{A_\infty + B_\infty}{2} + \Psi_{rt}\delta_t T_\infty\right] \tag{15}$$





$$S^*_{w_A}(\infty) = S^*_{w_A}(0) + (1-\alpha_w)\left[\Psi_{wr}\tau_{wr}\left(\delta_r R_\infty + \delta_g G_\infty\right) + \Psi_{ww}\tau_{ww}B_\infty + 0.5\Psi_{wt}\delta_t T_\infty\right] \tag{16}$$

$$S^*_{w_B}(\infty) = S^*_{w_B}(0) + (1-\alpha_w)\left[\Psi_{wr}\tau_{wr}\left(\delta_r R_\infty + \delta_g G_\infty\right) + \Psi_{ww}\tau_{ww}A_\infty + 0.5\Psi_{wt}\delta_t T_\infty\right] \tag{17}$$

$$S^*_t(\infty) = \frac{1}{\delta_t}\left[\left(\left(S^{\Downarrow}_t - S^{\gg}_t\right) + S^{\downarrow}_t\right) - \left(S^*_s(\infty) + \delta_r S^*_r(\infty) + \delta_g S^*_g(\infty) + \frac{2h}{w}\ \frac{S^*_{w_A}(\infty) + S^*_{w_B}(\infty)}{2}\right)\right] \tag{18}$$

It seems reasonable to presume that at the first reflection, the solar radiation is mainly redirected from the leaves to the sky or to the top part of walls seen by leaves. On the contrary, the multiple reflections occuring within the canyon are computed in an isotropic way. Though reflections can have different behaviors, we use a single view factor for each pair of surfaces.

Our purpose is to neglect the small amount of shortwave radiation which is supposed to be reflected by the high vegetation stratum towards the low part of the canyon during multiple reflections, but to favor a realistic representation of the first upward reflection of solar radiation by trees. In Appendix A the view factors linked to the high vegetation stratum are expressed for both options, and their impact on solar reflections calculations are fully explained in Appendix C.

## 5   Comparative exercise with the SOLENE model

An objective and exhaustive assessment of the new solar radiation calculations in TEB related to the inclusion of tree layer effects is not an easy exercise, essentially due to the lack of experimental data. Indeed, very few measurements for documenting radiative effects of trees in urban environment are available (Park et al., 2012). The objective is here to quantify the performances of TEB in simulating the different contributions of the solar radiation budget for the ensemble of urban facets and vegetation. For this purpose, we compare TEB with the SOLENE software, which is a high spatial-resolution solar and

lighting architectural model and which is used in this case as a reference. Various configurations of urban canyons with street trees (that differ in terms of vegetation density and spatial distribution) are studied so that the capacities and limits of the TEB approach can be highlighted and evaluated.

### 5.1   General presentation of the SOLENE model

The SOLENE model (Miguet and Groleau (2002); Robitu et al. (2006); Groleau and Mestayer (2013)) incorporates a radiative

transfers scheme based on the radiosity method which is applied to meshed scenes with triangular facets, particularly adapted to complex geometries (Bouyer et al. (2011); Malys et al. (2014)). This model provides a good tool to study urban radiation distinguishing solar radiations (separate direct and diffuse components, 0.3-2.5 $\mu m$) and infrared thermal radiations (2.5-18 $\mu m$).



In this research work, only solar radiations are considered. The incoming direct solar radiation is calculated by considering the sun as a point source, related to solar height (following formula in De Brichambaut (1963)) and angle of incidence of rays at the surface (Miguet, 2000). The incoming diffuse part of solar radiation is represented as a non-uniform distribution coming from a sky vault defined by an hemisphere of infinite radius, which is meshed using a geodesic triangulation. The

luminance values that are mapped on the hemisphere are derived from Perez model (Perez et al., 1993). This model, based on sky clearness and sky brightness, provides statistical distribution depending on weather type represented. Multiple reflections are computed assuming that the surfaces are lambertian and opaque for urban surfaces while the vegetation surfaces are semi-transparent according to their Leaf Area Density (Robitu, 2006). Solar simulations are performed in successive stages. The first one, based on geometric procedures, determines the visibility considering solar masks between two mesh elements or a

mesh element and a sky patch of the sky vault model (including sun). View factors (including sky view factor) are produced for each mesh. The second stage calculates the solar radiation received by each mesh element for the time-step. Then, the multiple reflections are computed by the radiosity method. This last stage provides the net solar flux received by each mesh as well as absorbed and reflected parts.

The trees have been implemented from the evolution of SOLENE into the microclimate model named SOLENE-microclimat (Robitu, 2006). In this new model, the trees are geometrically modeled by their external envelope and are considered as semi-transparent. Therefore, a percentage of the solar radiation is transmitted by the tree canopy according to a transmission coefficient and reaches other elements of the urban scene.

### 5.2   Configuration of numerical experiments

#### 5.2.1   Canyon modelling in SOLENE

The urban canyon geometry chosen for building the SOLENE's mock-ups is as simple as possible to reflect the hypotheses of TbrichambautEB: an infinite street (150 m in length in the mock-ups) bordered by two identical buildings with flat roofs. As shown in Fig. 2, it is declined in three different urban canyon forms corresponding to aspect ratios (referred to as $h/w$) of 0.5, 1 and 2. For the first two canyons, the building height is 8 m and the width of the street is 16 m for $h/w = 0.5$ and 8 m for

$h/w = 1$. For $h/w = 2$, building height is 16 m and width of the street is 8 m.

For each of these urban canyons, 13 different vegetation layouts are prescribed (Fig.3), as well as a control case without vegetation. The vegetation blocks are parallelepipeds representing the three-dimensional tree crowns, without trunk. The vegetation blocks are continuous lengthwise the canyon. Depending on the configurations, the trees can be organized in single or

double rows. According to a cut plan through the canyon, the tree crowns can fill 30, 60, or 90 % of canyon width. For the three aspect ratios, height of trees is prescribed to 5 m or 7.5 m with trunks of 2.5 or 5 m, so that the thickness of tree canopy varies between 2.5 and 5 m. An additional configuration is tested for aspect ratio $h/w = 2$, for which height of trees is prescribed to 10 m or 15 m with trunks of 5 or 10 m. For all experiments, the LAI of trees is prescribed to 1, and the albedo is prescribed to



0.25 for road, 0.30 for walls, and 0.25 for trees. All configurations are described in Fig. 1, 2, 3 and Table 2.

To treat the ensemble of configurations, 55 digital mock-ups (52 canyons with vegetation and 3 canyons without vegetation) have been built with the computer-aided-design (CAD) Salome V7_4_0 software. All mock-ups have been meshed by

the GMSH software which is a finite element mesh generator. We have applied here a non-uniform meshing in order to refine the spatial discretization of vegetation blocks, whose smallest ones for some of the vegetation layouts do not exceed 2.4 m of width and 2.5 m of height.

Each canyon is projected following the four street orientations 0°, 45°, 90°, and 135° (degrees from geographical north, in

the counter trigonometric direction). A specific location (defined by latitude and longitude) must be prescribed for astronomic calculations. The city of Nantes (France) is chosen in an arbitrarily manner (46°N, 1°E). The solar radiation exchanges are then calculated for a single daily cycle, and under sunlight conditions covering the four seasons by selecting dates close to equinoxes and solstices of year 2010, i.e. $20^{\text{th}}$ March, $21^{\text{st}}$ June, $23^{\text{rd}}$ September, and $23^{\text{rd}}$ December of 2010.

### 5.2.2 Canyon modelling in TEB

In the same way, the TEB model is run for equivalent configurations but not exactly the same, since hypotheses, approaches, and spatial resolutions are not identical in both models. For TEB simulations, the geometrical parameters describing the urban canyon form, as well as the height of trees and trunks are quite comparable to those of SOLENE simulations. But the different spatial arrangements of trees simulated by SOLENE are simply prescribed as cover fractions in TEB. As a result, some configurations (for instance, B1 and B2 or B3 and B4 in Fig.3) cannot be distinguished by the TEB approach, and are associated

to the same cover fraction of tree canopy in TEB. All geometric features of both sets of simulations with SOLENE and TEB are summarized in Table 2. In terms of process modelling, the formulation of transmissivity of radiation through the foliage according to a Beer Lambert law (Eq. 3) applied in TEB is here simplified for the evaluation stage. In order to be consistent with the SOLENE approach which applies an attenuation of 50 % when radiation crosses a cell of the external envelope which composes the foliage, the downward exponential attenuation is replaced by the expression $1 - 0.5(LAD/LAI)$ (see Appendix

B for LAD details). Note also that the TEB model is forced by the same conditions of incoming solar radiation than those calculated for SOLENE, in order to minimize gaps and differences in the two modelling approaches, that would compromise the comparison of results.

### 5.2.3 Comparison method

Finally, 880 solar radiation simulations are performed with both TEB and SOLENE. For each of them, hourly outputs are

stored. They include the direct and diffuse solar radiation received by the separated elements ($road$, $walls$, and $tree$) before reflections, as well as the total (direct and diffuse) solar radiation absorbed by the separated elements after multiple reflections. For comparing SOLENE and TEB, only the central part of the SOLENE's mock-up is used in order to avoid any boundary effect (see the scheme in Fig. 3). For each flux, the values calculated by SOLENE are summed over all grid points that compose





each element of the canyon (separately, the road, the two walls, and the trees). Finally, for both models SOLENE and TEB, the fluxes are weighted to be expressed according to the total ground-based surface of the canyon, so that they can be compared to each other and compared to incoming radiation. Note that tables with statistical scores presented hereafter have been fulfilled with the same procedure, i.e., the root mean squared error (RMSE in W m$^{-2}$), the relative error (Err in %) computed from the mean daily fluxes, and the mean bias (Bias in W m$^{-2}$), according to following equations:

$$RMSE = \sqrt{\frac{\sum_{i=1}^{n_s} \sum_{j=1}^{n_o} \sum_{k=1}^{n_h} \sum_{l=1}^{n_t} \left(F_{SOL}(i,j,k,l) - F_{TEB}(i,j,k,l)\right)^2}{n_s \cdot n_o \cdot n_h \cdot n_t}} \tag{19}$$

$$Err = 100 \cdot \frac{\sum_{i=1}^{n_s} \sum_{j=1}^{n_o} \sum_{k=1}^{n_h} \left|\frac{\overline{F_{SOL}(i,j,k)} - \overline{F_{TEB}(i,j,k)}}{\overline{F_{SOL}(i,j,k)}}\right|}{n_s \cdot n_o \cdot n_h} \tag{20}$$

$$Bias = \frac{\sum_{i=1}^{n_s} \sum_{j=1}^{n_o} \sum_{k=1}^{n_h} \sum_{l=1}^{n_t} \left(F_{SOL}(i,j,k,l) - F_{TEB}(i,j,k,l)\right)}{n_s \cdot n_o \cdot n_h \cdot n_t} \tag{21}$$

The indexes $i$, $j$, $k$, $l$ refer to as season, street orientation, aspect ratio of the street, and hour of the day, respectively, with $n_s = 4$, $n_o = 4$, $n_h = 3$, $n_t = 24$.

## 6   Results

### 6.1   General analysis and seasonal effects

The table 3 presents the statistical scores computed for the shortwave radiation absorbed by the different elements of the canyon, for both cases with and without vegetation. Note that the results are presented here by gathering the experiments performed with different street orientations and different vegetation layouts, but by distinguishing the seasons. In the light of the biases, the TEB model tends to systematically overestimate the absorption by road and walls compared to SOLENE (up to 15 and 9 W $m^{-2}$ for road and walls, respectively), and to underestimate the absorption by trees (up to 18 W $m^{-2}$ at maximum), whether the configuration or the season. Let's remind that temperate climate is characterized by four successive seasons with contrasted sunshine, air temperature and humidity conditions. Analysis of the results for each season separately indicates that relative errors are especially high for wintertime simulations for road and trees, due to a very low incoming solar radiation at that period. Nonetheless, the associated RMSE and bias are quite acceptable (<10 and < |3| W m$^{-2}$, respectively). Summer is the season which gives the best results when the canyon is tree-filled (relative errors less than 25 %) or not (relative errors around 3 %).This season is also the most relevant to be examined here because our first concern is to improve the simulation of the potential cooling effect of street trees in a urban environment, submitted to a strong UHI at this period. The evaluation is focused on effects of deciduous trees which are typical and widely present in cities under temperate climate. Such trees are leafless during winter, so that they have a negligible impact on thermal comfort and energy demand. That is why we focus on the summertime example hereafter, to assess the TEB performances for simulating the solar radiative exchanges in idealized canyons, vegetated or not.



## 6.2 Case of urban canyons without vegetation

The TEB model's radiative calculations are first evaluated for the cases without vegetation. Several comparisons with observations of radiation fluxes at neighbourhood-scale have been performed (Masson et al. (2002); Lemonsu et al. (2004, 2010); Pigeon et al. (2008)). They have shown a good capacity of the model in computing the upward short- and long-wave radiation at the top of the urban canopy, in real case configurations. But these evaluation exercises have not allowed us to analyse separately the radiative contributions of various elements that compose the urban environment. Such a model-model comparison in a controlled framework is ideal to deeply investigate and evaluate the TEB radiation parameterizations.

Both direct and diffuse solar radiation received by the road and the separate walls before any reflection, as well as the total solar radiation absorbed by these surfaces, are studied. An example of daily cycle is presented in Fig. 4 for the case $h/w = 1$ and the four street orientations, for summer season. The scatterplots integrate all the hourly fluxes simulated for the three aspects ratios, the four street orientations, and the four seasons (Fig. 5). As expected, they demonstrate a strong positive linear relationship between fluxes calculated by SOLENE and TEB models ($R^2 \geq 0.99$ except for diffuse solar radiation abosrbed by roads: $R^2 = 0.979$).

The comparison between SOLENE and TEB simulations for the direct solar radiation received by road and walls before any reflection highlights very good results. The TEB model is able to reproduce the geometrical effects of the canyon on radiation penetration according to the time of the day, as well as the street orientation. For NE-SW and NW-SE oriented streets, TEB simulates correctly the dissymmetry of fluxes between the two walls, as well as the temporal shift in peak of radiation received by the road in comparison with the N-S oriented street. For the E-W street case, the direct radiation received by the road is marked by a plateau effect between 8 am and 7 pm. The two walls have different behaviours: the wall the most exposed to sun receives the maximum direct radiation at solar noon, whereas the most shaded wall receives direct radiation only early in the morning and late in the afternoon. The scores confirm the good performances of TEB: RMSE is 8.91 and 4.54 W m$^{-2}$, and bias is -0.28 and +0.40 W m$^{-2}$ for road and walls, respectively (Table 4). They are associated with low relative errors of only 1 % for both road and walls.

For the diffuse solar radiation calculations, let's remind that TEB does not separate the two walls. Consequently, here are compared in Fig. 4 the diffuse solar radiation received by the composite wall of TEB, and the average of the diffuse solar radiation received by the separate walls of the SOLENE simulation. By considering an average surface, TEB underestimates the diffuse solar radiation received by walls in the morning and the afternoon, and it overestimates it at solar noon. Inversely, it overestimates the diffuse solar radiation received by the road in the morning and the afternoon, and it underestimates it at solar noon. In this case, RMSE and biases remain weak (<6 and < |1| W m$^{-2}$, respectively) because the involved fluxes are not very high (depending on the season, the diffuse component is only 15-25 % of the total incident solar radiation), but relative errors are slightly higher than for direct solar radiation, reaching 7 and 5 % for road and walls, respectively (Table 5). However, these



discrepancies have no impact on performances of TEB neither at daily scale on fluxes of walls nor on instantaneous sum of canyon fluxes. Moreover, the dissymetry of received solar radiation fluxes is no longer observed in E/W oriented cases, where models are fitting very well for roads with relative errors less than 1 %.

Finally, the total solar radiation absorbed by road and walls is well estimated by TEB despite the simplified hypotheses of the model and the use of a unique sky view factor by surface: RMSE and biases are 9.88 and +3.50 W m$^{-2}$ for road, respectively, and 5.72 and +2.80 W m$^{-2}$ for walls (Table 6). In view of the important incident radiation flux, exceeding 1 000 W m$^{-2}$ at solar noon, the relative error of 3 % for both surfaces remains moderate.

### 6.3 Case of urban canyons with vegetation

The same evaluations are conducted for vegetated canyons. The statistical scores are computed as previously (see Eq. 19, 20, 21) but by accounting for the 13 vegetation layouts. As an example, a comparison of hourly fluxes is presented in Fig. 6, for the layout A and an aspect ratio $h/w = 1$, and for the summer daily cycle. This configuration is one of the most simple and comparable layouts between the two models: trees are 7.5 m high (i.e. almost the same height as buildings), tree crowns have a 5 m thickness, they are centered in the middle of the canyon and cover 90 % of the canyon width on the horizontal plane.

The daily evolution of direct solar radiation received by the different elements of the canyon can be compared to the case without vegetation (Fig. 4). The same patterns are obtained, whether for walls or road and the four street orientations, with an attenuation due to the presence of trees. Here, a significant part of direct incoming solar radiation is intercepted by the foliage. Due to its semi-transparency properties (the transmissivity is here prescribed to 0.5), the foliage allows half of radiation to pass

through. As a result, the urban surfaces (walls and road) receive less incoming direct radiation but are never totally obstructed by trees. These processes are correctly simulated by TEB and the scores (based on the simulations at summertime) for the configuration A are quite acceptable: RMSE are 18.32, 6.86 and 10.04 W m$^{-2}$ for road, walls, and trees, respectively, with mean errors of 14, 5, and 4 % only (Table 4). Due to the expression of direct solar radiation intercepted by high vegetation at the top of the crown which is treated as an horizontal surface in TEB model (Eq. 1), the fluxes reaching the trees in TEB are

globally underestimated compared to the SOLENE fluxes that include contributions on the vertical faces of the crown envelope (Fig. 7 a., all seasons and configurations). Consequently, the solar radiation which is not intercepted by the tree layer in TEB simulations is assigned to the road. The thicker the crown is, the greater is the error (B1 vs D1 or B2 vs D2), particularly when the tree rows are away from the walls (B1 vs B2, D1 vs D2 or D3 vs D4) and separated (B4 vs B3) contrary to a continuous tree layer occupying almost the entire width of the canyon (A, C1, C2).

As expressed in Eq. 8 and 9, the diffuse solar radiation received by road and walls (walls are managed together as a mean surface), depends on the sky-view factor of the given surface and on an attenuation coefficient of the incoming radiation through the foliage. Regarding all seasons, the received solar radiation by ground-based surfaces is also overestimated while the diffuse solar flux reaching tree crowns is underestimated in the TEB simulations (Fig. 7 b.). Contrasted results are obtained





for the vertical surfaces whether the vegetation layouts, in particular between double-row or centered trees (B3 vs B4). As previously discussed in section 5.4, these defects are related to the use of a single sky-view factor for each surface, which is computed at mid-height of buildings for walls and in the middle of the street for road. Since the diffuse solar radiation received by the trees is calculated by TEB as the residual part of the incoming diffuse solar radiation which was not received by walls

and road (see Eq. 10), it is inversely slightly underestimated. Considering the extremely vegetated canyon A, relative errors are 15, 9 and 20 % for road, walls, and tree, respectively, corresponding to RMSE of 2.92, 2.72 and 5.42 W m$^{-2}$ (Table 5).

For this configuration, the total solar radiation absorbed by the different elements of the canyon is simulated with a correct daily dynamics and good magnitudes, so that statistical scores are good: 27, 19 and 5 % of relative error for road, walls and tree,

respectively, which mean RMSE of only 19.67, 20.15 and 14.53 W m$^{-2}$ (Table 6). The global scatterplot (Fig. 7 c., all seasons and configurations) confirms a good agreement between TEB and SOLENE simulations ($R^2 \geq 0.90$) but an underestimation trend for the fluxes received and absorbed by trees is highlighted in some configurations.

### 6.4  Sensitivity to vegetation layout characteristics

The simple representation of tree canopy inside the canyon in the TEB model obviously present some limitations and can

lead to more or less important biases in the radiative calculations. On the basis of the diverse vegetation layouts that have been tested, three issues are addressed here: (1) the effect of the tree horizontal coverage; (2) the effect of tree canopy height compared to building height; and (3) the effect of tree location - centered or on side - in the canyon.

The comparison of statistical scores for the different vegetation layouts shows that the foliage horizontal coverage influences

the performances of the TEB model (Fig. 8). In its parameteriztion of radiative echanges, TEB represents the tree canopy as a layer of leaves which covers the total width of the canyon but which is more or less diluted depending on the prescribed cover fraction ($\delta_t$). This approach leads to better results when this fraction is high (here for $\delta_t = 90\%$).

The vertical location of tree canopy in relation to building height has also a significant impact on TEB performances (Fig.

9). Both for direct and diffuse solar radiation received by road, walls, and trees, the model gives better results when the foliage layer covers the high half of the canyon. The results are less good when the foliage thickness is smaller, especially when the foliage layer is located in the middle part of the canyon.

It is more difficult to conclude on the influence of horizontal location. The differences seem less significant when compar-

ing the configurations for which the vegetation is centered along the canyon axis or located on the edge of the canyon (Fig. 10). By contrast, in case of double lines of trees, it is noted that TEB simulates less well the direct and diffuse solar radiation received by road.





## 6.5 Analysis of an integrated mean daily canyon albedo

The mean daily albedo of the canyon ($\alpha_{can}$) is calculated as the ratio between the outgoing shortwave radiation (which is deduced from the difference between the incoming radiation $S_{can}^{\Downarrow} + S_{can}^{\downarrow}$ and the absorbed radiation $S_{can}^{*}$) and the incoming shortwave radiation:

$$\alpha_{can} = \frac{\left(S_{can}^{\Downarrow} + S_{can}^{\downarrow}\right) - S_{can}^{*}}{\left(S_{can}^{\Downarrow} + S_{can}^{\downarrow}\right)} \tag{22}$$

For each simulation, these fluxes are integrated over the daily cycle in order to compute a mean canyon albedo for each case. The results are presented as boxplots (Fig. 11) that gather seasons, street orientations, vegetation layouts, but that distinguish the canyon aspect ratios. It is a synthetic indicator of TEB performance to estimate the shortwave radiative budget of the entire canyon, which is crucial when SURFEX is run coupled with an atmospheric model such as the research model MESO-NH (**?**) or the numerical prediction model AROME (Seity et al., 2011).

The previous comparisons (section 6.2) have shown that TEB tends to slightly overestimate the solar absorption by road and walls in case of non-vegetated canyons, so that the mean canyon albedo is slightly weaker than in SOLENE: -1.5 % for the $h/w = 0.5$, -2.5 % for $h/w = 1$, and -3.5 % for $h/w = 2$. This underestimation is less marked for wide canyons because sunlight can totally penetrate and enlighten walls or ground-based surfaces; in this configuration, the approximation related to the use of unique sky-view-factor per surface is less disadvantageous than in case of narrow canyons.

For vegetated canyons, the tree canopy absorbs less solar radiation in TEB than in SOLENE whereas the absorption by road and walls remains underestimated. Despite the combination of these compensatory effects, the mean canyon albedo is systematically underestimated by TEB, which translates a global overestimation of solar radiation absorption by the canyon. This defect is accentuated for narrow canyons. In case of $h/w = 0.5$ or 1 the mean error for ($\alpha_{can}$) is less than 10 % and 22 %, respectively, but exceeds 35 % for greater aspect ratios. In case of vegetated canyons, two simplifications may explain these biases: the use of a unique sky-view-factor for each surface, and the calculation of the sky-view-factor for high vegetation which is done at the mid-height of the crown.

Finally, one can note the variability of ($\alpha_{can}$) obtained with TEB is less than with SOLENE (Fig. 11) since TEB cannot manage differently some horizontal arrangements of the vegetation layouts (e.g. B1 and B2 configurations, see Fig. 3).

## 6.6 Benefit of TEB developments

The shortwave radiation received and absorbed by the walls and the road can be strongly affected by the presence of tree vegetation. The comparison between the initial version of TEB which deals with vegetation at ground level, and the new version which explicitly includes an additional tree stratum, shows differences (Fig. 12). The solar radiation absorbed by the urban facets (road and vertical surfaces) is largely overestimated in the TEB initial version, since the shadowing effects by tree veg-



etation are not taken into account. Inversely, the vegetation is much more subject to shade of buildings since it is assumed to be on the ground for radiative calculations. With the configuration A as an example (i.e., with a tree coverage of 90 % inside the canyon, see Fig. 3), the reference version of the TEB model is characterized by an overestimation in solar radiation absorption of +80 % for the road and +64 % for the walls, while an underestimation of -6 % is noted for the trees, compared to
the implemented TEB version. This can have a significant impact on thermal comfort conditions for pedestrians as well as on energy consumption of buildings for air-conditioning usage.

However, the reference case presents a strong dispersion in the simulation of solar fluxes absorbed by road, depending on the configurations (aspect ratio of canyon, street orientation, and season). In some cases especially, the solar flux absorbed by
vegetation is greater than in the new version of TEB. This effect is related to the fact that the reference version does not treat the vegetation as a semi-transparent object and does not take into account its transmissivity properties. As a result, the vegetation absorbs all the solar radiation which is received and not reflected. In the present case, this part is 75 % of the incident radiation since $\alpha_t = 0.25$. In the new version, 50 % of the received solar radiation is transmitted through the foliage and 75 % of the remaining flux is absorbed by trees, i.e. 37.5 % of the total solar radiation received by trees.

**7   Conclusions**

The TEB model has been refined and improved in order to explicitly represent street trees and their impacts on radiative transfers. The new parameterization is based on the quite simple hypotheses of TEB: (1) a little detailed geometry without specific spatial arrangement of ground-based surfaces and (2) a single view factor to each emitting and receiving surface applied for radiative calculations.

To take into account the tree canopy in TEB, it was however required to add a new vegetated stratum on the vertical plane, which can superimpose to road and low vegetation. This modification has obviously complexified the radiative calculations, but has been done with a concern to preserve a certain level of simplicity and to limit the number of new input parameters for TEB. It is important to emphasize that the model is designed to be run over whole cities, for which it can simulate the local
climatic variability related to urban landscape heterogeneities at the neighborhood scale. This means that computing times must be acceptable, and that input urban data must be available or quite easy to define. Consequently, the high vegetation is here described using only five input parameters: cover fraction of trees, height of trees, height of trunks, LAI, and albedo. This simplified characterization of high vegetation necessarily induces some uncertainties on solar radiative exchanges. This has been highlighted here through the comparison of TEB with a high spatial-resolution solar and lighting model (SOLENE). The
TEB results are nonetheless quite acceptable for the majority of studied configurations in terms of canyon geometry, street orientation, vegetation layout, or sunshine conditions. Indeed, TEB simulations of summer gathered best scores ($|bias| < 20$ W m$^{-2}$) for all configurations and surfaces considered, which is precisely the most relevant season to assess the cooling effect of deciduous trees under temperate climate. Additionally, it is necessary to put in perspective obtained scores with the fact



that they include the error generated by the treatment of walls as a mean wall in TEB in comparison with separate walls of SOLENE. This identified error is no longer existing at the canyon or daily scales. The study of mean daily canyon albedos reveals mean errors less than 9 % for $h/w = 0.5$ and 22 % for $h/w = 1$, respectively. Despite of worse results for higher aspect ratios, the vegetalization of very large avenues with medium or high buildings is more likely than within narrower and deeper

5   streets.

The parameterization of shortwave radiation exchanges within the canyon is now more realistic: shading effects of trees on vertical and ground-based surfaces but also shading effects of buildings on trees are computed. Infinite reflections within the canyon are also conditioned to the transmissivity term calculated per pairs of exchanging surfaces. The future developments

10   will focus on the separate calculation of turbulent energy fluxes for ground-based and high vegetation. The aerodynamic effect of trees on air flow within the canyon should also be parameterized. Based on this more sophisticated version of the TEB model, new impact studies could be conducted and greening adaptation strategies could be evaluated more precisely.

## 8   Code availability

The TEB code is available in open source via the surface modelling platform called SURFEX, downloadable at http://www.cnrm-

15   game-meteo.fr/surfex/. This Open-SURFEX will be updated at relatively low frequency (each 3 to 6 months) and developments presented here are not included in the last version yet. If you need more frequent updates, or if you need what is not in Open-SURFEX (DrHOOK, FA/LFI formats, GAUSSIAN grid), we invite you to follow the procedure to get a SVN account and to access real-time modifications of the code (see instructions at the previous link).




## Appendix A: Sky-view and view factors

Sky-view-factors for road, garden and wall (Eq. **??**) as well as the view factors between elements remain unchanged in relation to initial version of the radiative calculations in TEB (described in Masson (2000) and Lemonsu et al. (2012)):

$$\Psi_{rs} \;=\; \Psi_{sr} = \sqrt{\left(\frac{h}{w}\right)^2 + 1} - \frac{h}{w} \tag{A1}$$

$$\Psi_{wr} \;=\; \Psi_{ws} = \frac{\frac{1}{2}\left(\frac{h}{w} + 1 - \sqrt{\left(\frac{h}{w}\right)^2 + 1}\right)}{\frac{h}{w}} \tag{A2}$$

$$\Psi_{sw} \;=\; 1 - \Psi_{sr} \tag{A3}$$

$$\Psi_{ww} \;=\; 1 - 2\Psi_{ws} = \frac{\sqrt{\left(\frac{h}{w}\right)^2 + 1} - 1}{\frac{h}{w}} \tag{A4}$$

$$\Psi_{rw} \;=\; 1 - \Psi_{rs} = 1 - \left(\sqrt{\left(\frac{h}{w}\right)^2 + 1} - \frac{h}{w}\right) \tag{A5}$$

For the tree canopy, the sky-view-factor and view factors from road and walls are computed in the middle of the canyon and at mid-height of crown:

$$\Psi_{st} \;=\; \sqrt{\left(\frac{h}{w} \cdot \frac{h - h_{cw}}{h}\right)^2 + 1} - \left(\frac{h}{w} \cdot \frac{h - h_{cw}}{h}\right) \tag{A6}$$

$$\Psi_{rt} \;=\; 0 \tag{A7}$$

$$\Psi_{wt} \;=\; 1 - \Psi_{st} \tag{A8}$$

## Appendix B: Mean radiative transmissivity of canyon tree canopy

The multiple reflections of solar radiation inside the canyon (as detailed in Masson (2000) and Lemonsu et al. (2012)) are now affected by the presence of trees whose foliage intercepts, reflects and absorbs a part of the energy. The transmissivity of radiation through the foliage of tree canopy is variable according to the way the rays cross the foliage and the distance they travel. According to the location of tree crowns inside the canyon and the dominant orientation of radiation (e.g., emission from sky to ground or from wall to ground do not reach the foliage the same way), these rays can cross all the foliage thickness





or only a small portion. The vertical distribution of leaves in the tree crowns has also an impact on transmissivity.

Different transmissivity functions (referred to as $\tau_{12}$ for exchanges between element 1 and element 2) are calculated depending of the surfaces involved in radiation exchanges. One distinguishes four cases of radiation exchanges, by hypothesizing that

transmissivity functions are symmetric, i.e., exchanges from element 1 to element 2 is equivalent to the reverse way: between ground-based surfaces and sky ($\tau_{rs} = \tau_{sg}$) or wall ($\tau_{rw} = \tau_{gw}$), between wall and sky ($\tau_{ws}$), between wall and wall ($\tau_{ww}$). For each case we admit, according to Lee and Park (2008), that majority of radiation exchanges occurs in a specific zone of the canyon, for which the Leaf Area Density ($LAD_t$ expressed in $m^3$ of leaves per $m^2$ of ground) is calculated:

$$\tau_{rs} = 1 - \delta_t \left[ 1 - \exp^{\left( -k \int_0^h LAD_t \, dz \right)} \right] \tag{B1}$$

$$\tau_{rw} = 1 - \delta_t \left[ 1 - \exp^{\left( -k \int_0^{\frac{h}{2}} LAD_t \, dz \right)} \right] \tag{B2}$$

$$\tau_{ws} = 1 - \delta_t \left[ 1 - \exp^{\left( -k \int_{\frac{h}{2}}^h LAD_t \, dz \right)} \right] \tag{B3}$$

$$\tau_{ww} = 1 - \delta_t \left[ 1 - \exp^{\left( -k \int_{\frac{h}{4}}^{\frac{3h}{4}} LAD_t \, dz \right)} \right] \tag{B4}$$

Note that this expression is in accordance with the one applied in Eq. **??** in which $LAD_t$ is integrated on the complete thickness of foliage so that it is equivalent to $LAI_t$. As mentioned above, since the tree crown is described as a parallelepiped with a regular distribution of leaves, a uniform vertical profile of LAD is here applied.

## Appendix C: Total solar radiation absorption by solving infinite reflections

For solar radiation calculations, the TEB model takes into account an infinite number of reflections between all elements composing the urban canyon. At each reflection, the isotropic radiation intercepted by a given element (1) after reflections on one of the other elements (2) is conditioned by the view factor of (2) from (1) referred to as $\Psi_{12}$ (see Appendix A), the mean radiative transmissivity $\tau_{12}$ (see Appendix B) and the reflection is then determined according to reflective properties of (1). As seen in the section 4.4, the total solar radiation absorbed by each elements of the canyon or redirected towards the sky

is function of infinite reflections $R_\infty$, $G_\infty$, $A_\infty$, $B_\infty$, and $T_\infty$ that are still unknowns at this stage. These terms involve the first reflection on each element: $R_0$, $G_0$, $A_0$, $B_0$, $T_0$. For road, garden, and walls, $R_0$, $G_0$, $A_0$ and $B_0$ simply depend on the incident solar radiation on the surface and the albedo. From the equations 14, 15, 16, 17 and 18, we can deduce the reflected part occuring at the $n+1^{th}$ absorption as the complementary term of the $n+1^{th}$ solar radiation reception for only opaque





elements of the canyon, $road$, $garden$, $wall\ A$, and $wall\ B$, respectively. As an exemple, we obtain for road:

$$R_0 \;=\; \alpha_r \left( S_r^{\Downarrow} + S_r^{\downarrow} \right) \tag{C1}$$

$$R_{n+1} \;=\; R_0 + \alpha_r \left[ \Psi_{rw}\tau_{rw}W_n + \Psi_{rt}\delta_t T_n \right] \tag{C2}$$

For the first tree canopy reflection, the part of received direct solar radiation is corrected by the transmitted flux (see Eq. 2):

$$T_0 \;=\; \alpha_t \left[ \left( S_t^{\Downarrow} - S_t^{\ggg} \right) + S_t^{\downarrow} \right] \tag{C3}$$

Some uncertainties remain about relevance of sky-view or view factors which ones could formulate to represent reflective contributions from other surfaces at $n^{th}$ to the absorption or reflection by the tree layer at $n+1^{th}$, as well as potential absorption of energy within the tree's crown. Consequently, the solar flux reflected by $tree$ at $n+1^{th}$ has been determined as the residual term by assuming that the $n^{th}$ solar reflection coming from each element ($road$, $garden$, $wall\ A$, and $wall\ B$) which is not received by $road$, $garden$, $wall\ A$, $wall\ B$ or which is not returned to $sky$ at $n+1$ is received by $tree$.

During each interreflection, a part of $n^{th}$ reflected flux which is potentially available for the $(n+1)^{th}$ reflection or absorption is intercepted by high vegetation. This intercepted part related to the presence of foliage on the way of scattered rays towards each receiving element or sky are formulated as following:

$$I_s\left(n+1\right) \;=\; \Psi_{sr}(1-\tau_{sr})(\delta_r R_n + \delta_g G_n) + \Psi_{sw}\left(1-\tau_{sw}\right)\frac{A_n+B_n}{2} \tag{C4}$$

$$I_r\left(n+1\right) \;=\; \Psi_{rw}\left(1-\tau_{rw}\right)\frac{A_n+B_n}{2} \tag{C5}$$

$$I_g\left(n+1\right) \;=\; \Psi_{rw}\left(1-\tau_{rw}\right)\frac{A_n+B_n}{2} \tag{C6}$$

$$I_{w_A}\left(n+1\right) \;=\; \Psi_{wr}(1-\tau_{wr})(\delta_r R_n + \delta_g G_n) + \Psi_{ww}\left(1-\tau_{ww}\right)B_n \tag{C7}$$

$$I_{w_B}\left(n+1\right) \;=\; \Psi_{wr}(1-\tau_{wr})(\delta_r R_n + \delta_g G_n) + \Psi_{ww}\left(1-\tau_{ww}\right)A_n \tag{C8}$$

Finally, the solar flux which is intercepted by $tree$ at $(n+1)$ is expressed as the sum of interceptions on the way of receiving elements of the canyon or sky:

$$I_t\left(n+1\right) = \frac{1}{\delta_t}\left(I_s\left(n+1\right) + \delta_r I_r\left(n+1\right) + \delta_g I_g\left(n+1\right) + \frac{2h}{w}\frac{I_{w_A}\left(n+1\right) + I_{w_B}\left(n+1\right)}{2}\right) \tag{C9}$$

The solar energy which is reflected by tree at (n+1) is consequentely:

$$T_{n+1} \;=\; \alpha_t\, I_t\left(n+1\right) \tag{C10}$$





As a result, after an infinite number of reflections, the equation system can be written:

$$R_\infty = R_0 + \alpha_r \left[\Psi_{rw}\tau_{rw}W_\infty + \Psi_{rt}\delta_t T_\infty\right] \tag{C11}$$

$$G_\infty = G_0 + \alpha_g \left[\Psi_{rw}\tau_{rw}W_\infty + \Psi_{rt}\delta_t T_\infty\right] \tag{C12}$$

$$W_\infty = W_0 + \alpha_w \left[\Psi_{ww}\tau_{ww}W_\infty + \Psi_{wr}\tau_{rw}\left(\delta_r R_\infty + \delta_g G_\infty\right) + \Psi_{wt}\delta_t T_\infty\right] \tag{C13}$$

$$T_\infty = T_0 + \frac{\alpha_t}{\delta_t}\left[\left(\Psi_{sw}(1-\tau_{sw}) + \Psi_{rw}(1-\tau_{rw}) + \Psi_{ww}(1-\tau_{ww})\right)W_\infty \right.$$
$$\left. + \left(\Psi_{sr}(1-\tau_{sr}) + \Psi_{wr}(1-\tau_{wr})\right)\left(\delta_r R_\infty + \delta_g G_\infty\right)\right] \tag{C14}$$

The formulations can be simplified by gathering the equations for walls in a single expression for a mean wall according to:

$$W\left(n+1\right) = \frac{A\left(n+1\right) + B\left(n+1\right)}{2} \tag{C15}$$

As a result, we resolve the linear system of four equations with four unknowns:

$$R_\infty = R_0 + \mathcal{F}_{rw}W_\infty + \mathcal{F}_{rt}T_\infty \tag{C16}$$

$$G_\infty = G_0 + \mathcal{F}_{gw}W_\infty + \mathcal{F}_{gt}T_\infty \tag{C17}$$

$$W_\infty = W_0 + \mathcal{F}_{ww}W_\infty + \mathcal{F}_{wr}R_\infty + \mathcal{F}_{wg}G_\infty + \mathcal{F}_{wt}T_\infty \tag{C18}$$

$$T_\infty = T_0 + \mathcal{F}_{tw}W_\infty + \mathcal{F}_{tr}R_\infty + \mathcal{F}_{tg}G_\infty \tag{C19}$$

The geometric and reflective factors are computed as following:

$$\mathcal{F}_{rw} = \Psi_{rw}\,\tau_{rw}\,\alpha_r \tag{C20}$$

$$\mathcal{F}_{rt} = \Psi_{rt}\,\delta_t\,\alpha_r \tag{C21}$$

$$\mathcal{F}_{gw} = \Psi_{rw}\,\tau_{rw}\,\alpha_g \tag{C22}$$

$$\mathcal{F}_{gt} = \Psi_{rt}\,\delta_t\,\alpha_g \tag{C23}$$

$$\mathcal{F}_{wr} = \Psi_{wr}\,\tau_{wr}\,\delta_r\,\alpha_w \tag{C24}$$

$$\mathcal{F}_{wg} = \Psi_{wr}\,\tau_{wr}\,\delta_g\,\alpha_w \tag{C25}$$

$$\mathcal{F}_{ww} = \Psi_{ww}\,\tau_{ww}\,\alpha_w \tag{C26}$$

$$\mathcal{F}_{wt} = 0.5\Psi_{wt}\,\delta_t\,\alpha_w \tag{C27}$$

$$\mathcal{F}_{tw} = \left[\Psi_{sw}(1-\tau_{sw}) + \Psi_{rw}(1-\tau_{rw}) + \Psi_{ww}(1-\tau_{ww})\right]\frac{1}{\delta_t}\,\alpha_t \tag{C28}$$

$$\mathcal{F}_{tr} = \left[\Psi_{sr}(1-\tau_{sr}) + \Psi_{wr}(1-\tau_{wr})\right]\frac{\delta_r}{\delta_t}\,\alpha_t \tag{C29}$$

$$\mathcal{F}_{tg} = \left[\Psi_{sr}(1-\tau_{sr}) + \Psi_{wr}(1-\tau_{wr})\right]\frac{\delta_g}{\delta_t}\,\alpha_t \tag{C30}$$



The resolution of the equation system gives the following expressions for multiple reflections on tree, walls, road, and garden:

$$
\begin{aligned}
T_\infty = \ & \left( \frac{1 - \mathcal{F}_{wr}\mathcal{F}_{rw} - \mathcal{F}_{wg}\mathcal{F}_{gw} - \mathcal{F}_{ww}}{\mathcal{D}} \right) T_0 \\
& + \left( \frac{\mathcal{F}_{tr}\left(1 - \mathcal{F}_{wg}\mathcal{F}_{gw} - \mathcal{F}_{ww}\right) + \mathcal{F}_{wr}\left(\mathcal{F}_{tg}\mathcal{F}_{gw} + \mathcal{F}_{tw}\right)}{\mathcal{D}} \right) R_0 \\
& + \left( \frac{\mathcal{F}_{tg}\left(1 - \mathcal{F}_{wr}\mathcal{F}_{rw} - \mathcal{F}_{ww}\right) + \mathcal{F}_{wg}\left(\mathcal{F}_{tr}\mathcal{F}_{rw} + \mathcal{F}_{tw}\right)}{\mathcal{D}} \right) G_0 \\
& + \left( \frac{\mathcal{F}_{tr}\mathcal{F}_{rw} + \mathcal{F}_{tg}\mathcal{F}_{gw} + \mathcal{F}_{tw}}{\mathcal{D}} \right) W_0
\end{aligned} \tag{C31}
$$

$$
\begin{aligned}
W_\infty = \ & \left( \frac{\mathcal{F}_{wr}\mathcal{F}_{rt} + \mathcal{F}_{wg}\mathcal{F}_{gt} + \mathcal{F}_{wt}}{\mathcal{D}} \right) T_0 \\
& + \left( \frac{\mathcal{F}_{wr}\left(1 - \mathcal{F}_{tg}\mathcal{F}_{gt}\right) + \mathcal{F}_{tr}\left(\mathcal{F}_{wg}\mathcal{F}_{gt} + \mathcal{F}_{wt}\right)}{\mathcal{D}} \right) R_0 \\
& + \left( \frac{\mathcal{F}_{wg}\left(1 - \mathcal{F}_{tr}\mathcal{F}_{rt}\right) + \mathcal{F}_{tg}\left(\mathcal{F}_{wr}\mathcal{F}_{rt} + \mathcal{F}_{wt}\right)}{\mathcal{D}} \right) G_0 \\
& + \left( \frac{1 - \mathcal{F}_{rt}\mathcal{F}_{tr} - \mathcal{F}_{tg}\mathcal{F}_{gt}}{\mathcal{D}} \right) W_0
\end{aligned} \tag{C32}
$$

$$
\begin{aligned}
R_\infty = \ & \left( \frac{\mathcal{F}_{rw}\left(\mathcal{F}_{rt}\mathcal{F}_{wr} + \mathcal{F}_{gt}\mathcal{F}_{wg} + \mathcal{F}_{wt}\right) + \mathcal{F}_{rt}\left(1 - \mathcal{F}_{wr}\mathcal{F}_{rw} - \mathcal{F}_{wg}\mathcal{F}_{gw} - \mathcal{F}_{ww}\right)}{\mathcal{D}} \right) T_0 \\
& + \left( 1 + \frac{\mathcal{F}_{rw}\left(\mathcal{F}_{wr}\left(1 - \mathcal{F}_{tg}\mathcal{F}_{gt}\right) + \mathcal{F}_{tr}\left(\mathcal{F}_{wg}\mathcal{F}_{gt} + \mathcal{F}_{wt}\right)\right) + \mathcal{F}_{rt}\left(\mathcal{F}_{tr}\left(1 - \mathcal{F}_{wg}\mathcal{F}_{gw}\right) + \mathcal{F}_{wr}\left(\mathcal{F}_{tg}\mathcal{F}_{gw} + \mathcal{F}_{tw}\right)\right)}{\mathcal{D}} \right) R_0 \\
& + \left( \frac{\mathcal{F}_{rw}\left(\mathcal{F}_{wg}\left(1 - \mathcal{F}_{tr}\mathcal{F}_{rt}\right) + \mathcal{F}_{tg}\left(\mathcal{F}_{wr}\mathcal{F}_{rt} + \mathcal{F}_{wt}\right)\right) + \mathcal{F}_{rt}\left(\mathcal{F}_{tg}\left(1 - \mathcal{F}_{wr}\mathcal{F}_{rw} - \mathcal{F}_{ww}\right) + \mathcal{F}_{wg}\left(\mathcal{F}_{tr}\mathcal{F}_{rw} + \mathcal{F}_{tw}\right)\right)}{\mathcal{D}} \right) G_0 \\
& + \left( \frac{\mathcal{F}_{rt}\left(\mathcal{F}_{tr}\mathcal{F}_{rw} + \mathcal{F}_{tg}\mathcal{F}_{gw} + \mathcal{F}_{tw}\right) + \mathcal{F}_{rw}\left(1 - \mathcal{F}_{tr}\mathcal{F}_{rt} - \mathcal{F}_{tg}\mathcal{F}_{gt}\right)}{\mathcal{D}} \right) W_0
\end{aligned} \tag{C33}
$$

$$
\begin{aligned}
G_\infty = \ & \left( \frac{\mathcal{F}_{gw}\left(\mathcal{F}_{rt}\mathcal{F}_{wr} + \mathcal{F}_{gt}\mathcal{F}_{wg} + \mathcal{F}_{wt}\right) + \mathcal{F}_{gt}\left(1 - \mathcal{F}_{wr}\mathcal{F}_{rw} - \mathcal{F}_{wg}\mathcal{F}_{gw} - \mathcal{F}_{ww}\right)}{\mathcal{D}} \right) T_0 \\
& + \left( \frac{\mathcal{F}_{gw}\left(\mathcal{F}_{wr}\left(1 - \mathcal{F}_{tg}\mathcal{F}_{gt}\right) + \mathcal{F}_{tr}\left(\mathcal{F}_{wg}\mathcal{F}_{gt} + \mathcal{F}_{wt}\right)\right) + \mathcal{F}_{gt}\left(\mathcal{F}_{tr}\left(1 - \mathcal{F}_{wg}\mathcal{F}_{gw}\right) + \mathcal{F}_{wr}\left(\mathcal{F}_{tg}\mathcal{F}_{gw} + \mathcal{F}_{tw}\right)\right)}{\mathcal{D}} \right) R_0 \\
& + \left( 1 + \frac{\mathcal{F}_{gw}\left(\mathcal{F}_{wg}\left(1 - \mathcal{F}_{tr}\mathcal{F}_{rt}\right) + \mathcal{F}_{tg}\left(\mathcal{F}_{wr}\mathcal{F}_{rt} + \mathcal{F}_{wt}\right)\right) + \mathcal{F}_{gt}\left(\mathcal{F}_{tg}\left(1 - \mathcal{F}_{wr}\mathcal{F}_{rw} - \mathcal{F}_{ww}\right) + \mathcal{F}_{wg}\left(\mathcal{F}_{tr}\mathcal{F}_{rw} + \mathcal{F}_{tw}\right)\right)}{\mathcal{D}} \right) G_0 \\
& + \left( \frac{\mathcal{F}_{gt}\left(\mathcal{F}_{tr}\mathcal{F}_{rw} + \mathcal{F}_{tg}\mathcal{F}_{gw} + \mathcal{F}_{tw}\right) + \mathcal{F}_{gw}\left(1 - \mathcal{F}_{tr}\mathcal{F}_{rt} - \mathcal{F}_{tg}\mathcal{F}_{gt}\right)}{\mathcal{D}} \right) W_0
\end{aligned} \tag{C34}
$$

The denominator is expressed as following:

$$
\begin{aligned}
\mathcal{D} = \ & \left(1 - \mathcal{F}_{wr}\mathcal{F}_{rw} - \mathcal{F}_{wg}\mathcal{F}_{gw} - \mathcal{F}_{ww}\right)\left(1 - \mathcal{F}_{tr}\mathcal{F}_{rt} - \mathcal{F}_{tg}\mathcal{F}_{gt}\right) \\
& - \left(\mathcal{F}_{wr}\mathcal{F}_{rt} + \mathcal{F}_{wg}\mathcal{F}_{gt} + \mathcal{F}_{wt}\right)\left(\mathcal{F}_{tr}\mathcal{F}_{rw} + \mathcal{F}_{tg}\mathcal{F}_{gw} + \mathcal{F}_{tw}\right)
\end{aligned} \tag{C35}
$$





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





### a. Without explicit high vegetation

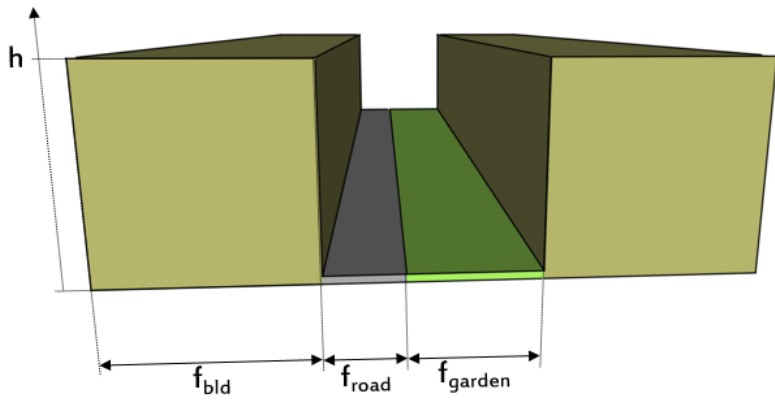

### b. With explicit high vegetation

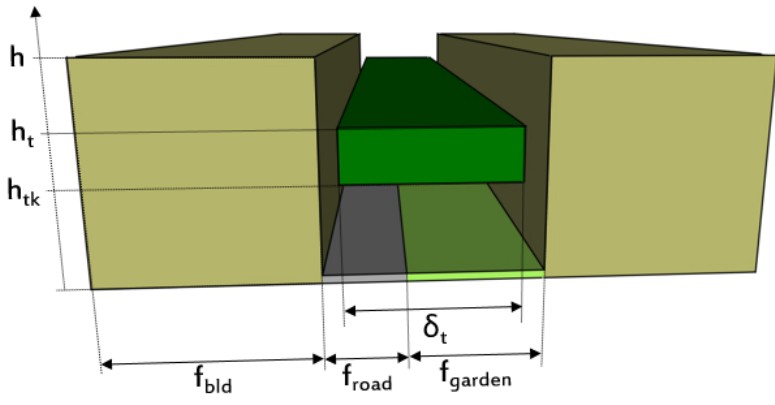

**Figure 1.** Comparison of the spatial arrangement of elements composing the urban canyon and of associated geometric parameters applied in the TEB model in the reference case (top) and in the case with explicit high vegetation (bottom).





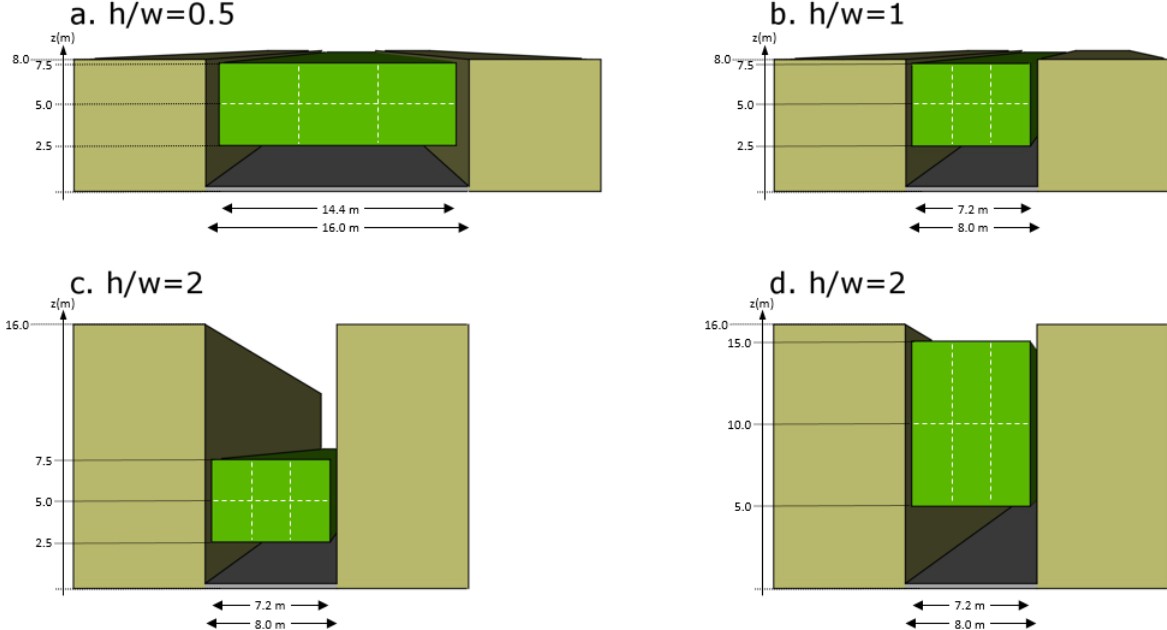

**Figure 2.** Description of simple geometries of urban canyon selected for the comparison between TEB and SOLENE simulations. For each of them, the potential location of tree canopy is illustrated by dotted rectangles.





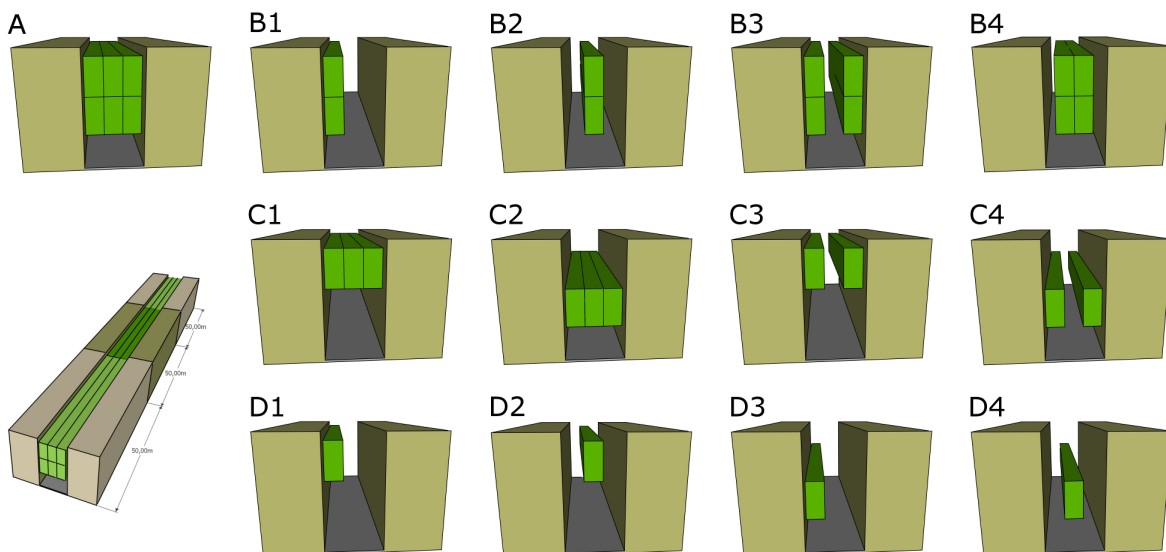

**Figure 3.** Description of the SOLENE mock-up, and presentation of the ensemble of vegetation layouts selected for the comparison between TEB and SOLENE simulations. The cases are presented here for the example of urban canyon with $h/w = 2$ and doubled thickness of crowns.







**Figure 4.** Comparison of TEB and SOLENE simulations of hourly direct (top) and diffuse (middle) solar radiation received by urban facets before reflections and total solar radiation absorbed by urban facets after interreflections (bottom), for urban canyons without vegetation. The results are presented here only for the aspect ratio equal to 1 and for the four street orientations at summertime.



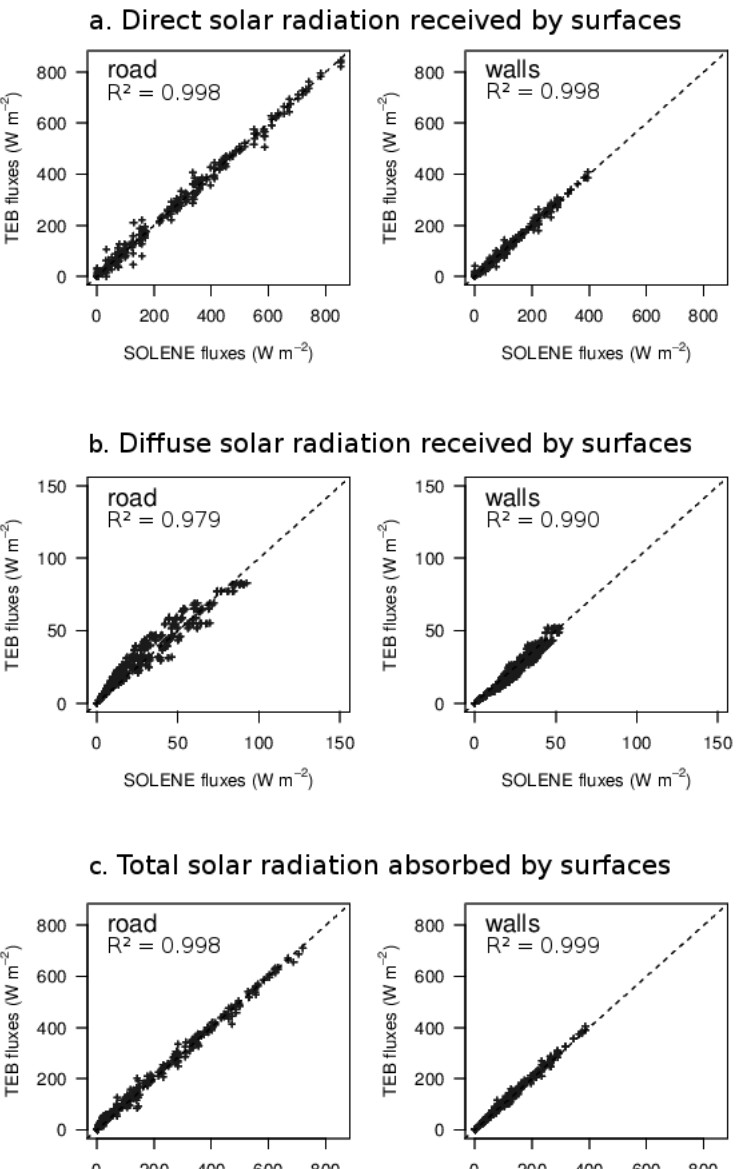

**Figure 5.** Scatterplots comparing TEB and SOLENE simulations of hourly direct (top) and diffuse (middle) solar radiation received by urban facets before reflections and total solar radiation absorbed by urban facets after interreflections (bottom), for urban canyons without vegetation. Each scatterplot gathers the hourly fluxes simulated for the four seasons, the three aspect ratios, and the four street orientations.





**Figure 6.** Comparison of TEB and SOLENE simulations of hourly direct (top) and diffuse (middle) solar radiation received by urban facets before reflections and total solar radiation absorbed by urban facets after interreflections (bottom), for urban canyons with vegetation. The results are presented here only for the aspect ratio equal to 1 and for the four street orientations at summertime.





**Figure 7.** Scatterplots comparing TEB and SOLENE simulations of hourly direct (top) and diffuse (middle) solar radiation received by facets before reflections and total solar radiation absorbed by facets after interreflections (bottom), for urban canyons with vegetation. Each scatterplot gathers the hourly fluxes simulated for the four seasons, the four aspect ratios, and the four street orientations.





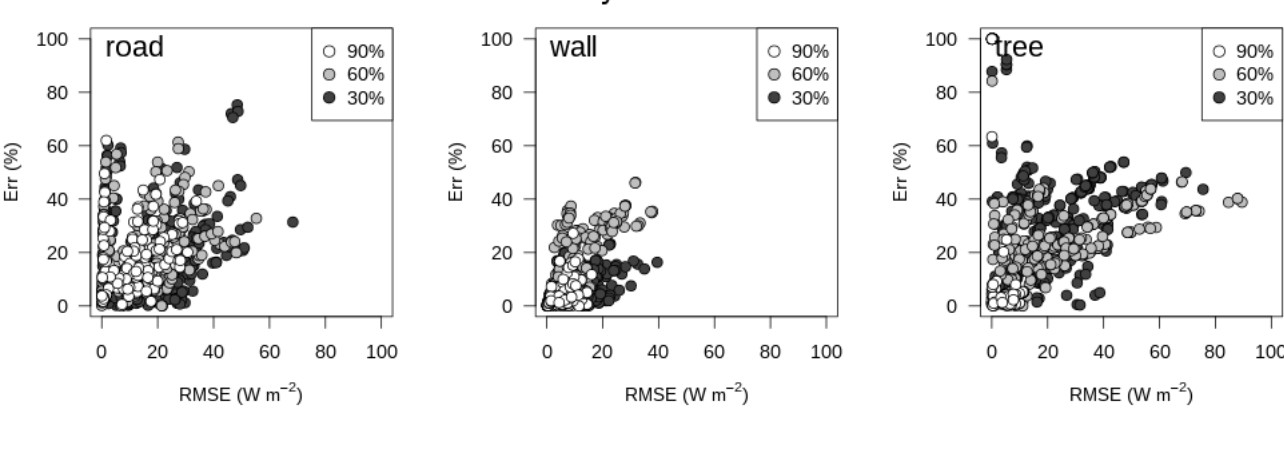

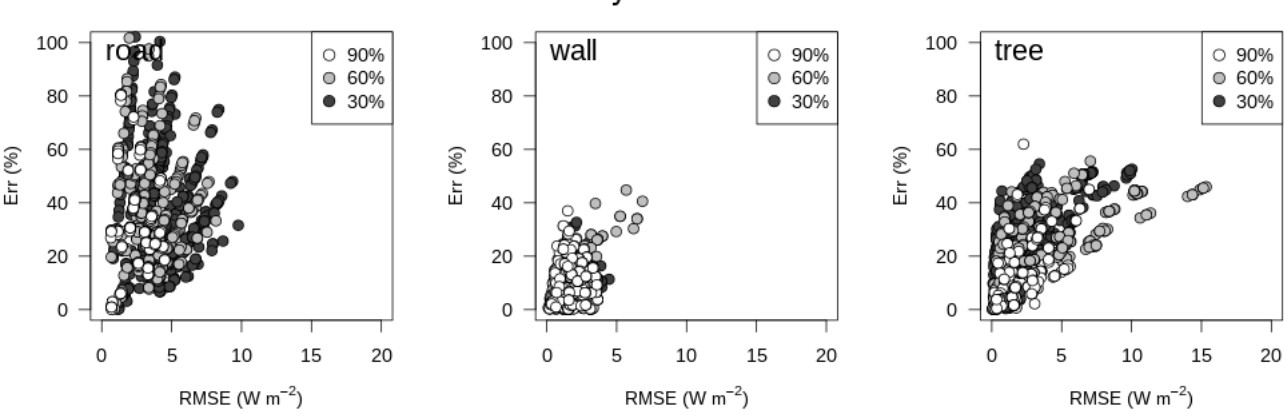

**Figure 8.** Effect of horizontal cover fraction of tree canopy on relative error and root mean squared error. Vegetation layouts are classified in three groups according to the cover fraction: 30% for layouts B1, B2, D1, D2, D3, D4; 60% for layouts B3, B4, C3, C4; and 90% for layouts A, C1, C2.




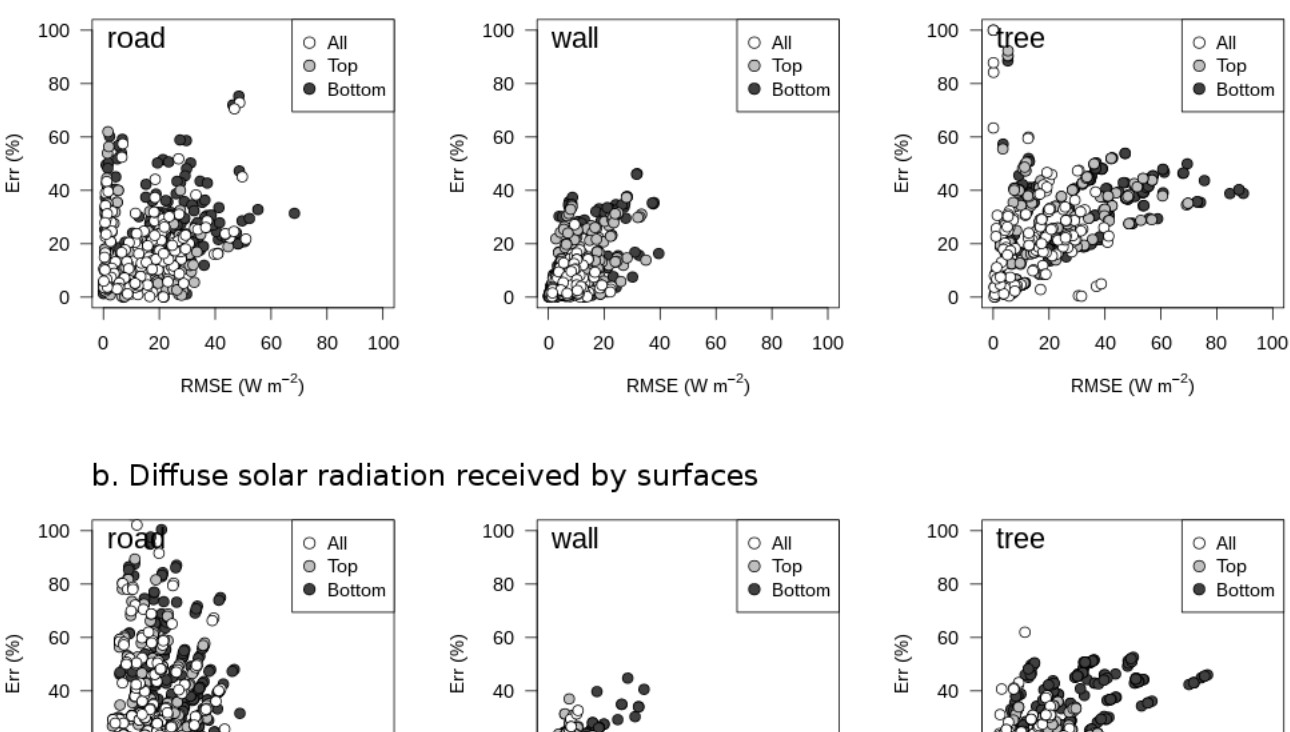

**Figure 9.** Effect of location of tree canopy on the vertical plane on relative error and root mean squared error. Vegetation layouts are classified in three groups: *Bottom* when the foliage is in the middle part of the canyon (layouts C2, C4, D3, D4); *Top* when the foliage is in the higher part of the canyon (layouts C1, C3, D1, D2); and *All* when the foliage extends in the high half of the canyon (layouts A, B1, B2, B3, B4).




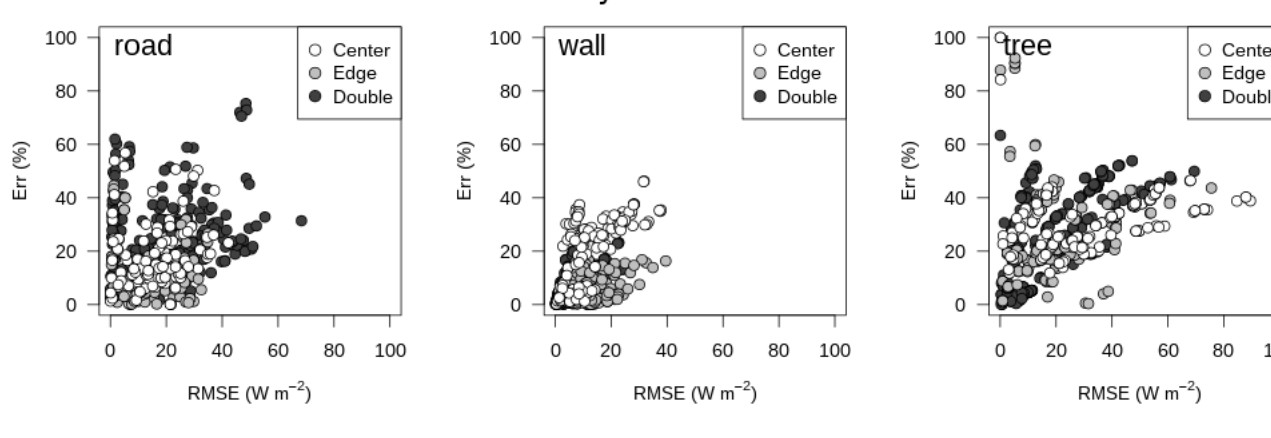

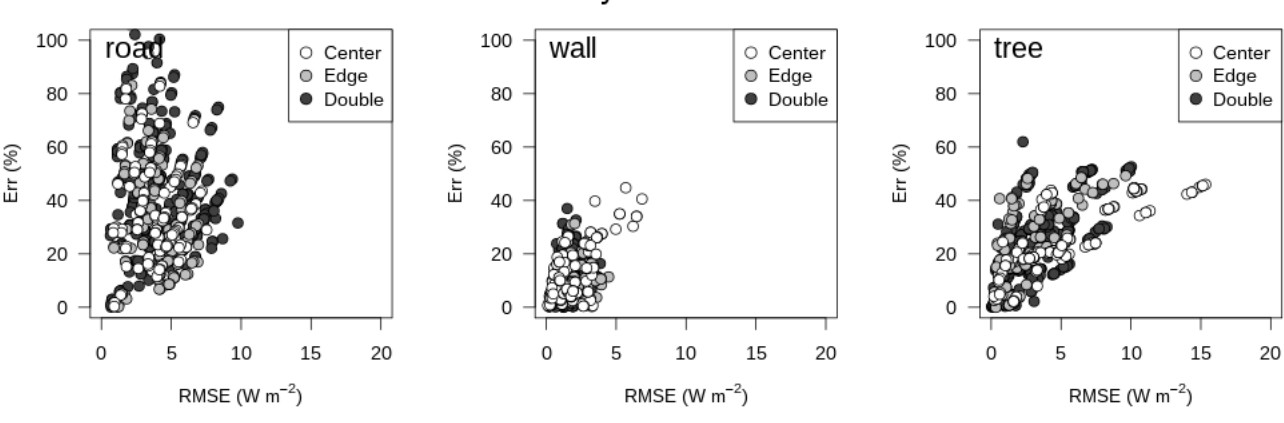

**Figure 10.** Effect of location of tree canopy on the horizontal plane on relative error and root mean squared error. Vegetation layouts are classified in three groups: *Center* when the foliage is centered along the canyon axis (layouts A, B2, B4, C1, C2, D2, D4); *Edge* when the foliage is on the edge of the canyon (layouts B1, D1, D3); and *Double* when the foliage is separated in two lines (layouts B3, C3, C4).





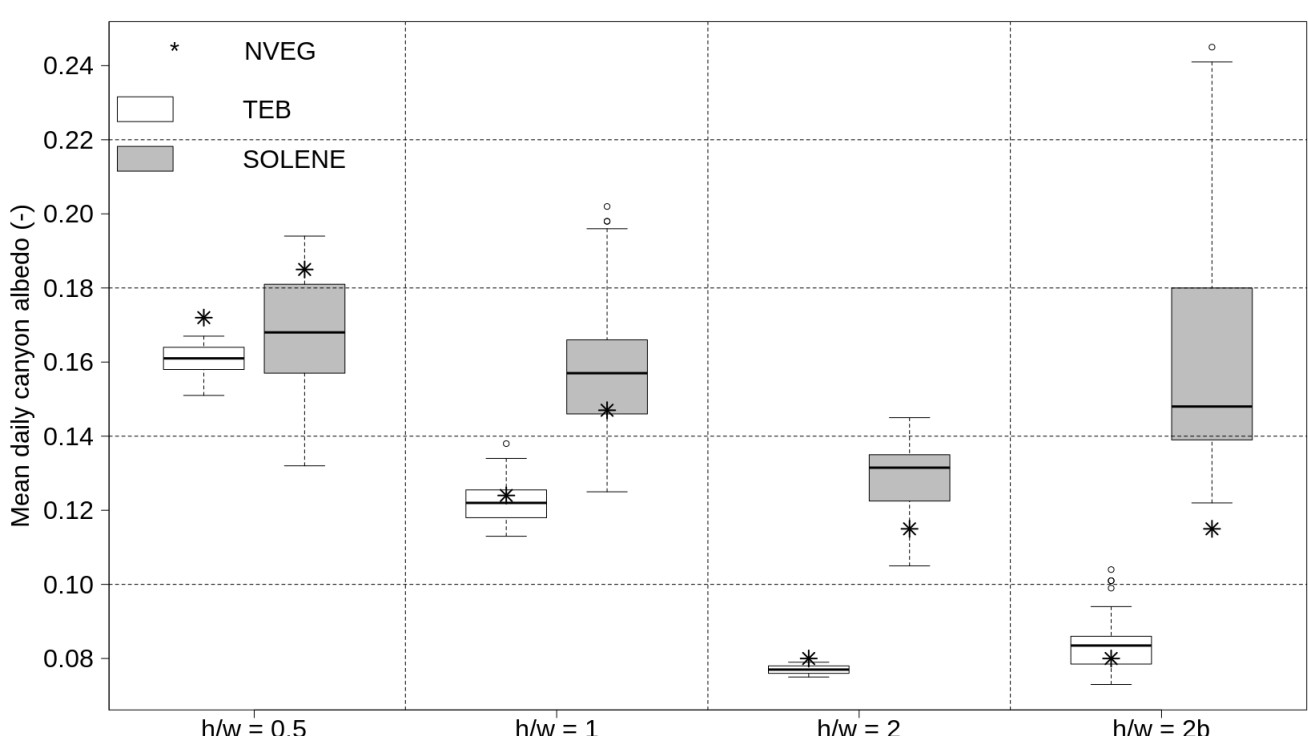

**Figure 11.** Comparison of the mean daily canyon albedo between TEB and SOLENE models by aspect ratios for all orientations and vegetation layouts confounded at summertime.





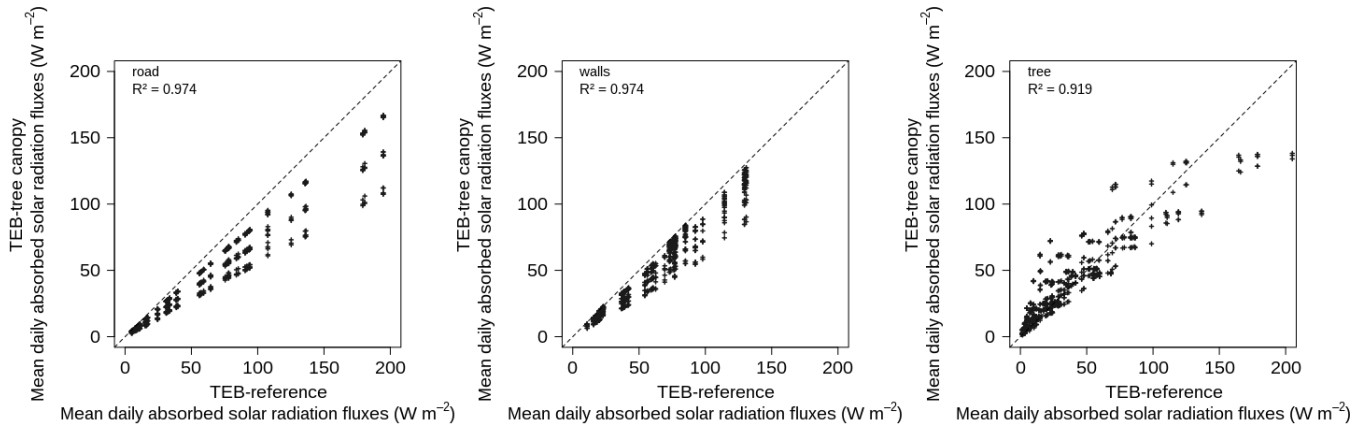

**Figure 12.** Comparison of mean daily total solar radiation absorbed by road, walls, and tree between the reference TEB simulation without explicit high vegetation and the new version including a tree canopy.



**Table 1.** Main descriptive parameters of the urban canyon, including vegetation, in the TEB model. The bold parameters are the input data prescribed by user, the other ones are computed in the model using the input parameters.

| Parameters | Symbol | Unit |
|---|---|---|
| Cover fraction of buildings | $\mathbf{f_{bld}}$ | - |
| Cover fraction of ground-based natural covers (garden) | $\mathbf{f_{garden}}$ | - |
| Proportion of bare soil in gardens | $\boldsymbol{\delta_n v}$ | - |
| Proportion of low vegetation in gardens | $\delta_{lv} = 1 - \delta_{nv}$ | - |
| Cover fraction of tree canopy | $\boldsymbol{\delta_t}$ | - |
| Mean building height | $\mathbf{h}$ | m |
| Wall plan area ratio | $\mathbf{r_w}$ | - |
| Canyon aspect ratio | $h/w = 0.5 r_w / (1 - f_{bld})$ | - |
| Height of tree canopy | $\mathbf{h_t}$ | m |
| Height of trunk | $\mathbf{h_{tk}}$ | m |
| Mid-height of tree's crown | $h_{cw} = (h_t + h_{tk})/2$ | m |
| Sky-view factor of wall, road, garden, tree | $\Psi_w s, \Psi_r s, \Psi_g s, \Psi_t s$ | - |
| View factor of road, tree from wall | $\Psi_{wr}, \Psi_{wt}$ | - |
| View factor of wall, tree from road | $\Psi_{rw}, \Psi_{rt}$ | - |
| View factor of wall,road from tree | $\Psi_{tw}, \Psi_{tr}$ | - |
| Leaf Area Index of low vegetation in gardens | $\mathbf{LAI_g}$ | $m^2 m^{-2}$ |
| Leaf Area Index of tree canopy | $\mathbf{LAI_t}$ | $m^2 m^{-2}$ |
| Leaf Area Density of tree canopy | $LAD_t$ | $m^3 m^{-2}$ |
| Albedo of wall, road, garden, tree | $\boldsymbol{\alpha_w, \alpha_r, \alpha_g, \alpha_t}$ | - |
| Emissivity of wall, road, garden, tree | $\boldsymbol{\epsilon_w, \epsilon_r, \epsilon_g, \epsilon_t}$ | - |





**Table 2.** List of input parameters for the ensemble of simulations performed with TEB.

| Parameters | A | B1 | B2 | B3 | B4 | C1 | C2 | C3 | C4 | D1 | D2 | D3 | D4 |
|---|---|---|---|---|---|---|---|---|---|---|---|---|---|
| $f_{bld}$ | 0.5 | 0.5 | 0.5 | 0.5 | 0.5 | 0.5 | 0.5 | 0.5 | 0.5 | 0.5 | 0.5 | 0.5 | 0.5 |
| $f_{garden}$ | 0. | 0. | 0. | 0. | 0. | 0. | 0. | 0. | 0. | 0. | 0. | 0. | 0. |
| $\delta_t$ | 0.9 | 0.3 | 0.3 | 0.6 | 0.6 | 0.9 | 0.9 | 0.6 | 0.6 | 0.3 | 0.3 | 0.3 | 0.3 |
| h | | | | 8. (for $h/w = 0.5$ , $h/w = 1$ , $h/w = 2$ (case a)) | | | / | 16. (for $h/w = 2$ (case b)) | | | | | |
| $r_w$ | | | 0.5 (for $h/w = 0.5$) | | / | 1.0 (for $h/w = 1$) | | / | 2.0 (for $h/w = 2$ (cases a and b)) | | | | |
| $h_t$ | 7.5 | 7.5 | 7.5 | 7.5 | 7.5 | 7.5 | 5.0 | 7.5 | 5.0 | 7.5 | 7.5 | 5.0 | 5.0 |
| $h_{tk}$ | 2.5 | 2.5 | 2.5 | 2.5 | 2.5 | 5.0 | 2.5 | 5.0 | 2.5 | 5.0 | 5.0 | 2.5 | 2.5 |
| $LAI_t$ | 3. | 3. | 3. | 3. | 3. | 3. | 3. | 3. | 3. | 3. | 3. | 3. | 3. |
| $\tau_{sr}$ | 0.5 | 0.5 | 0.5 | 0.5 | 0.5 | 0.5 | 0.5 | 0.5 | 0.5 | 0.5 | 0.5 | 0.5 | 0.5 |
| $\alpha_r$ | 0.25 | 0.25 | 0.25 | 0.25 | 0.25 | 0.25 | 0.25 | 0.25 | 0.25 | 0.25 | 0.25 | 0.25 | 0.25 |
| $\alpha_w$ | 0.30 | 0.30 | 0.30 | 0.30 | 0.30 | 0.30 | 0.30 | 0.30 | 0.30 | 0.30 | 0.30 | 0.30 | 0.30 |
| $\alpha_t$ | 0.15 | 0.15 | 0.15 | 0.15 | 0.15 | 0.15 | 0.15 | 0.15 | 0.15 | 0.15 | 0.15 | 0.15 | 0.15 |

**Table 3.** Statistical scores for absorbed solar radiation by surfaces.

| | Winter | | | Spring | | | Summer | | | Autumn | | |
|---|---|---|---|---|---|---|---|---|---|---|---|---|
| Config - Surf | RMSE | Err | Bias | RMSE | Err | Bias | RMSE | Err | Bias | RMSE | Err | Bias |
| Units | W m$^{-2}$ | % | W m$^{-2}$ | W m$^{-2}$ | % | W m$^{-2}$ | W m$^{-2}$ | % | W m$^{-2}$ | W m$^{-2}$ | % | W m$^{-2}$ |
| **NVEG** | | | | | | | | | | | | |
| Road | 4.35 | 43 | +2.02 | 13.91 | 18 | +4.47 | 9.88 | 3 | +3.50 | 9.37 | 17 | +4.21 |
| Walls | 1.94 | 4 | +0.42 | 6.37 | 2 | +0.91 | 5.72 | 3 | +2.80 | 3.72 | 2 | +1.06 |
| **ALL** | | | | | | | | | | | | |
| Road | 5.41 | 109 | +2.60 | 19.43 | 64 | +9.12 | 26.44 | 24 | +15.37 | 17.11 | 64 | +8.97 |
| Walls | 5.01 | 14 | +1.77 | 15.36 | 14 | +5.39 | 20.60 | 15 | +9.09 | 14.53 | 14 | +5.47 |
| Tree | 9.00 | 44 | -1.55 | 28.82 | 28 | -8.52 | 42.74 | 22 | -17.90 | 28.70 | 28 | -8.53 |

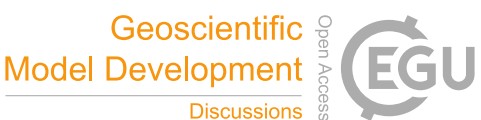

**Table 4.** Statistical scores for direct solar radiation received by surfaces before reflections for summertime.

| Config | Road RMSE | Err | Bias | Walls RMSE | Err | Bias | Tree RMSE | Err | Bias |
|---|---|---|---|---|---|---|---|---|---|
| Units | W m$^{-2}$ | % | W m$^{-2}$ | W m$^{-2}$ | % | W m$^{-2}$ | W m$^{-2}$ | % | W m$^{-2}$ |
| NVEG | 8.91 | 1 | -0.28 | 4.54 | 1 | +0.40 | - | - | - |
| A | 18.32 | 14 | +8.55 | 6.86 | 5 | -1.55 | 10.04 | 4 | -5.43 |
| B1 | 22.78 | 9 | +8.93 | 21.12 | 6 | +2.12 | 41.38 | 32 | -13.17 |
| B2 | 45.16 | 26 | +22.06 | 15.10 | 8 | +2.25 | 47.44 | 37 | -26.76 |
| B3 | 31.23 | 22 | +16.14 | 21.80 | 20 | +10.14 | 62.28 | 30 | -36.78 |
| B4 | 40.54 | 29 | +20.01 | 12.02 | 8 | -0.94 | 32.30 | 17 | -18.35 |
| C1 | 18.06 | 14 | +8.38 | 6.86 | 5 | -1.57 | 9.66 | 4 | -5.18 |
| C2 | 25.90 | 21 | +13.20 | 10.83 | 9 | -4.98 | 7.49 | 3 | -3.17 |
| C3 | 16.08 | 9 | +7.64 | 18.31 | 16 | +8.67 | 43.87 | 22 | -25.24 |
| C4 | 21.82 | 13 | +10.93 | 9.22 | 4 | +1.58 | 25.15 | 18 | -13.91 |
| D1 | 23.23 | 6 | +6.28 | 18.74 | 5 | +1.29 | 31.39 | 29 | -8.86 |
| D2 | 31.24 | 17 | +14.98 | 15.09 | 8 | +2.23 | 35.87 | 30 | -19.56 |
| D3 | 12.91 | 4 | +3.82 | 14.45 | 4 | -1.68 | 29.52 | 20 | -0.21 |
| D4 | 38.66 | 21 | +18.24 | 9.32 | 5 | -3.57 | 25.90 | 23 | -11.11 |





**Table 5.** Statistical scores for diffuse solar radiation received by surfaces before reflections for summertime.

| Config | Road | | | Walls | | | Tree | | |
|--------|------|-----|------|-------|-----|------|------|-----|------|
| | RMSE | Err | Bias | RMSE | Err | Bias | RMSE | Err | Bias |
| Units | W m$^{-2}$ | % | W m$^{-2}$ | W m$^{-2}$ | % | W m$^{-2}$ | W m$^{-2}$ | % | W m$^{-2}$ |
| NVEG | 5.11 | 7 | +0.87 | 2.59 | 5 | -0.43 | - | - | - |
| A | 2.92 | 15 | +1.34 | 2.72 | 9 | +1.07 | 5.42 | 20 | -3.50 |
| B1 | 4.58 | 14 | +1.87 | 3.16 | 8 | +1.09 | 6.35 | 43 | -4.10 |
| B2 | 6.08 | 29 | +3.72 | 2.81 | 7 | +0.79 | 8.16 | 49 | -5.38 |
| B3 | 4.69 | 24 | +2.69 | 4.83 | 27 | +2.72 | 12.21 | 44 | -8.22 |
| B4 | 5.05 | 30 | +3.15 | 2.51 | 6 | -0.74 | 7.05 | 32 | -4.71 |
| C1 | 2.89 | 14 | +1.29 | 2.53 | 11 | -0.55 | 3.59 | 13 | +0.24 |
| C2 | 2.91 | 14 | +1.33 | 2.48 | 4 | -0.19 | 3.32 | 12 | -0.88 |
| C3 | 3.73 | 13 | +1.49 | 3.02 | 12 | +1.38 | 6.36 | 28 | -4.36 |
| C4 | 3.63 | 13 | +1.47 | 2.69 | 5 | +0.52 | 4.27 | 22 | -2.47 |
| D1 | 4.35 | 10 | +1.28 | 2.54 | 3 | +0.36 | 3.28 | 26 | -2.05 |
| D2 | 5.00 | 20 | +2.68 | 2.29 | 5 | +0.25 | 4.83 | 35 | -3.25 |
| D3 | 4.23 | 9 | +1.08 | 2.55 | 3 | -0.11 | 2.13 | 17 | -0.86 |
| D4 | 5.42 | 22 | +3.01 | 2.48 | 5 | -0.65 | 3.10 | 27 | -1.72 |





**Table 6.** Statistical scores for total (direct and diffuse) solar radiation absorbed by surfaces after reflections for summertime.

| | Road | | | Walls | | | Tree | | |
|---|---|---|---|---|---|---|---|---|---|
| Config | RMSE | Err | Bias | RMSE | Err | Bias | RMSE | Err | Bias |
| Units | W m$^{-2}$ | % | W m$^{-2}$ | W m$^{-2}$ | % | W m$^{-2}$ | W m$^{-2}$ | % | W m$^{-2}$ |
| NVEG | 9.88 | 3 | +3.50 | 5.72 | 3 | +2.80 | - | - | - |
| A | 19.67 | 27 | +12.88 | 20.15 | 19 | +10.18 | 14.53 | 5 | -4.80 |
| B1 | 21.00 | 14 | +11.76 | 22.16 | 16 | +11.85 | 41.70 | 29 | -18.38 |
| B2 | 39.02 | 32 | +23.39 | 21.43 | 12 | +10.26 | 52.64 | 41 | -30.90 |
| B3 | 33.61 | 36 | +21.34 | 38.51 | 47 | +22.10 | 86.57 | 38 | -52.53 |
| B4 | 36.65 | 39 | +22.50 | 20.28 | 14 | +8.87 | 35.32 | 19 | -19.48 |
| C1 | 18.18 | 23 | +11.82 | 13.59 | 11 | +5.49 | 16.71 | 8 | -3.50 |
| C2 | 24.27 | 31 | +15.61 | 5.86 | 5 | -0.26 | 13.99 | 4 | +1.79 |
| C3 | 19.71 | 20 | +12.96 | 30.31 | 32 | +17.37 | 68.13 | 31 | -40.47 |
| C4 | 23.59 | 23 | +15.44 | 18.87 | 13 | +10.02 | 41.20 | 24 | -21.58 |
| D1 | 20.13 | 11 | +9.23 | 16.77 | 9 | +7.59 | 30.17 | 27 | -10.39 |
| D2 | 27.11 | 21 | +16.49 | 17.58 | 9 | +7.54 | 37.21 | 29 | -20.31 |
| D3 | 13.87 | 9 | +7.15 | 10.72 | 5 | +4.53 | 24.50 | 8 | -1.04 |
| D4 | 32.65 | 25 | +19.19 | 8.27 | 4 | +2.61 | 23.95 | 23 | -11.10 |