# Peer review of "Implementation of street trees in solar radiative exchange parameterization of TEB in SURFEX v8.0"

_Geoscientific Model Development, 2016_

## Referee Comment (RC1) · Anonymous Referee #1 · 26 Aug 2016

The paper presents a new parameterization for street trees within the solar radiative exchange scheme in the existing Town Energy Balance (TEB) scheme. The authors provide a clear motivation as to why trees warrant inclusion in urban land surface schemes and critique the latest modelling techniques for representing them. A brief overview of the TEB model and how it treats solar radiation is presented before the new street tree parameterization is described and how its formulation is implemented within the existing scheme (good use of appendices to present the full equations). Comparison and evaluation is made against a complex microscale radiative transfer scheme (SOLENE) for a range of tree configurations (13) and three different canyon aspect ratios, as well as an existing version of TEB with low vegetation. The evaluation focused on the seasonal differences between models in solar receipt for canyon facets for non-vegetated and vegetated canyons, sensitivity to the vegetation layout and comparison

of canyon albedo. Results are presented in a number of tables and graphs which on the whole are intuitive, clearly presented and discussed in sufficient detail in the results section. However the section on sensitivity to vegetation layout characteristics requires further work. The discussion and conclusion clearly highlight the differences between the models, while exhuming the benefits and limitations of the scheme.

Overall the paper is very well written and adheres to the requirements for publication within Geoscientific Model Development. There are a few points within the paper that require clarification from the authors coupled with a few minor grammatical and presentation errors that require correction (please see below). With these points satisfactorily answered and errors amended I recommend this paper be accepted for publication in the journal.

Points for Correction and Clarification

1) At a number of points you refer to keeping computing times acceptable (e.g. Page 1, Line 7; Page 6, Line 21). It is unclear what is an acceptable computing time is in this context. Do you mean relative to a complex model? How is the representation of this process going to significantly add to computing time? Surely representing the process correctly is more important in the first instance, and computing time shouldn't determine a limit on how we approach a problem from a modelling perspective as computational optimisation and simplification can be applied later.

2) Page3 Lines 27-31. Since the conference paper of Young et al. (2015) there have been a number of developments relating to the Trees in Urban Areas model (TUrban). The tree representation is not currently implemented within MORUSES (Met Office – Reading Urban Surface Exchange Scheme). Instead the scheme has been developed and is currently been tested within the Single Column Reading Urban Model (SCRUM) as described in Harman and Belcher (2006) and Porson et al. (2009). The view factors for both fully visible and partially occluded facets are calculated analytically based on Hottel's crossed string construction (Hottel 1954). A paper is about to be submitted on

this method to Boundary-Layer Meteorology but unfortunately to have gone through the review process in time for this papers publication (Young et al, 2016). These sentences need amending in light of this new information.

3) Section 3.2. You talk about improvements of the radiation budget but only mention the longwave radiation budget on Page 6 Line 3. How is this to be modelled for the high vegetation? Although the paper is focused on shortwave radiation it would add to the paper if you described briefly how longwave radiation is treated.

4) Page 6, Lines 24 – 30. The hypothesis about street trees and why they are confined within the canyon is not particularly compelling without explicit examples of such rules on tree management and location. Is this specific to France? Surely the justification is more to do with the model assumptions of the canyon and roof being treated separately in the modelling of the surface energy balance in that the canyon and roof are assumed to be independent of each other? Either that or are you just considering the effects of trees that are not taller than the buildings, as to consider these is not possible within the current TEB configuration?

5) Page 16, Lines 9-10. The use of the words 'good' and 'only' are very subjective. You need to state what is it good relative to?

6) Section 6.4. This section on sensitivity to vegetation layout characteristics and associated figures is not particularly clear. Statements are made without the use of statistical values nor clear comparison using examples from the relevant figures (Figs 8 – 10). Page 16 Line 19 states 'The comparison of statistical scores', these statistics need to be presented within the text or in a table.

7) Section 6.4. Figures 8, 9 and 10 do not show any particular clear patterns (there is a lot of scatter and points on top of each other) that allow the reader to determine the full impact of differences in vegetation or whether it was significant, an alternative method for showing this data is required. It is unclear of the utility of comparing two types of error (RMSE and %Err). Could you clarify why you have done this and how this shows

how sensitive the model is to these changes? An explanation of this is required in the text.

8) Section 6.5, Figure 11. There are four aspect ratios presented in figure 11 what is the difference between h/w = 2 and h/w = 2b? The second is not referred to in the text nor the figure caption.

9) Appendix A. Why is the view factor between the road and the tree = 0? Surely this would have an implication when calculating the canyon longwave radiation balance as the road will see the underside of the tree layer.

Grammatical and Format Errors

1) TITLE. The authors may wish to amend the title to read more clearly as 'Implementation of street trees within the solar radiative exchange parameterization of TEB in SURFEX v8.0'.

2) ABSTRACT, Line 1. With the first use of an abbreviation the term/phrase/name it is used to abbreviate should be stated e.g. Town Energy Balance (TEB) or TEB (Town Energy Balance). The same applies to SOLENE.

3) ABSTRACT, Line 3. The word 'obviously' is not required here as it is not obvious without reading the paper that there is increased complexity. It would be more appropriate to state this as a fact by removing the words 'has obviously'.

4) Page 2, Line 34. 'Surimpose' should read 'superimpose'.

5) Page 5, Line 13. The work 'especially' is not required in this sentence.

6) Page 6, Line 32. Should the word 'refined' be 'defined'?

7) Page 7, Lines 7 -10. This paragraph is not clear. Should it read as follows? 'In order to calculate these terms in TEB, the following section describes how direct solar radiation reaches canyon surfaces. Then, absorption is obtained by separately resolving the first absorption of total solar radiation on each surface and the sum of absorbed

shortwave radiation after infinite reflections within the canyon'.

8) Page 11, Line 21. Should the first word be TEB on this line?

9) Page 12, Line 15. This first sentence doesn't read well. Consider splitting into two sentences, one explaining that TEB was run with equivalent configurations and another stating the differences between models.

10) Page 13, Line 15. Use of the word 'The' is redundant in this sentence. Start with 'Table 3 presents'.

11) Page 13, Line 20. 'Let's remind' is not the correct style nor grammatically correct (Let's = Let is. If using this use 'Lets'). You may consider the following change to the sentence. 'Considering that the temperate climate is characterized by four distinct seasons with contrasting sunshine, air temperature and humidity conditions, seasonal analysis was undertaken'.

12) Page 14, Line 27. Incorrect style and grammar using 'let's remind'. The word 'considering' would be more suitable.

13) Page 14, Lines 27-28. This sentence doesn't make sense.

14) Page 16, Line 5. I am not sure what you mean by 'inversely slightly underestimated'?

15) Page 16, Line 20. Misspelt 'exchanges'

16) Page 17, Line 9. A missing paper reference at the end of this line.

17) Page 17, Lines 9-10. Acronyms for models without full model names (as in point 2).

18) Page 20, Line 2. Missing equation reference.

19) Page 22, Line 1. Misspelt 'example'.

20) Figures 4 and 6. The shades of blue used to represent 'wall A', 'wall B' and 'walls'

are not clear and will not reproduce well if printed in black and white. Consider changing colours or using different line thickness. The subplots are also too small, consider reducing white space between subplots. This coul be achieved by limiting the number of axis labels especially as you are using the same scale and variable on each row.

References

Harman, I. N. and Belcher S. E. 2006. The surface energy balance and boundary layer over urban street canyons. Quarterly Journal of the Royal Meteorological Society, 132, 2749 – 2768.

Hottel, H. 1954. Radiant-Heat Transmission. In: Heat Transmission. Ed. McAdams W., McGraw-Hill, New York, pp 55 – 125.

Porson, A., Harman, I. N., Bohnenstengel, S. I., and Belcher, S. E. 2009. How many facets are needed to represent the surface energy balance of an urban area? Boundary-Layer Meteorology, 132, 107 – 128.

Young, D. T., Clark, P. A., Barlow, J.F. and Hendry, M. 2016. An analytical method for radiative exchange within a vegetated urban canyon. To be submitted to: Boundary-Layer Meteorology

---

## Referee Comment (RC2) · Anonymous Referee #2 · 6 Sep 2016

**Review of "Implementation of street trees in solar radiative exchange parameterization of TEB in SURFEX v8.0" by E. Redon et al.**
*Geosci Model Dev,* doi:10.5194/gmd-2016-157

**Overview**

This article presents a new development to the TEB urban canopy model: addition of effects of trees on shortwave radiation exchange in urban canyons. It appropriately reviews the literature and makes a good case for the importance of this development. The paper discusses the simplifications and assumptions made to generate a fit-for-purpose model addition. It then exhaustively compares the new TEB model with added trees to the old (without trees) as well as to a higher resolution model (SOLENE). The model comparison appears to have been a very large undertaking, and the results are nicely presented. However, there are some significant concerns related to the model formulation, as well as its neighbourhood-scale performance. It may be that the model formulation requires correction and/or improvement. This article has the potential to be a strong contribution, provided the major comments (below) are adequately addressed. Major comment #2 is particularly critical.

**Major comments**

**1. English.** Ideally, the English should be edited by a native English speaker.

**2. Error in application of Beer's law.** Equations 3-5 appear to be mixed up. Eq. 4 with albedo removed should be the transmission; Eq. 3 as written is the intercepted radiation (multiply it by scattering coefficient to equal scattered radiation); Eq. 5 should be modified accordingly. Please check your model implementation carefully to ensure that it is correct according to the updated equations, and redo the simulations if the model equations were incorrect.

**3. Neglect of forward scattering.** From Eqs. 3-6, it appears that forward scattering of intercepted radiation by vegetation is neglected (see Campbell and Norman 1989 for more detail; also, consider revising lines 7-8 on p. 10). Perhaps the albedo and extinction of the foliage are/can be adjusted to account for this. Either way, please explain and/or justify more fully. Can you assume all scattered radiation is scattered upward without introducing significant error? The broadband scattering coefficient is on the order of 0.50 for the leaves of many trees, and approximately 50% of this is forward scattered – so forward scattering potentially represents 25% of the shortwave radiative energy (very approximately). Interception by lower leaves of radiation forward scattered by upper leaves increases total absorption by tree foliage, which may correct the underestimation you find.

**4. Robustness of SOLENE as evaluation tool.** How accurate are the SOLENE calculations of solar absorption and scattering by trees? p. 11, lines 7-8: Treatment of foliage in the SOLENE should be further discussed, as well as any associated evaluation, given that this treatment is what TEB

is being compared to. Details are sparse in Robitu et al. (2006). Does it include forward scattering, multiple reflection between tree foliage and the urban canyon underneath, or between different tree foliage elements, for example? Differences in model geometries between TEB and SOLENE are discussed; how different are the physics?

**5. Large overall albedo difference.** The large differences in mean albedo for higher H/W canyons in Fig. 11 are worrying, and suggests to this reviewer than one or more assumptions made in the formulation in the TEB model are inadequate (assuming the SOLENE model is robust – to point 4 above). It could relate to one of points 2 or 3 above. My sense is that tree foliage limited to the canyon should become less important as H/W increases – if that is true, why would performance degrade? Could this be an issue with the TEB shortwave radiation scheme without trees? A primary purpose of TEB is to provide neighbourhood-scale fluxes to the overlying atmospheric model, and overall albedo is the parameter to which the energy balance is often most sensitive. Hence, this result requires more investigation.

**Minor comments**

p. 1, line 8: "…uncertainties in terms of the solar radiative exchanges, as quantified by comparison of TEB…"

p.1, line 18: remove "soil artificialisation due to"

p.1, line 22: refresh, or cool, clean/filter, etc?

p. 2: "surimpose" is not a word I don't think – do you mean "shade"?  And again, p. 6, line 18 I think "shade" or a similar word (obscure?) might be better than "superimpose" (please check throughout).

p. 3, line 2: "…by a microscale radiation model: SOLENE…"

p. 3, line 12: Krayenhoff et al (2014)

p. 4, line 4: "…radiative (Krayenhoff et al. 2014) and dynamic (Krayenhoff et al. 2015) effects…"; the reference for BEP-Tree is "Krayenhoff (2015)" at present.

p. 4, line 6: "…both within and above the canyon and above roofs."

p. 4, lines 14-19: There are some significant assumptions in the view factor calculations and radiation exchange in ENVI-met that could be discussed. However, since it is a microscale model, it may not be relevant to go into much detail.

p. 4, lines 28-29: I suggest beginning the paragraph as follows: "At each mesoscale model grid point, TEB describes the average characteristics of the local environment by a single urban canyon…"

p. 8, line 6: This assumes tree foliage is uniformly distributed across the canyon, if I understand correctly? If so, this is worth stating in the text.

p. 11, lines 3-6: Is this relevant, if TEB assumes an isotropic distribution? Presumably the same is chosen in SOLENE?

Eq. 19: RMSD (difference, not error, since you are comparing two models); also note the critiques of RMSD relative to MAE (Mean Absolute Error), e.g. Willmott et al. 2009. I suggest calculation of MAE instead.

p. 17, line 9: Reference missing.

p. 26, line 27: Year is 2015, not 2014.

p. 26, line 29: Year is 2014, not 2013.

Appendix A, line 1: Eq. ??; and again line 16 on p. 21.

Appendix A, line 19: View factor from road to trees is zero??? Please explain!

Appendix A, line 8: LAD is in m2 of leaf area per m3 of volume.

**References**

Willmott CJ, Matsuura K, Robeson SM, 2009: Ambiguities inherent in sums-of-squares-based error statistics. *Atmos Environ* **43**, 749-752.

---

## Author Comment (AC1) · 28 Oct 2016

SUMMARY OF CHANGES

Firstly, we would like to express our thankfulness and appreciation to the reviewers for their useful comments to improve the paper. We have addressed all the comments as explained below.

GENERAL COMMENT: Major changes in the new version of the article

Considering the useful comments of the Referees we proceed to some major changes in the article in order to provide full explanations of the purpose of our experiment and avoid confusion with future works (particularly about the longwave balance). The view factors are based on the equations of sky view factors in Masson, 2000. Those related

to the high vegetation are defined on all the directions. In this way, the interactions between all the facets of the canyon, the sky and trees are allowed in Eq. A6, A7, A8. Psi(wt) is now expressed for one wall. Specific coefficients are applied, in the shortwave scheme only, to constrain the reflections from the high vegetation toward sky and top part of walls. It ensure the compatibility with the calculations of the long-wave exchanges, for which the same view factors but no coefficients are involved. The statistics are based on the Mean Absolute Error instead of the Root Mean Square Error, according to the suggestion of the Referee 2 and the Willmott and Matsuura works (2005). The section about the sensibility of TEB model to the vegetation layouts have been removed in absence of clear and systematic behaviors in the simulations. We also permuted the figures 8 and 9 (after removing the former figures 8, 9, 10) and changed the figure 9 about the canyon albedos from a comparison SOLENE vs TEB to a comparison TEB reference vs TEB with a tree canopy layer. The physic assumptions (e.g., the nature of radiation, specular or isotropic, during the multiple reflections) are too different to allow an accurate comparison of absorption by canyon surfaces or canyon albedo. All the figures and text have been adapted.

We payed attention to express better the two major objectives of this work:

(1) technical objective of evaluating the geometrical assumptions related to the high vegetation by the confrontation with a model with explicit tree crowns where the radiative budget is solved at high resolution (through numerical meshed mock-ups);

(2) scientific objective for further studies of simulating different vegetation layouts, potentially including urban trees, to provide information about impacts of greening on microclimate, thermal comfort, energy demand by buildings and needs of water to maintain the vegetation at the city-scale.

REFEREE 1Ăă:

PART 1 : Points for Correction and Clarification

1. " At a number of points you refer to keeping computing times acceptable (e.g. Page 1, Line 7; Page 6, Line 21). It is unclear what is an acceptable computing time is in this context. Do you mean relative to a complex model? How is the representation of this process going to significantly add to computing time? Surely representing the process correctly is more important in the first instance, and computing time shouldn't determine a limit on how we approach a problem from a modelling perspective as computational optimisation and simplification can be applied later. "

The TEB model (integrated in the SURFEX land-surface modeling system) has been initially developed for mesoscale modeling, with the objective of including the cities in meteorological models and of being able to simulate the urban heat island. Such applications at city scale entail some constraints about the level of accuracy in the description of urban environment. The model does not pretend to explicitly describe the exact arrangement of streets and buildings, but adopts a simplified approach of mean urban canyons which enables to distinguish the main urban typologies. Even if we are working to improve the TEB model by including new physical processes that seem important to us especially for evaluating adaptation and mitigation strategies, we are committed to maintain a certain level of simplicity. Besides the numerical aspects and time computing, we think that uncertainties associated with the prescription of input data (maps of morphological parameters, materials properties, or vegetation characteristics) significantly limit the potential gain of a very complex modeling. Note that in case of very fine-scale studies, suitable models already exist, e.g. the CFD models, but they are run over much smallest areas.

See page 6 at lines 21-26 the revisions in the paper.

2. " Page3 Lines 27-31. Since the conference paper of Young et al. (2015) there have been a number of developments relating to the Trees in Urban Areas model (TUrban). The tree representation is not currently implemented within MORUSES (Met Office Reading Urban Surface Exchange Scheme). Instead the scheme has been developed and is currently been tested within the Single Column Reading Urban Model (SCRUM)

as described in Harman and Belcher (2006) and Porson et al. (2009). The view factors for both fully visible and partially occluded facets are calculated analytically based on Hottel's crossed string construction (Hottel 1954). A paper is about to be submitted on this method to Boundary-Layer Meteorology but unfortunately to have gone through the review process in time for this papers publication (Young et al, 2016). These sentences need amending in light of this new information. "

We apologize for our confusing description on TUrban developments. We noted the Referee's information and consequently amended the text.

See page 3 at lines 28-32 the revisions in the paper.

3. " Section 3.2. You talk about improvements of the radiation budget but only mention the longwave radiation budget on Page 6 Line 3. How is this to be modelled for the high vegetation? Although the paper is focused on shortwave radiation it would add to the paper if you described briefly how longwave radiation is treated. "

Here, we made the choice to focus on solar radiation budget for two main reasons. The comparison to the SOLENE model is not suitable for infrared radiation budget, because the two models deal differently with the resolution of surface turbulent exchanges and the calculation of surface temperatures of urban facets. The comparison SOLENE vs TEB for the calculation of infrared radiation fluxes becomes consequently tricky, since these fluxes are strongly dependent on surface emissions. The validation of our shortwave radiation scheme contributes to verify our future longwave radiation scheme because the same view factors are used for the calculations of multiple reflections in shortwave radiation but also infrared radiation exchanges. Specific coefficients are now applied, in the shortwave scheme only, to constrain the reflections from the high vegetation toward sky and top part of walls. It ensure the compatibility with the calculations of the longwave exchanges, for which the same view factors but no coefficients are involved. In nature, the solar radiation is mainly redirected upwards by the receiving face of sunlit leaves in the top of crown during the first reflection. We suggest here to neglect the small amount of shortwave radiation which the tree stratum is supposed to reflect to the low part of the canyon during multiple reflections in favor of representing realistically the upward first reflection of solar radiation, which is by far the most energetic reflection. On the contrary, emission of longwave radiation by the vegetation is intrinsically isotropic and can reach all the viewed canyon facets. In this case, all the leaves of the crown, statistically associated to all orientations, participate to the emission.

We added further explanations in the paper page 6, at lines 12-17 and page 10-11, at lines 26-28 and 1-5.

4. " Page 6, Lines 24 – 30. The hypothesis about street trees and why they are confined within the canyon is not particularly compelling without explicit examples of such rules on tree management and location. Is this specific to France? Surely the justification is more to do with the model assumptions of the canyon and roof being treated separately in the modelling of the surface energy balance in that the canyon and roof are assumed to be independent of each other? Either that or are you just considering the effects of trees that are not taller than the buildings, as to consider these is not possible within the current TEB configuration? "

Choices on model developments are firstly driven by observations. We refer to the typical positions of trees in urban environment. Depending on the urban form, we observe different vegetation layouts in the townscape. Large avenues of six or height-storey buildings are generally tree-filled with a double row of trees (e.g., Aesculus hippocastanum, Platanus x acerifolia) which the mature height is near the same of the buildings height, around 20 m. The top of trees is commonly cut in order to provide neat and homogeneous sky line and facilitate the long-term management. Along the second or third order streets axes, smaller trees are planted such as Tilia x euchlora or Acer platanoïdes 'Globosum'. At the center of urban groups, squares are greened with grass and trees. In addition to the road, the places are surrounded by wide walkways and recreational areas equipped by benches; so the distance from trees to buildings is significant. Some single specimens can be found in crossroads of boulevards and second order streets or at roundabouts. Each planting in the public area of cities is submitted to compliance with the local tree management rules. These documents are precisely specifying species chosen in adequacy with the aspect ratio of the street, function and aesthetic of the site. They ensure a satisfactory juxtaposition of trees with urban structures for the dwellers (by avoiding excessive shade of trees on facades or disruption of underground services by roots) but also prevent damages of roots by potential new soil removing. It is stated as fundamental urban design principle that streets with 1:1.4 aspect ratio are unsuitable for tree planting. Typically, trunks are planted at least at 3 to 5 meters from walls or balconies. The minimum distance between crowns and walls or balconies is 1.5 meters. In case of not appropriate tree planting, for example in the Queen's Park Estate where London planes would overwhelm the three-storey Victorian buildings of the estate, trees have been pollarded to just below eaves level. In suburban areas or private gardens, common-sense rules are applied in the same way in planting tree to several meters (at least the house height of distance) from walls to avoid obstruction of windows, damage of roots on water pipes or swimming-pools but also falling trees onto housings. Following these practices, we observe a great shading effect on facades but they generally can't shade roofs both in urban and suburban areas.

Even if some tall trees have been planted without respecting elementary requirements specifications, it is a statistically very limited design. It could be occur mainly in suburban or commercial zones. Places that could be concerned by significant impacts on the energy balance of the roof are places located in low or mid latitude because of a higher sunlight. The potential bias would occur in the early morning or the late evening when the Sun is low on the horizon to allow the shadow cast to reach a significant part of the roof. However, the solar radiation is far lower energetic than around the zenith when the shadow cast can't extend to the roof. So, the potential bias would be very small.

In addition, this process is by far marginal and supposed to have a limited impact on the microclimate within the canyon. Indeed, at 2 meters height, the impact of roof shading on air moisture and temperature could be considered as negligible. Remind that the purpose of this paper is to reduce the uncertainties of the prediction of the impact of urban planing (including greening) on thermal comfort of pedestrians.

Regarding these widespread practices and physical elements of discussion, the implementation of the process of shading of trees on roofs was not a priority for our team but remains technically a feasible project.

We added further explanations in the paper page 6-7, at lines 30-32, 1-8.

Bibliography: "Cahier de l'espace public, Chapitre II: Les arbres d'alignement", Mairie de Toulouse, 2008 (in French) "Street tree Management in Barcelona", Barcelona City Council, 2011 (in French) "Streets for all: A guide to the management of London's streets", Historic England, 2000 "Buildings height in the Royal Borough", Royal Borough of Kensington and Chelsea, "British Standard 5837: Trees in relation to construction — Recommendations", British Standards Institution, 2005 "Trees and the Public Realm (draft version)", City of Westminster, 2009

5. " Page 16, Lines 9-10. The use of the words 'good' and 'only' are very subjective. You need to state what is it good relative to? "

We agreed with the comment of the Referee and reformulated the interpretation of the results.

See in section 6.3 the revisions in the paper.

6. " Section 6.4. This section on sensitivity to vegetation layout characteristics and associated figures is not particularly clear. Statements are made without the use of statistical values nor clear comparison using examples from the relevant figures (Figs 8– 10). Page 16 Line 19 states 'The comparison of statistical scores', these statistics need to be presented within the text or in a table. "

Since we have not highlighted any significant and systematic patterns by studying the effect of tree horizontal coverage, tree canopy height, and tree location, we propose to remove this section which does not bring additional interesting information (as well as associated figures 8, 9, 10).

7. " Section 6.4. Figures 8, 9 and 10 do not show any particular clear patterns (there is a lot of scatter and points on top of each other) that allow the reader to determine the full impact of differences in vegetation or whether it was significant, an alternative method for showing this data is required. It is unclear of the utility of comparing two types of error (RMSE and %Err). Could you clarify why you have done this and how this shows how sensitive the model is to these changes? An explanation of this is required in the text. "

See response to comment 6.

8. " Section 6.5, Figure 11. There are four aspect ratios presented in figure 11 what is the difference between h/w = 2 and h/w = 2b? The second is not referred to in the text nor the figure caption. "

We acknowledge that our paper did not provide sufficient explanations about these additional cases, referred as "h/w=2b" in the former version and referred now (in the updated version) as "h/w=2 rescaled vegetation". These cases show identical urban morphology than classical h/w=2 cases but vegetation layer is doubly thicker and higher than h/w=0.5 and h/w=1 cases for each vegetation configurations in order to rescale it for higher buildings and verify the effect of adapted vegetation layouts regarding to the typology of the street (Cf Fig. 02 and section 5.2.1).

We hope that the readers will find clearer presentation of h/w=2b cases in page 12, at lines 20-24.

9. " Appendix A. Why is the view factor between the road and the tree = 0? Surely this would have an implication when calculating the canyon longwave radiation balance as

the road will see the underside of the tree layer. "

In the updated version of the paper, the view factors are based on the equations of sky view factors in Masson, 2000. Those related to the high vegetation are defined on all the directions. In this way, the interactions between all the facets of the canyon, the sky and trees are allowed in Eq. A6, A7, A8. It ensure the compatibility with the calculations of the longwave exchanges, for which the same view factors but no coefficients are involved. In addition, specific coefficients are applied, in the shortwave scheme only, to constrain the reflections from the high vegetation toward sky and top part of walls. In nature, the solar radiation is mainly redirected upwards by the receiving face of sunlit leaves in the top of crown during the first reflection. We suggest here to neglect the small amount of shortwave radiation which the tree stratum is supposed to reflect to the low part of the canyon during multiple reflections in favor of representing realistically the upward first reflection of solar radiation, which is by far the most energetic reflection.

See page 10, at lines 26-28 and page 11 lines 1-5 the revisions in the paper.

Part 2 : Grammatical and Format Errors

1. " TITLE. The authors may wish to amend the title to read more clearly as 'Implementation of street trees within the solar radiative exchange parameterization of TEB in SURFEX v8.0'. "

We changed the title in accordance with the suggestion of the Referee.

2. " ABSTRACT, Line 1. With the first use of an abbreviation the term/phrase/name it is used to abbreviate should be stated e.g. Town Energy Balance (TEB) or TEB (Town Energy Balance). The same applies to SOLENE. "

We changed the fist mention of TEB to "Town Energy Balance (TEB)" in the abstract page 1 line 1 and in the text, page 2 line 22. According to the team who created SOLENE, the name has no official meaning and it is not an acronym. Nevertheless, they inform me of a defined way of typing the name with capital letters and using a

smaller font size for the 'OLENE' part, consistently with previous papers.

3. " ABSTRACT, Line 3. The word 'obviously' is not required here as it is not obvious without reading the paper that there is increased complexity. It would be more appropriate to state this as a fact by removing the words 'has obviously'. "

We removed the expression 'has obviously' in accordance with the suggestion of the Referee (page 1 line 3).

4. " Page 2, Line 34. 'Surimpose' should read 'superimpose'. "

We corrected this typing error (page 2 line 35). According to the suggestion of the second Referee, we replace 'superimpose' by 'shade' in all the text in order to avoid confusion with contacting surfaces (see comment n°4, part 2 in the section of the second Referee comments).

5. " Page 5, Line 13. The work 'especially' is not required in this sentence. "

We removed this word.

6. " Page 6, Line 32. Should the word 'refined' be 'defined'? "

We corrected this typing error (page 7 line 10).

7. " Page 7, Lines 7 -10. This paragraph is not clear. Should it read as follows? "In order to calculate these terms in TEB, the following section describes how direct solar radiation reaches canyon surfaces. Then, absorption is obtained by separately resolving the first absorption of total solar radiation on each surface and the sum of absorbed shortwave radiation after infinite reflections within the canyon'. "

Thank you for reformulating correctly this paragraph. We changed the paragraph in accordance with the suggestion of the Referee (page 7 line 18-21) and added some details as following : "In order to calculate these terms in TEB, the 4.1 and 4.2 sections describe how direct and diffuse solar radiations reach canyon surfaces. Then, absorption is obtained by separately resolving the first absorption of total solar radiation

on each surface and the sum of absorbed shortwave radiation after infinite reflections within the canyon.".

8. " Page 11, Line 21. Should the first word be TEB on this line? "

We corrected this typing error.

9. " Page 12, Line 15. This first sentence doesn't read well. Consider splitting into two sentences, one explaining that TEB was run with equivalent configurations and another stating the differences between models. "

Thank you for this suggestion. We simplified the sentence (page 13, line 7-8) as following : "In the same way, the TEB model is run for equivalent configurations to SOLENE configurations, respecting hypotheses, approaches, and spatial resolutions differences between the two models.".

10. " Page 13, Line 15. Use of the word 'The' is redundant in this sentence. Start with 'Table 3 presents'. "

We changed the sentence (page 14, line 21) in accordance with the suggestion of the Referee.

11. " Page 13, Line 20. 'Let's remind' is not the correct style nor grammatically correct (Let's = Let is. If using this use 'Lets'). You may consider the following change to the sentence. 'Considering that the temperate climate is characterized by four distinct seasons with contrasting sunshine, air temperature and humidity conditions, seasonal analysis was undertaken'. "

We changed the paragraph (page 14, line 28-29) in accordance with the suggestion of the Referee.

12. " Page 14, Line 27. Incorrect style and grammar using 'let's remind'. The word 'considering' would be more suitable. "

We corrected the sentence (page 16, line 1) in accordance with the suggestion of the

Referee.

13. " Page 14, Lines 27-28. This sentence doesn't make sense. "

Thank you for pointing out this misuse of "Inversely". We reformulated the paragraph section 6.2.

14. " Page 16, Line 5. I am not sure what you mean by 'inversely slightly underestimated'? "

Thank you for pointing out this misuse of "inversely". We reformulated the paragraph section 6.3.

15. " Page 16, Line 20. Misspelt 'exchanges' "

We corrected this typing error (but the section has been removed).

16. " Page 17, Line 9. A missing paper reference at the end of this line. "

We corrected the .bib file.

17. " Page 17, Lines 9-10. Acronyms for models without full model names (as in point 2). "

We added the full names of the cited models page 19, lines 17-18.

18. " Page 20, Line 2. Missing equation reference. "

We corrected the .tex file.

19. " Page 22, Line 1. Misspelt 'example'. "

We corrected this typing error.

20. " Figures 4 and 6. The shades of blue used to represent 'wall A', 'wall B' and 'walls' are not clear and will not reproduce well if printed in black and white. Consider changing colours or using different line thickness. The subplots are also too small, consider reducing white space between subplots. This coul be achieved by limiting the

number of axis labels especially as you are using the same scale and variable on each row. "

Thank you for these suggestions. We changed the colors and untitled the axes of figures 4 and 4 to allow the reduction of white space between subplots in accordance with the suggestion of the Referee. We hope that description and cited units in captions are enough to understand the figures.

REFEREE 2Âă:

PART 1 : Major comments

1. " English. Ideally, the English should be edited by a native English speaker. "

After the revision process and following proofreading we hope that the English is more understandable.

2. " Error in application of Beer's law. Equations 3-5 appear to be mixed up. Eq. 4 with albedo removed should be the transmission; Eq. 3 as written is the intercepted radiation (multiply it by scattering coefficient to equal scattered radiation); Eq. 5 should be modified accordingly. Please check your model implementation carefully to ensure that it is correct according to the updated equations, and redo the simulations if the model equations were incorrect. "

Thank you for pointing out these errors. We corrected the equations 3, 4 and 5, page 8, in the paper. The coded equations were already correct but we run new simulations with the different way to code the option of upwards reflections from the high vegetation using classical view factors and specific coefficients in the shortwave radiation scheme (see response 3 and 9, part 1 in the response of the Referee 1, section 4.4 and Appendix A for further explanations). In addition, some adaptations about the calculation of the attenuation in TEB model have been done for the comparison exercise (see section 5.2.2.). SOLENE applies for each vegetation envelope mesh the same 'full' attenuation of 0,5 (assuming that it replace the exponential expression above including

Interactive
comment

the Leaf Area Index). In this way, at each attenuation process, the vegetated meshes in SOLENE attenuates 50% and transmits 50% of the received radiation. Where the Leaf Area Density is involved in the TEB equations (see Eq. B1, B2, B3, B4), we replaced the exponential term by the expression $1 – 0,5(LAD/LAI)$ to express the transmissivity by modulating the theoretical maximum attenuation of 0,5. Considering the LAD likely to be crossed by rays, we can obtain a transmissivity greater than 0,5 with TEB model. On the opposite, SOLENE can not take into account the layer of leaves where the rays are going through and always apply a full attenuation. The hypotheses of the two models are divergent and results have to be interpreted in the light of this divergence. The authors have amended the text in accordance with the suggestion of the Referees and adapted it to the results of the new simulations.

3. " Neglect of forward scattering. From Eqs. 3-6, it appears that forward scattering of intercepted radiation by vegetation is neglected (see Campbell and Norman 1989 for more detail; also, consider revising lines 7-8 on p. 10). Perhaps the albedo and extinction of the foliage are/can be adjusted to account for this. Either way, please explain and/or justify more fully. Can you assume all scattered radiation is scattered upward without introducing significant error? The broadband scattering coefficient is on the order of 0.50 for the leaves of many trees, and approximately 50% of this is forward scattered – so forward scattering potentially represents 25% of the shortwave radiative energy (very approximately). Interception by lower leaves of radiation forward scattered by upper leaves increases total absorption by tree foliage, which may correct the underestimation you find. "

Thank you for this very interesting comment. We acknowledge that our paper did not provide sufficient explanations about how TEB treats the forward scattering.

As shown in Eq. 2, the direct solar radiation flux potentially received by the high vegetation can be divided into three components: the transmitted part, the reflected part and the absorbed part. In the code, this potential flux reaching the high vegetation is calculated following the Eq. 1 (see page 7, lines 26-28).

[Figure]

For backward scattered (Eq. 4) and absorbed (Eq. 5) fluxes by high vegetation we use the potential received flux (Eq. 1) corrected by the transmitted part (using x (1 - exp (-k LAI) )) in the classical equation or x 0.5 in the comparison cases) in order to obtain the part of the incident flux which theoretically reaches only the leaves in the fraction of high vegetation (in other words, holes within the crown have been removed at this stage). In this way, TEB treats the forward scattering of the incident direct solar radiation flux by the high vegetation. Eq. 10 shows that the diffuse solar radiation flux reaching the high vegetation is also corrected by the transmitted part (page 9, lines 17-18). Indeed, we consider that the available flux, deduced subtracting the fluxes received by the other facets of the canyon (using view factors) from the incident diffuse solar radiation, reaches the high vegetation.

The forward scattering computed at infinite reflections between surfaces of the canyon in shortwave radiation is expressed by the tau terms (Eq. B1, B2, B3, B4). These supplementary elements demonstrate that the forward scattering is not neglected during all the processes occurring in the shortwave radiation scheme.

Concerning the backward scattering from the high vegetation, we suggest here to neglect the small amount of shortwave radiation which the tree stratum is supposed to reflect to the low part of the canyon during multiple reflections in favor of representing realistically the upward first reflection of solar radiation, which is by far the most energetic reflection (see Appendix C).

View factors related to high vegetation are based on the height of the middle of the crown (see Appendix A). In this way, the interception by leaves is assumed to be maximum at this level, so the forward scattering within the crown (on the tree fraction) is allowed to this limit. Explicit interactions from leaves to leaves inside the tree crown are not computed in the current version of TEB.

4. " Robustness of SOLENE as evaluation tool. How accurate are the SOLENE calculations of solar absorption and scattering by trees? p. 11, lines 7-8: Treatment of

foliage in the SOLENE should be further discussed, as well as any associated evaluation, given that this treatment is what TEB is being compared to. Details are sparse in Robitu et al. (2006). Does it include forward scattering, multiple reflection between tree foliage and the urban canyon underneath, or between different tree foliage elements, for example? Differences in model geometries between TEB and SOLENE are discussed; how different are the physics? "

SOLENE model can simulate the radiative budget of a urban scene represented as a numerical mock-up with explicit vegetation layouts where each mesh is associated to a view factor. Sections 5.2.1 and 5.2.2 have been complemented.

The forward scattering is represented by roughly transmitting 50% of the 'incident' (reaching) flux on the vegetation envelope at the first contact and not a second time when leaving the same envelope by side or toward the ground. Thus, the trees envelopes are considered as semi-transparent.

Multiple reflections are calculated by the radiosity method between all surfaces including tree foliage and the urban canyon underneath or between different lines of crowns. A crucial difference between TEB and SOLENE is the potential interaction between line of crowns. In TEB, we compute a cumulative fraction of high vegetation. In this manner, interactions between tree lines are not taken into account. On the contrary, in SOLENE, when a ray is intercepted a first time by a vegetation envelope and passed through to reach a second envelope, the ray is attenuated a second time (for example for the vegetation layouts (B3, C3, C4). In this way, an horizontal ray crossing twice a vegetated envelope is attenuated in order to release only 25 % of the initial flux when the rays are leaving the second envelope. Additionally, unlike in SOLENE, the attenuation can be modulated in TEB, respecting the surface-surface interaction considered using the Leaf Area Density (see response 2, part 1).

Finally, the multiple reflections are treated in different ways. In TEB the reflections of shortwave radiation are computed in an isotropic way (see Appendix C) while they

are represented as specular reflections in the SOLENE model, using the method of radiosity.

These divergences on the modulation of the attenuation and on the nature of the reflected radiation could partly explain the differences between the received and absorbed solar radiation fluxes of the high vegetation from the two models, apart from the geometrical assumptions.

5. " Large overall albedo difference. The large differences in mean albedo for higher H/W canyons in Fig. 11 are worrying, and suggests to this reviewer than one or more assumptions made in the formulation in the TEB model are inadequate (assuming the SOLENE model is robust – to point 4 above). It could relate to one of points 2 or 3 above. My sense is that tree foliage limited to the canyon should become less important as H/W increases – if that is true, why would performance degrade? Could this be an issue with the TEB shortwave radiation scheme without trees? A primary purpose of TEB is to provide neighbourhood-scale fluxes to the overlying atmospheric model, and overall albedo is the parameter to which the energy balance is often most sensitive. Hence, this result requires more investigation. "

Thank you for this very interesting comment. As explained in the previous response, the calculations of the multiple reflections within the canyon between SOLENE and TEB model are too divergent to allow an accurate comparison of absorbed fluxes. For this reason, we changed the figure presenting the mean daily canyon albedo (Fig. 9) in order to represent the comparison of canyon albedos between the simulations of the reference and current TEB version. The results are in line with our expectations, as discussed in the section 6.4.2.

Part 2 : Minor comments

1." p. 1, line 8: "...uncertainties in terms of the solar radiative exchanges, as quantified by comparison of TEB..." "

Thank you for this better reformulation. We corrected the text (page 1, line 8) in accordance with the suggestion of the Referee.

2. " p.1, line 18: remove "soil artificialisation due to" "

Thank you for this better reformulation. We corrected the text in accordance with the suggestion of the Referee.

3. " p.1, line 22: refresh, or cool, clean/filter, etc? "

Thank you for pointing out this error. We corrected the text (page 1, line 24) with a more suitable term ('these processes cool').

4. " p. 2: "surimpose" is not a word I don't think – do you mean "shade"? And again, p. 6, line 18 I think "shade" or a similar word (obscure?) might be better than "superimpose" (please check throughout). "

Thank you for pointing out this typing error (page 2, line 35). We have chosen the term 'superimpose' in the previous version to express the fact that the high vegetation is located above the ground-based surfaces (road and low vegetation). Indeed, this term could be confusing because the low and high vegetation are not in contact. We amended all the text in accordance with the suggestion of the Referee, using the 'shade' term.

5. " p. 3, line 2: "...by a microscale radiation model: SOLENE..." "

Thank you for this better reformulation. We corrected the text (page 3, line 2) in accordance with the suggestion of the Referee.

6. " p. 3, line 12: Krayenhoff et al (2014) "

Thank you for pointing out this error. We corrected the reference (page 3, line 13) in accordance with the suggestion of the Referee.

7. " p. 4, line 4: "...radiative (Krayenhoff et al. 2014) and dynamic (Krayenhoff et al.

2015) effects..."; the reference for BEP-Tree is "Krayenhoff (2015)" at present. "

Thank you for pointing out this error. We corrected the references (page 4, line 6) in accordance with the suggestion of the Referee.

8. " p. 4, line 6: "...both within and above the canyon and above roofs." "

Thank you for these supplementary details. We corrected the text (page 4, line 8-9) in accordance with the suggestion of the Referee.

9. " p. 4, lines 14-19: There are some significant assumptions in the view factor calculations and radiation exchange in ENVI-met that could be discussed. However, since it is a microscale model, it may not be relevant to go into much detail. "

Thank you for this very interesting comment. We are agree with the comment of the Referee and shortened the description of the ENVI-met model (page 4, lines 15-18).

10. " p. 4, lines 28-29: I suggest beginning the paragraph as follows: "At each mesoscale model grid point, TEB describes the average characteristics of the local environment by a single urban canyon..." "

Thank you for this better reformulation. We corrected the text (page 4, line 25) in accordance with the suggestion of the Referee.

11. " p. 8, line 6: This assumes tree foliage is uniformly distributed across the canyon, if I understand correctly? If so, this is worth stating in the text. "

Thank you for this very interesting comment. We acknowledge that our paper did not provide sufficient explanations about how the processes of interception and transmission of the high vegetation are computed respecting the cover fraction approach. For example, Eq. 1 (page 7 lines 29-30) calculates the direct solar radiation flux potentially reaching the high vegetation, considering the height of trees only. So, it express the flux available at this height, corrected by the shading effects of the walls. In further calculations, we associate the fluxes related to the high vegetation (as the reflected

and the absorbed ones) to the high vegetation fraction (Eq. 6, Eq. 7). That means that the tree foliage is assumed uniformly distributed across the canyon in Eq. 1 (page 8, lines 20-21), as highlighted by the Referee.

12. " p. 11, lines 3-6: Is this relevant, if TEB assumes an isotropic distribution? Presumably the same is chosen in SOLENE? "

Thank you for this very interesting comment. At lines 21-26, page 13, we precise that TEB model is forced with the same conditions of incoming solar radiation than those calculated for the roofs in the SOLENE model. We add a missing element at lines 4-5, page 13: SOLENE has been parameterized to generate perfectly clear cloudless skies. Using a unique forcing for each component (direct or diffuse) of the solar radiation from the SOLENE simulations, TEB forcings do not take into account the non-uniform distribution of incoming diffuse solar radiation on walls. From this imprecision result differences from 1% to 4% between fluxes of the two walls, depending on their orientation, for studied aspect ratios during summertime. Note that an option also allows to represent a non-uniform distribution of the incoming diffuse solar radiation in TEB model but it was deactivated for this experiment.

13. " Eq. 19: RMSD (difference, not error, since you are comparing two models); also note the critiques of RMSD relative to MAE (Mean Absolute Error), e.g. Willmott et al. 2009. I suggest calculation of MAE instead. "

Thank you for this excellent suggestion. We have noted the advantages of using the Mean Absolute Error to quantify the mean error instead of the Root Mean Square Error in our study. Consequently, we present the results of MAE in the updated version of our paper, in accordance with the suggestion of the Referee.

14. " p. 17, line 9: Reference missing. "

Thank you very much for pointing out this error. We corrected the .bib file.

15. " p. 26, line 27: Year is 2015, not 2014. "

Thank you very much for pointing out this error. We corrected the .tex file.

16. " p. 26, line 29: Year is 2014, not 2013. "

Thank you very much for pointing out this error. We corrected the .tex file.

17. " Appendix A, line 1: Eq. ??; and again line 16 on p. 21. "

Thank you very much for pointing out this error (page 22, line 2). We corrected the .tex file.

18. " Appendix A, line 19: View factor from road to trees is zero??? Please explain! "

In the updated version of the paper, the view factors are based on the equations of sky view factors in Masson, 2000. Those related to the high vegetation are defined on all the directions. In this way, the interactions between all the facets of the canyon, the sky and trees are allowed in Eq. A6, A7, A8. It ensure the compatibility with the calculations of the longwave exchanges, for which the same view factors but no coefficients are involved. In addition, specific coefficients are applied, in the shortwave scheme only, to constrain the reflections from the high vegetation toward sky and top part of walls. In nature, the solar radiation is mainly redirected upwards by the receiving face of sunlit leaves in the top of crown during the first reflection. We suggest here to neglect the small amount of shortwave radiation which the tree stratum is supposed to reflect to the low part of the canyon during multiple reflections in favor of representing realistically the upward first reflection of solar radiation, which is by far the most energetic reflection.

See section 4.4 the revisions in the paper.

19. " Appendix A, line 8: LAD is in m2 of leaf area per m3 of volume. "

Thank you very much for pointing out this error. We corrected the text page 23, line 8.

Please also note the supplement to this comment:
http://www.geosci-model-dev-discuss.net/gmd-2016-157/gmd-2016-157-AC1-

supplement.pdf

[Figure]

**Supplement:**

**Implementation of street trees within the solar radiative exchange parameterization of TEB in SURFEX v8.0**

Emilie Redon[a], Aude Lemonsu[a], Valéry Masson[a], Benjamin Morille[b], and Marjorie Musy[b]

[a]CNRM UMR 3589, Météo-France/CNRS, Toulouse, France, 42 avenue Gaspard Coriolis, 31057 Toulouse cedex 1, France
[b]Ecole nationale supérieure d'Architecture de Nantes, UMR 1563, quai François Mitterrand, 44262 Nantes cedex 2, France
*Correspondence to:* A. Lemonsu (aude.lemonsu@meteo.fr)

**Abstract.** The Town Energy Balance (TEB) model has been refined and improved in order to explicitly represent street trees and their impacts on radiative transfers: a new vegetated stratum on the vertical plane, which can shade the road, the walls and the low vegetation has been added. This modification led to more complex radiative calculations, but has been done with a concern to preserve a certain level of simplicity and to limit the number of new input parameters for TEB to the cover fraction of trees, the mean height of trunks and trees, their specific Leaf Area Index and albedo. Indeed, the model is designed to be run over whole cities, for which it can simulate the local climatic variability related to urban landscape heterogeneity at the neighborhood scale. This means that computing times must be acceptable, and that input urban data must be available or easy to define. This simplified characterization of high vegetation necessarily induces some uncertainties in terms of the solar radiative exchanges, as quantified by comparison of TEB with a high spatial-resolution solar enlightenment model (SOLENE). On the basis of an idealized geometry of urban canyon with various vegetation layouts, TEB is evaluated regarding the direct and diffuse solar radiation received by the elements that compose the canyon. TEB simulations in summer gathered best scores for all configurations and surfaces considered, which is precisely the most relevant season to assess the cooling effect of deciduous trees under temperate climate. Biases less than $\pm 1$ W m$^{-2}$ related to the direct and diffuse solar radiation fluxes received by road and vertical surfaces have been recorded for vegetationless canyons. Concerning the vegetated canyons we noted a high variability of statistical scores depending on the vegetation layout. The greater uncertainties are found for the solar radiation fluxes received by the high vegetation. The Mean Absolute Error averaged over the vegetation configurations during summertime is $17.27 \pm 9.31$ W m$^{-2}$ for the direct component and $3.45 \pm 1.93$ W m$^{-2}$ for the diffuse component but these scores are associated to acceptable biases: $-14.44 \pm 10.60$ W m$^{-2}$ and $-3.21 \pm 2.20$ W m$^{-2}$, respectively.

[revised manuscript text omitted]

More specifically for the shortwave radiation budget, three contributions are considered for a given element:

1. The direct solar radiation received before any reflections. This contribution depends on zenith angle since the incident direct radiation is unidirectional, street orientation, and canyon aspect ratio.

2. The diffuse solar radiation received before any reflections. This contribution depends on the sky view factor of the considered element since the diffuse radiation is assumed to be isotropic.

3. The total shortwave radiation received after multiple reflections within the canyon. After a first reflection on one of the elements of the canyon, initial contributions of direct and diffuse radiation are isotropic and are treated the same way. The part of radiation received by a given element then depends on the view factors of all the other elements and on their albedo that determine the reflected radiation part.

Although this paper focuses on resolution and evaluation of the shortwave radiation budget, it is worth to note that the validation of our shortwave radiation scheme contributes to verify our future longwave radiation scheme. Indeed, the same view factors used for the multiple reflections will be applied to the longwave radiation interactions within the canyon. The longwave exchanges are computed following the typical Stefan-Boltzmann law. Moreover, because of high longwave emissivities, only one re-emission is computed. For numerical stability purpose, an implicit formulation is applied for longwave radiation budgets; it includes the surface temperatures at the previous numerical time step and at the current time step.

**3.3 Inclusion of a high vegetation stratum for solar radiation calculation**

To take into account the tree canopy in TEB, it is required to add a new vegetated stratum on the vertical plane, which can shade the road, the walls and the low vegetation. This modification led to more complex radiative calculations, but is done with a concern to preserve a certain level of simplicity and to limit the number of new input parameters for TEB. This is motivated by the type of applications which are conducted with the TEB model, and more generally with the SURFEX land surface modeling platform (Masson et al., 2013) which TEB is part of. The system is frequently applied over domains of several hundred square kilometers with horizontal resolutions between few hundred meters to few kilometers and can be run for long time periods up to several years in case of climatic studies (e.g., Lemonsu et al., 2013; de Munck, 2013, in French). This means that computing times must be acceptable, and that urban input data must be available or easy to define.

The arrangement of tree canopy is here described using three parameters only (Fig. 1 and Table 1): its cover fraction ($\delta_t$), i.e. the proportion of canyon which is covered by the foliage stratum on the horizontal plane, as well as the mean height of trees ($h_t$), and the mean height of trunks ($h_{tk}$). In the current version of TEB (official SURFEX v8.0 version), urban trees are assumed to be less tall than surrounding buildings and systematically confined inside the canyon so that they cannot provide shade for roofs. This hypothesis is in accordance with urban planning requirements specifications for street trees management

(in French, Municipality of Toulouse, 2008; City of Westminster, 2009; Barcelona City Council, 2011). These documents ensure a satisfactory juxtaposition of trees with urban structures for dwellers. Minimum distances between trunks or crowns and walls or balconies are strictly imposed to avoid problems such as excessive obstruction of crowns facing windows, disruption of underground services by roots or subsidence of buildings. These widespread practices are also applied in private gardens or
5  suburban areas and tends to avoid shadow on roofs even if street trees can be taller than buildings. Design with trees shading on roofs are statistically sparse and their impact on surface balance limited. They are probably located in high latitudes where incoming fluxes are lesser than in mid or low latitudes and this potential bias would only occur in the early morning or late evening, when the zenith angle is large but the solar radiation flux far lower energetic than around noon.

10  For now, the shape of the foliage and the vertical distribution of leaves are not defined. The crowns of trees are considered as parallelepipeds (namely computed as a rectangular 2D cross-section) with homogeneous foliage which is described by a Leaf Area Index ($LAI_t$) and an albedo ($\alpha_t$). It is however possible to vary the LAI during the year, in order to simulate the seasonal cycle of deciduous trees. Note that trunks are not taken into account in radiative calculations. The tree vegetation stratum is considered as a partially-transparent element for shortwave radiation. A part of the incident radiation received by
15  trees is transmitted through the foliage. The part of radiation which is not transmitted is consequently reflected or absorbed, depending on albedo. These processes and the associated calculations are detailed hereafter.

**4  Solar radiation absorption of vegetated street canyon surfaces**

In this part, equations related to the implementation of a tree layer into the TEB model are presented. In order to calculate these terms in TEB, the 4.1 and 4.2 following sections describe how direct and diffuse solar radiations reach canyon surfaces. Then,
20  absorption is obtained by separately resolving the first absorption of total shortwave radiation on each surface and the sum of absorbed shortwave radiation after infinite reflections within the canyon.

**4.1  Direct solar radiation received by each element**

The foliage of trees plays a role of obstruction and attenuation of incident direct solar radiation ($S^{\Downarrow}$) for the other elements of the urban canyon. Consequently, to determine the direct solar radiation received by each element of the canyon, we need to
25  solve first the equations related to high vegetation.

The direct solar radiation potentially reaching the top of trees by geometrically taking into account the shading of buildings depends on buildings height ($h$), canyon aspect ratio ($h/w$), street orientation ($\theta_{can}$), zenith and azimuth angles ($\lambda$, $\theta_{sun}$), as well as tree height ($h_t$):

30  $$S_t^{\Downarrow} = S^{\Downarrow} \, max \left[ 0 \, ; \, 1 - \frac{h}{w} \left( \frac{h - h_t}{h} \right) \, tan(\lambda) \, sin \, |\theta_{sun} - \theta_{can}| \right] \qquad (1)$$

As explained previously, this radiation flux is partially transmitted through the foliage ($S_t^{\gg}$), whereas the remaining solar radiation is reflected ($S_t^{\Uparrow}$) or absorbed ($S_t^{*}$):

$$S_t^{\Downarrow} = S_t^{\gg} + S_t^{\Uparrow} + S_t^{*} \tag{2}$$

The proportion of direct solar radiation transmitted through the foliage is estimated by a Beer Lambert law (Campbell and Norman, 1989) where the Leaf Area Index ($LAI_t$ expressed in $m^2$ of leaves per $m^2$ of ground) of tree canopy and an extinction coefficient ($k$) are involved. The extinction coefficient is fixed to 0.5, a default value corresponding to homogeneous repartition of leaves in terms of density and orientation (in other words, a spherical leaf angle distribution):

$$S_t^{\gg} = S_t^{\Downarrow} \exp^{(-k\,LAI_t)} \tag{3}$$

The reflected radiation part simply depends on the part of incident solar radiation untransmitted through the foliage and on the albedo of trees ($\alpha_t$):

$$S_t^{\Uparrow} = \alpha_t\, S_t^{\Downarrow} \left(1 - \exp^{(-k\,LAI_t)}\right) \tag{4}$$

Finally, the incident direct solar radiation part absorbed by trees is neither transmitted nor reflected and calculated as the residual term from Eq. 2:

$$S_t^{*} = (1 - \alpha_t)\, S_t^{\Downarrow} \left(1 - \exp^{(-k\,LAI_t)}\right) \tag{5}$$

The direct solar radiation received by the ground (indiscriminately road or garden fraction) is deduced by correcting the incident solar radiation above canyon from the interception of radiation by high vegetation canopy (i.e., reflected and absorbed radiation weighted by high vegetation cover fraction referred to as $\delta_t$), and then from the shading effects of buildings (according to Lemonsu et al., 2012). The same equations are obtained for road ($S_r^{\Downarrow}$) and garden ($S_g^{\Downarrow}$):

$$S_r^{\Downarrow} = \left(S^{\Downarrow} - \delta_t \left(S_t^{\Uparrow} + S_t^{*}\right)\right) max\left[0\,;\, 1 - \frac{h}{w}\,tan(\lambda)\,sin\,|\theta_{sun} - \theta_{can}|\right] \tag{6}$$

In this way, tree foliage is assumed to be uniformly distributed accross the canyon at the height of the trees ($h_t$), consistently with the Eq. 1.

The direct solar radiation which is not received either by high vegetation, or by road or garden is assigned to the sunlighted wall, whereas the opposite wall is in the shadow. By convention in TEB in the case of an oriented canyon, we define $wall\,A$ as the most sunlit wall and $wall\,B$ as the shaded one.

$$S_{w_A}^{\Downarrow} = \left(S^{\Downarrow} - S_r^{\Downarrow} - \delta_t \left(S_t^{\Uparrow} + S_t^{*}\right)\right) \frac{w}{h} \qquad\qquad S_{w_B}^{\Downarrow} = 0 \tag{7}$$

Note that shading effects of high vegetation on roofs are not represented, since urban trees are less tall than buildings by definition in the current version of TEB (SURFEX v8.0).

**4.2 Diffuse solar radiation received by each element**

The incoming diffuse solar radiation ($S^\downarrow$) is assumed to emit isotropically. Each urban surface of the canyon ($wall$, $road$, and $garden$) receives a part of diffuse solar radiation according to the sky view factor of the surface $\Psi_\star$ (see Appendix A) and the mean radiative transmissivity between the sky and the given surface $\tau_{\star s}$ (see Appendix B). Note that the sky view factor of $wall$ is defined at mid-height of buildings; for ground-based surfaces $road$ and $garden$, a single sky view factor $\Psi_{rs} = \Psi_{gs}$ is defined at the center of the street (Masson, 2000; Lee and Park, 2008). The following equations are obtained for $road$ (same expression for $garden$) and for $wall$:

$$S_r^\downarrow = S^\downarrow \, \Psi_{rs} \, \tau_{rs} \tag{8}$$

$$S_w^\downarrow = S^\downarrow \, \Psi_{ws} \, \tau_{ws} \tag{9}$$

We admit that the residual flux of diffuse solar radiation which is not intercepted in the canyon by previous surfaces reaches the tree canopy:

$$S_t^\downarrow = \frac{S^\downarrow - \left(\delta_r \, S_r^\downarrow + \delta_g \, S_g^\downarrow + \frac{2h}{w} \, S_w^\downarrow\right)}{\delta_t} \tag{10}$$

This method presents two major advantages: (1) the diffuse solar radiation budget is always closed and (2) the computed diffuse solar radiation flux for the high vegetation is already corrected from the transmitted part, reaching the other surfaces.

The fluxes of each surface are here expressed according to the total ground-based surface of the canyon, with $\delta_r$ and $\delta_g$ the cover fractions of road and garden in the canyon ($\delta_r + \delta_g = 1$), respectively.

**4.3 First absorption of total shortwave radiation by each element**

The first absorption of total shortwave radiation $S^*(0)$, before any reflections, is only function of the total shortwave radiation received by the considered element and of its albedo ($\alpha_\star$). The same expression is obtained for walls and ground-based surfaces ($\star$ which stands for $r$, $g$, $w_A$ or $w_B$):

$$S_\star^*(0) = (1 - \alpha_\star) \left(S_\star^{\Downarrow} + S_\star^\downarrow\right) \tag{11}$$

For the tree canopy, the part of absorbed direct solar radiation is corrected by the transmitted flux:

$$S_t^*(0) = (1 - \alpha_t) \left[\left(S_t^{\Downarrow} - S_t^{\ggg}\right) + S_t^\downarrow\right] \tag{12}$$

$S_t^{\Downarrow}$ includes the transmitted flux $S_t^{\ggg}$ (see Eq. 2) contrary to $S_t^\downarrow$ which is calculated as a residual flux (Eq. 10), corrected from the transmitted flux.

**4.4 Sum of total shortwave radiation absorbed by each element**

Our goal is to compute the total shortwave radiation absorption for each element $S^*(\infty)$ by taking into account an infinite number of reflections between all elements composing the urban canyon. At each reflection, the isotropic radiation intercepted by a given element (1) after reflections on one of the other elements (2) is conditioned by the view factor of (2) from (1) referred to as $\Psi_{12}$ (see Appendix A), the mean radiative transmissivity $\tau_{12}$ (see Appendix B) and the absorption is then determined according to reflective properties of (1). Using a single view factor in TEB radiation calculations is obviously a limitation for accurately representing the various contributions of canyon's surfaces to high vegetation. Additionally, we have a poor knowledge of a potential contribution of leaves with each others, within the tree's crown. To ensure a closing system, we define the total absorbed shortwave radiation by high vegetation as the remaining shortwave radiation, after accounting for absorption and reflection from the total incident solar radiation by all other elements of the canyon. This requires to calculate the part of shortwave radiation which leaves the canyon towards the sky. The terms $R_\infty$, $G_\infty$, $A_\infty$, $B_\infty$, and $T_\infty$ make reference to the sum of total shortwave radiation reflected by each surface, respectively, after an infinite number of reflections (see detailed resolution in Appendix C). Here is the expression of the total absorbed solar flux per surface:

$$S_s^*(\infty) = \Psi_{sr}\tau_{sr}(\delta_r R_\infty + \delta_g G_\infty) + \Psi_{sw}\tau_{sw}\frac{A_\infty + B_\infty}{2} + \Psi_{st}\delta_t T_\infty \tag{13}$$

$$S_r^*(\infty) = S_r^*(0) + (1 - \alpha_r)\left[\Psi_{rw}\tau_{rw}\frac{A_\infty + B_\infty}{2} + c_{rt}\Psi_{rt}\delta_t T_\infty\right] \tag{14}$$

$$S_g^*(\infty) = S_g^*(0) + (1 - \alpha_g)\left[\Psi_{rw}\tau_{rw}\frac{A_\infty + B_\infty}{2} + c_{rt}\Psi_{rt}\delta_t T_\infty\right] \tag{15}$$

$$S_{w_A}^*(\infty) = S_{w_A}^*(0) + (1 - \alpha_w)\left[\Psi_{wr}\tau_{wr}(\delta_r R_\infty + \delta_g G_\infty) + \Psi_{ww}\tau_{ww}B_\infty + c_{wt}\Psi_{wt}\delta_t T_\infty\right] \tag{16}$$

$$S_{w_B}^*(\infty) = S_{w_B}^*(0) + (1 - \alpha_w)\left[\Psi_{wr}\tau_{wr}(\delta_r R_\infty + \delta_g G_\infty) + \Psi_{ww}\tau_{ww}A_\infty + c_{wt}\Psi_{wt}\delta_t T_\infty\right] \tag{17}$$

$$S_t^*(\infty) = \frac{1}{\delta_t}\left[\left(\left(S_t^\Downarrow - S_t^\gg\right) + S_t^\downarrow\right) - \left(S_s^*(\infty) + \delta_r S_r^*(\infty) + \delta_g S_g^*(\infty) + \frac{2h}{w}\frac{S_{w_A}^*(\infty) + S_{w_B}^*(\infty)}{2}\right)\right] \tag{18}$$

The view factors related to the high vegetation stratum are expressed in Appendix A. Specific coefficients are applied, in the shortwave scheme only, to constrain the reflections from the high vegetation toward sky and top part of walls. In nature, the solar radiation is mainly redirected upwards by the receiving face of sunlit leaves in the top of crown during the first reflection.

We suggest here to neglect the small amount of shortwave radiation which the tree stratum is supposed to reflect to the low part of the canyon during multiple reflections in favor of representing realistically the upward first reflection of solar radiation, which is by far the most energetic reflection. Solar reflections calculations are fully explained in Appendix C. As previously mentioned, view factors used for the multiple reflections in the shortwave radiation scheme will be applied to the longwave radiation interactions within the canyon in future works.

**5    Comparative exercise with the SOLENE model**

[revised manuscript text omitted]

For each of these urban canyons, 13 different vegetation layouts are prescribed (Fig.3), as well as a control case without vegetation. The vegetation blocks are parallelepipeds representing the three-dimensional tree crowns, without trunk. The vegetation blocks are continuous lengthwise the canyon. Depending on the configurations, the trees can be organized in single or double rows. According to a cut plan through the canyon, the tree crowns can fill 30, 60, or 90 % of canyon width. For the three aspect ratios, height of trees is prescribed to 5 m or 7.5 m with trunks of 2.5 or 5 m, so that the thickness of tree canopy varies between 2.5 and 5 m. Additional configurations (referred to as $h/w = 2$ rescaled vegetation) are tested for which height of trees is prescribed to 10 m or 15 m with trunks of 5 or 10 m, depending on the thickness and the location of the crown (Cf. Fig. 2). In other words, vegetation layer is doubly thicker and higher than $h/w = 0.5$ and $h/w = 1$ cases for each vegetation configuration in order to rescale it for higher buildings and verify the effect of adapted vegetation layouts regarding the typology of the street. For all experiments, the Leaf Area Index of trees is prescribed to 1, and the albedo is prescribed to 0.25 for road, 0.30 for walls, and 0.25 for trees. All configurations are described in Fig. 1, 2, 3 and Table 2.

To treat the ensemble of configurations, 55 digital mock-ups (52 canyons with vegetation and 3 canyons without vegetation) have been built with the computer-aided-design (CAD) Salome V7_4_0 software. All mock-ups have been meshed by the GMSH software which is a finite element mesh generator. We have applied here a non-uniform meshing with a characteristic length of only 1 m in order to refine the spatial discretization of vegetation blocks, whose smallest ones for some of the vegetation layouts do not exceed 2.4 m of width and 2.5 m of height.

Each canyon is projected following the four street orientations 0 ˚, 45 ˚, 90 ˚, and 135 ˚ (degrees from geographical north, in

the counter trigonometric direction). A specific location (defined by latitude and longitude) must be prescribed for astronomic calculations. The city of Nantes (France) is chosen in an arbitrarily manner (46 ° N, 1 ° E). The solar radiation exchanges are then calculated for a single daily cycle, and under sunlight conditions covering the four seasons by selecting dates close to equinoxes and solstices of year 2010, i.e. 20th March, 21st June, 23rd September, and 23rd December of 2010. Note also that
5    the Perez model (see section 5.1) has been parameterized in order to generate perfectly clear cloudless skies.

**5.2.2    Canyon modeling in TEB**

In the same way, TEB is run for equivalent configurations to SOLENE configurations, respecting hypotheses, approaches, and spatial resolutions differences between the two models. For TEB simulations, the geometrical parameters describing the urban canyon form, as well as the height of trees and trunks are comparable to those of SOLENE simulations. But the different spatial
10    arrangements of trees simulated by SOLENE are simply prescribed as cover fractions in TEB. As a result, some configurations (e.g. B1 and B2 with different horizontal locations or B3 and B4 with different number of tree lines, shown in Fig.3) cannot be distinguished by the TEB approach, and are associated to the same cumulative cover fraction of tree canopy in TEB. In this manner, interactions between tree lines are not taken into account in TEB, contrary to SOLENE where rays are attenuated at each time crossing a mesh belonging to a vegetated envelope. All geometric features of both sets of simulations with SOLENE and
15    TEB are summarized in Table 2. In SOLENE, the incident radiation flux is roughly attenuated of 50 % at once when crossing a mesh of a vegetation envelope. In terms of process modeling, the formulation of transmissivity of radiation through the foliage in TEB is here simplified for the evaluation stage. In order to be consistent with the SOLENE approach, the exponential attenuation expressing a maximum interception (including the Leaf Area Index) in Eq. 3, 4, 5 is replaced by 0.5. The same way, the formulation proposed in Appendix B (Eq. B1, B2, B3, B4) for modulating the radiation attenuation depending on the
20    likely path of rays and based on the Leaf Area Density profile, is here substituted by the expression $1 - 0.5(LAD/LAI)$, so that the maximum attenuation is 0.5 when all thickness of tree canopy is passed through. Note also that TEB is forced by the same conditions of incoming solar radiation than those calculated for the roofs in SOLENE. Using a unique forcing for each component (direct or diffuse) of the solar radiation from the SOLENE simulations, TEB forcings do not take into account the non-uniform distribution of incoming diffuse solar radiation. From this imprecision result differences from 1 to 4 % between
25    the fluxes of the two walls, depending on their orientation, for studied aspect ratios during summertime (sensitive analysis not shown).

**5.2.3    Comparison method**

Finally, 880 solar radiation simulations are performed with both TEB and SOLENE models. For each of them, hourly outputs are stored. They include the direct and diffuse solar radiation received by the separated elements (*road*, *walls*, and *tree*) before
30    multiple reflections, as well as the total shortwave radiation absorbed by the separate elements after multiple reflections. The main objective of the comparative exercise is to evaluate the cover fraction approach of TEB against a model (SOLENE) resolving the urban radiation budget at fine scale and with trees explicitly represented by geometrical elements. For this purpose, gaps between the simulations of received direct or diffuse solar radiation fluxes by canyon surfaces have been investigated.

Because of divergent physics assumptions, the statistics concerning the absorption by the facets of the canyon need a cautious interpretation. During multiple reflections, the radiation is assumed to be isotropic in TEB while SOLENE computes reflections with specular behavior using the method of radiosity. For comparing SOLENE and TEB, only the central part of the SOLENE's mock-up is used in order to avoid any boundary effects (see the scheme in Fig. 3). For each flux, the values calculated by SOLENE are summed over all grid points that compose each element of the canyon (separately, the road, the two walls, and the trees). Finally, for both models, the fluxes are weighted to be expressed according to the total ground-based surface of the canyon, so that they can be compared to each other and compared to the incoming radiation. Note that tables with statistical scores presented hereafter have been fulfilled with the same procedure, i.e., the Mean Absolute Error (MAE in W m$^{-2}$), the Mean Absolute Percentage Error (MAPE in %) computed from the mean daily fluxes, and the mean bias (Bias in W m$^{-2}$), according to following equations:

$$MAE = \frac{\sum_{i=1}^{n_s} \sum_{j=1}^{n_o} \sum_{k=1}^{n_h} \sum_{l=1}^{n_t} |F_{SOL}(i,j,k,l) - F_{TEB}(i,j,k,l)|}{n_s \cdot n_o \cdot n_h \cdot n_t} \tag{19}$$

$$MAPE = 100 \cdot \frac{\sum_{i=1}^{n_s} \sum_{j=1}^{n_o} \sum_{k=1}^{n_h} |\frac{\overline{F_{SOL}(i,j,k)} - \overline{F_{TEB}(i,j,k)}}{\overline{F_{SOL}(i,j,k)}}|}{n_s \cdot n_o \cdot n_h} \tag{20}$$

$$Bias = \frac{\sum_{i=1}^{n_s} \sum_{j=1}^{n_o} \sum_{k=1}^{n_h} \sum_{l=1}^{n_t} (F_{SOL}(i,j,k,l) - F_{TEB}(i,j,k,l))}{n_s \cdot n_o \cdot n_h \cdot n_t} \tag{21}$$

The indexes $i$, $j$, $k$, $l$ refer to as season, street orientation, aspect ratio of the street, and hour of the day, respectively, with $n_s = 4$, $n_o = 4$, $n_h = 3$, $n_t = 24$. We also run simulations of the $h/w = 2$ additional cases where canyons are greened by a rescaled vegetation (see section 5.2.1 and Fig. 2).

**6 Results**

**6.1 General analysis and seasonal effects**

Table 3 presents the statistical scores computed for the shortwave radiation absorbed by the different elements of the canyon for both cases, with and without vegetation. Since multiple reflections are treated differently between TEB and SOLENE, these results are used to show the magnitude of variability of errors related to the presence of vegetation in the canyon or the considered season. Results are presented here by gathering the experiments performed with different street orientations and different vegetation layouts, but by distinguishing the seasons. In the light of the mean biases, the TEB model tends to systematically overestimate the absorption by road and walls compared to SOLENE (up to +16 and +10 W $m^{-2}$ for road and walls, respectively), and to underestimate the absorption by trees (up to -19 W $m^{-2}$), whether the configuration or the season. Considering that the temperate climate is characterized by four distinct seasons with contrasting sunshine, air temperature and humidity conditions, seasonal analysis was undertaken. Analysis of the results for each season separately indicates that relative

errors (MAPE) are especially high for wintertime simulations for road and trees, due to a very low incoming solar radiation at that period. Nonetheless, the associated MAE and biases are acceptable (less than 3.5 and $\pm$ 3 W m$^{-2}$, respectively). Summer is the season which provides the best results in terms of MAPE when the canyon is tree-filled (less than 25 %) or not (3 %). This season is also the most relevant to be examined here because our first concern is to improve the simulation of the potential

[revised manuscript text omitted]

of the crown envelope (Fig. 7 a., all seasons and configurations). Consequently, the solar radiation which is not intercepted by the tree layer in TEB simulations is assigned to the road and the bottom of walls. This leads for configuration A to MAE of 9.18, 3.52 and 6.22 W m$^{-2}$ for road, walls, and trees, respectively, and MAPE of 14, 5, and 4 % only (Table 4). The thicker the crown is, the greater is the error associated to the tree (D1 vs B1 or D2 vs B2), particularly when the tree rows are away from the walls (B1 vs B2, D1 vs D2 or D3 vs D4), contrary to a continuous tree layer occupying almost the entire width of the canyon (A, C1, C2).

As expressed in Eq. 8 and 9, the diffuse solar radiation fluxes received by road and walls (walls are managed together as an average surface), depends on the sky view factor of the given surface and on an attenuation coefficient of the incoming radiation through the foliage. Regarding all seasons and configurations confounded, the received solar radiation by ground-based surfaces is also overestimated while the diffuse solar radiation flux reaching tree crowns is underestimated in the TEB simulations (Fig. 7 b.). Remind that the diffuse solar radiation received by the trees is calculated in TEB as the residual part of the incoming diffuse solar radiation flux which was not received by road and walls (see Eq. 10). Contrasted results are obtained for the vertical surfaces whether the vegetation layouts, in particular between double-row or centered trees (B3 vs B4). As previously discussed in section 5.4, these defects are related to the use of a single sky view factor for each surface, which is computed at mid-height of buildings for walls and in the middle of the street for road. In addition, the isotropic nature of the diffuse solar radiation leads to an exacerbation of the underestimation of the received flux by the high vegetation, particularly in cases where the side surface of vegetation envelopes is larger: for example, thick crowns with longer sides and vegetation layouts including two rows (the number of sides is doubled). Considering the extremely vegetated canyon A, MAPE are 15, 9 and 20 % for road, walls, and tree, respectively, corresponding to MAE of 1.89, 1.51 and 3.51 W m$^{-2}$ (Table 5). Globally, the estimation of the direct solar radiation received by the road and walls are acceptable regarding the MAPE which are $\leq 30\%$. However, the relative errors associated to the tree layer are contrasted (from 12 % to 49 %), depending on the characteristics of the vegetation layout.

For the same configuration, comparing TEB results to the SOLENE simulations as reference, the total shortwave radiation absorbed by the different elements of the canyon is simulated with correct daily dynamics (Fig. 6). Despite similar temporal behavior of the fluxes, their magnitudes can greatly diverge, especially for the road and tree surfaces with MAPE greater than 30 % and biases greater than $\pm 25$ W m$^{-2}$ (Table 6). The global scatterplot (Fig. 7 c., all seasons and configurations) confirms the trends previously found and the absorbed flux by the trees is underestimated. The predictability based on SOLENE results is also weaker ($R^2 = 0.945$). These poor performances of TEB regarding the statistical scores have to be interpreted with caution. This work aims at evaluating the cost in term of performances of TEB to simulate a correct allocation of radiative fluxes for each facet of the canyon in spite of a simple approach based on cover fraction. For example, the tree fraction is computed as a cumulative fraction of all crowns in the street in TEB. Thus, the interactions between tree lines are not allowed, contrary to in SOLENE. However, some physical differences limit the comparison of the absorbed fluxes. Indeed, the way to calculate the multiple reflections is dissimilar between the two models (see section 5.2.3). TEB code computes infinite isotropic reflections

using a unique view factor for each facet of the canyon and mean radiative transmissivity terms. Remind that the mean radiative transmissivity terms are based on strong hypotheses on the path of rays considering the studied interaction (see Apendix B). On the contrary, SOLENE radiative scheme computes multiple reflections as specular radiation within the canyon for each triangular mesh and using the radiosity method. The vegetation envelopes are strictly semi-transparent because the penetration of light through the foliage can not be modulated by the Leaf Area Density. Whatever the foliage thickness which is crossed, as soon as a ray reaches a cell of vegetation envelope in SOLENE, this radiation is attenuated by half.

Further works (not shown) have investigated the sensitivity of TEB results and performances according to the characteristics of the different vegetation layouts. They do not demonstrate clear and systematic patterns when studying the impact of: (1) tree horizontal coverage (or the tree fraction); (2) tree canopy height compared to the building height; (3) tree location - centered or on side - in the canyon on the MAE and error percentages recorded. It could be explained by the interaction between opposite effects regarding the vegetation layouts characteristics.

**6.4 Benefits of TEB developments**

**6.4.1 Analysis of the mean daily absorbed shortwave radiation by the canyon**

The shortwave radiation received and absorbed by the walls and the road can be strongly affected by the presence of tree vegetation. The comparison between the initial version of TEB which deals with vegetation at ground level, and the new version which explicitly includes an additional tree stratum, shows differences. Before weighting the fluxes by the appropriate cover fractions, the shortwave radiation absorbed by the ground-based surfaces are largely overestimated in the initial version of TEB (+ 77% for road and +66% for the garden in the case A in summer). These defects highlight the lack of representation of the interception of rays before reaching the road related to a reduced absorption of the vegetation by its transparency properties. On the opposite, we observe an underestimation of the absorbed flux regarding the two walls: -41% for wall A and -53% for wall B. It is explained because during multiple reflections, the tree layer (which is thick in case A) facing walls accentuates the absorption of vertical surfaces in the implemented version. Nonetheless, weighting the fluxes at the canyon scale affects greatly the road and garden fractions. In the reference cases, the high vegetation is treated as a ground-based vegetation fraction which is included in the garden fraction ($\delta_g$). Consequently, the road fraction, which is always defined as $1 - \delta_g$, is varying according to the vegetation layout. In this way, compared to the current version, a larger absorbed shortwave radiation flux in W m$^{-2}$ of road may lead to a lower flux expressed in W m$^{-2}$ of canyon when the garden fraction is high. During the summer and all configurations confounded (Fig. 8), we observe a mean underestimation of the shortwave radiation absorption of -44% for the road and -10% for the walls, compared to the implemented version. This can have a significant impact on thermal comfort conditions for pedestrians as well as on energy consumption of buildings for air-conditioning usage.

Moreover, the reference case presents a strong dispersion in the simulation of solar fluxes absorbed by the vegetation, depending on the configurations (aspect ratio of canyon, street orientation and vegetation layout). In some cases especially, this

flux is greater than in the new version of TEB. This effect is related to the fact that the reference version does not treat the vegetation as a semi-transparent object and does not take into account its transmissivity properties. As a result, the vegetation absorbs all the solar radiation which is received and not reflected. In the present case, this part is 75 % of the incident solar radiation since $\alpha_t = 0.25$. In the new version, 50 % of the received solar radiation is transmitted through the foliage and 75 % of the remaining flux is absorbed by trees, i.e. 37.5 % of the total shortwave radiation received by trees. On the other hand, this defect could be counterbalanced by the fact that vegetation is much more submitted to shade of buildings since it is assumed to be on the ground in the former version of radiative calculations. The mean underestimation of the initial version is -6% but the boxplots show opposite tendencies regarding the medians because of contrasted results among the latest simulations.

**6.4.2   Analysis of an integreted mean daily canyon albedo**

The mean daily albedo of the canyon ($\alpha_{can}$) is calculated as the ratio between the outgoing shortwave radiation (which is deduced from the difference between the incoming radiation $S_{can}^{\Downarrow} + S_{can}^{\downarrow}$ and the absorbed radiation $S_{can}^{*}$ ) and the incoming shortwave radiation:

$$\alpha_{can} = \frac{\left(S_{can}^{\Downarrow} + S_{can}^{\downarrow}\right) - S_{can}^{*}}{\left(S_{can}^{\Downarrow} + S_{can}^{\downarrow}\right)} \tag{22}$$

For each simulation, these fluxes are integrated over the daily cycle in order to compute a mean canyon albedo for each case. The results in summertime are presented as boxplots (Fig. 9) that gather street orientations, vegetation layouts, but that distinguish the canyon aspect ratios. It is a synthetic indicator of TEB enhancement at the scale of the entire canyon, which is crucial when SURFEX is run coupled with an atmospheric model such as the research MESO-NH model (Lafore et al., 1997) or the operational model named AROME (Seity et al., 2011) to provide climatic simulations.

Since the geometry and radiative properties of the vegetationless canyons and ground-based vegetation canyons in the reference version of TEB are comparable, they provide similar canyon albedos. It is explained by the absence of foliage in the vertical plane and identical albedos of road and garden ($\alpha_r = \alpha_g = 0.25$). Their canyon albedos rise with an increasing aspect ratio. This demonstrates the impact of greater shading effects of walls on the total absorption by the facets in deep streets, restraining the penetration of the light. No differences are observed between simulations of $h/w = 2$ canyons since the ground-based vegetation have no defined thickness. On the opposite, the canyon albedo of the implemented version of TEB strongly decreases with an increasing aspect ratio. It is expected results because the absorption is enhanced by correcting the height of the intercepting surface at the first reflection (Eq. 1) which is less shaded by the buildings than a ground-based surface. Moreover, the trapping effect is strengthened by the presence of the canopy within the canyon during the multiple reflections. Adding the processes of attenuation occurring at each interactions between the other facets or the sky (Eq. B1, B2, B3, B4) also contributes to improve the simulation of albedo by the current TEB version. We observe a greater variability of the mean daily canyon albedo after the implementation of an explicit high vegetation stratum for the $h/w = 1$ urban forms for which the weight of the tree fraction is relatively higher. In deep streets, the impact of the vegetation layout on the canyon albedo depends

on the location of the trees. When the canopy fills the low part of a $h/w = 2$ canyon, vegetation have a limited influence, contrary to cases with rescaled vegetation in comparable urban forms.

**7  Conclusions**

In order to investigate some of the physical processes related to the presence of vegetation in urban environment, e.g. for mi-
5   croclimate, hydrology or building energy consumption issues, the modeling is definitely a necessary tool.

The TEB model has been refined and improved in order to explicitly represent street trees and their impacts on radiative transfers. The new parameterization is based on the simple hypotheses of TEB: (1) a little detailed geometry without specific spatial arrangement of ground-based surfaces and (2) a single view factor to each emitting and receiving surface applied for
10   radiative calculations.

To take into account the tree canopy in TEB, it was however required to add a new vegetated stratum on the vertical plane, which can shade the road, the walls and the low vegetation. This modification led to more complex radiative calculations, but has been done with a concern to preserve a certain level of simplicity and to limit the number of new input parameters for
15   TEB. It is important to emphasize that the model is designed to be run over whole cities, for which it can simulate the local climatic variability related to urban landscape heterogeneities at the neighborhood scale. This means that computing times must be acceptable, and that input urban data must be available or easy to define. Consequently, the high vegetation is here described using only five input parameters: cover fraction of trees, height of trees, height of trunks, Leaf Area Index and albedo.

20   This simplified characterization of high vegetation necessarily induces some uncertainties on solar radiative exchanges. We estimated it by carrying out a comparative exercise between TEB and a high spatial-resolution solar and lighting model (SOLENE). On the basis of an idealized geometry of urban canyon with various vegetation layouts, TEB is evaluated regarding the direct and diffuse solar radiation received by the elements that compose the canyon. TEB simulations in summer gathered best scores for all configurations and surfaces considered, which is precisely the most relevant season to assess the
25   cooling effect of deciduous trees under temperate climate. Statistical scores have demonstrated a good capacity of TEB to solve the radiative balance of canyons without vegetation despite the use of a unique sky view factor for each facet of the urban scene. Biases less than $\pm 1$ W m$^{-2}$ related to the direct and diffuse solar radiation fluxes received by road and vertical surfaces have been recorded. Additionally, it is necessary to put in perspective obtained scores with the fact that they include the error generated by the treatment of walls as a mean wall in TEB in comparison with separate walls of SOLENE. This identified error
30   is no longer existing at the canyon or daily scales. Concerning the vegetated canyons we noted a high variability of statistical scores depending on the vegetation layout. The greater uncertainties are found for the solar radiation fluxes received by the high vegetation. The Mean Absolute Error averaged over the vegetation configurations during summertime is $17.27 \pm 9.31$ W m$^{-2}$ for the direct component and $3.45 \pm 1.93$ W m$^{-2}$ for the diffuse component but these scores are associated to acceptable

biases: $-14.44 \pm 10.60$ W m$^{-2}$ and $-3.21 \pm 2.20$ W m$^{-2}$, respectively. The systematic underestimation of fluxes reaching the new tree stratum is explained by the cover fraction approach in TEB where sides of the crown are not represented.

The parameterization of shortwave radiation exchanges within the canyon is now more realistic: shading effects of trees on vertical and ground-based surfaces but also shading effects of buildings on trees are computed. This is achieved by adding a new specific cover fraction describing the horizontal extend of high vegetation. Infinite reflections within the canyon are also conditioned to the transmissivity term calculated per pairs of exchanging surfaces. This study demonstrate the enhancement of new developments on the computed absorbed shortwave radiation fluxes within the canyon between the former reference version of TEB and the implemented version. In the current version, trees can intercept and absorb the direct solar radiation at the canopy level instead of from the ground. Consequently, the walls are more shaded. The road fraction is now independent of the cover of high vegetation. This implies that the weighting of fluxes at the canyon scale is correct. The mean daily canyon albedo is lesser than in reference cases in relation with a better representation of the radiation trapping. Canyon albedo is also more vegetation-responsive for $h/w = 1$ urban forms or in cases with $h/w = 2$ canyons when the vegetation is scaled regarding the depth of the street.

The future developments will focus on the separate calculation of turbulent energy fluxes for ground-based and high vegetation. The aerodynamic effect of trees on air flow within the canyon should also be parameterized. Based on this more sophisticated version of TEB, new impact studies could be conducted and greening adaptation strategies could be evaluated more precisely.

**8 Code availability**

The TEB code is available in open source via the surface modeling platform called SURFEX, downloadable at http://www.cnrm-game-meteo.fr/surfex/. This Open-SURFEX will be updated at relatively low frequency (each 3 to 6 months) and developments presented here are not included in the last version yet. If you need more frequent updates, or if you need what is not in Open-SURFEX (DrHOOK, FA/LFI formats, GAUSSIAN grid), we invite you to follow the procedure to get a SVN account and to access real-time modifications of the code (see instructions at the previous link).

**Appendix A: Sky view and view factors**

Sky view factors for road, garden and wall (Eq. A1, A2) as well as the view factors between elements remain unchanged in relation to initial version of the radiative calculations in TEB (described in Masson (2000) and Lemonsu et al. (2012)):

$$\Psi_{rs} \quad = \quad \Psi_{sr} = \sqrt{\left(\frac{h}{w}\right)^2 + 1} - \frac{h}{w} \tag{A1}$$

$$\Psi_{wr} \quad = \quad \Psi_{ws} = \frac{\frac{1}{2}\left(\frac{h}{w} + 1 - \sqrt{\left(\frac{h}{w}\right)^2 + 1}\right)}{\frac{h}{w}} \tag{A2}$$

$$\Psi_{sw} \quad = \quad 1 - \Psi_{sr} \tag{A3}$$

$$\Psi_{ww} \quad = \quad 1 - 2\Psi_{ws} = \frac{\sqrt{\left(\frac{h}{w}\right)^2 + 1} - 1}{\frac{h}{w}} \tag{A4}$$

$$\Psi_{rw} \quad = \quad 1 - \Psi_{rs} = 1 - \left(\sqrt{\left(\frac{h}{w}\right)^2 + 1} - \frac{h}{w}\right) \tag{A5}$$

For the tree canopy, the sky view-factor and view factors from road and walls are computed in the middle of the canyon and at mid-height of crown:

$$\Psi_{st} \quad = \quad \sqrt{\left(\frac{h}{w} \cdot \frac{h - h_{cw}}{h}\right)^2 + 1} - \left(\frac{h}{w} \cdot \frac{h - h_{cw}}{h}\right) \tag{A6}$$

$$\Psi_{rt} \quad = \quad \sqrt{\left(\frac{h}{w} \cdot \frac{h_{cw}}{h}\right)^2 + 1} - \left(\frac{h}{w} \cdot \frac{h_{cw}}{h}\right) \tag{A7}$$

$$\Psi_{wt} \quad = \quad 1 - \frac{1}{2}\left(\Psi_{st} + \Psi_{rt}\right) \tag{A8}$$

**Appendix B: Mean radiative transmissivity of canyon tree canopy**

The multiple reflections of solar radiation inside the canyon (as detailed in Masson (2000) and Lemonsu et al. (2012)) are now affected by the presence of trees whose foliage intercepts, reflects and absorbs a part of the energy. The transmissivity of radiation through the foliage of tree canopy is variable according to the way the rays cross the foliage and the distance they travel. According to the location of tree crowns inside the canyon and the dominant orientation of radiation (e.g. emission from sky to ground or from wall to ground do not reach the foliage the same way), these rays can cross all the foliage thickness or

only a small portion. The vertical distribution of leaves in the tree crowns has also an impact on transmissivity.

Different transmissivity functions (referred to as $\tau_{12}$ for exchanges between element 1 and element 2) are calculated depending of the surfaces involved in radiative exchanges. One distinguishes four cases of radiation exchanges, by hypothesizing that transmissivity functions are symmetric, i.e., exchanges from element 1 to element 2 is equivalent to the reverse way: between ground-based surfaces and sky ($\tau_{rs} = \tau_{sg}$) or wall ($\tau_{rw} = \tau_{gw}$), between wall and sky ($\tau_{ws}$), between wall and wall ($\tau_{ww}$). For each case we admit, according to Lee and Park (2008), that majority of radiation exchanges occurs in a specific zone of the canyon, for which the Leaf Area Density ($LAD_t$ expressed in $m^2$ of leaf area per $m^3$ of volume) is calculated:

$$\tau_{rs} = 1 - \delta_t \left[ 1 - \exp^{\left( - k \int_0^h LAD_t \, dz \right)} \right] \tag{B1}$$

$$\tau_{rw} = 1 - \delta_t \left[ 1 - \exp^{\left( - k \int_0^{\frac{h}{2}} LAD_t \, dz \right)} \right] \tag{B2}$$

$$\tau_{ws} = 1 - \delta_t \left[ 1 - \exp^{\left( - k \int_{\frac{h}{2}}^h LAD_t \, dz \right)} \right] \tag{B3}$$

$$\tau_{ww} = 1 - \delta_t \left[ 1 - \exp^{\left( - k \int_{\frac{h}{4}}^{\frac{3h}{4}} LAD_t \, dz \right)} \right] \tag{B4}$$

Note that these expressions are in accordance with the one applied in Eq. 3 in which $LAD_t$ is integrated on the entire thickness of foliage so that it is equivalent to $LAI_t$. As mentioned above, since the tree crown is described as a parallelepiped with a regular distribution of leaves, a uniform vertical profile of LAD is here applied.

**Appendix C: Total shortwave radiation absorption by solving infinite reflections**

For solar radiation calculations, the TEB model takes into account an infinite number of reflections between all elements composing the urban canyon. At each reflection, the isotropic radiation intercepted by a given element (1) after reflections on one of the other elements (2) is conditioned by the view factor of (2) from (1) referred to as $\Psi_{12}$ (see Appendix A), the mean radiative transmissivity $\tau_{12}$ (see Appendix B) and the reflection is then determined according to reflective properties of (1). As seen in the section 4.4, the total shortwave radiation absorbed by each elements of the canyon or redirected towards the sky is function of infinite reflections $R_\infty$, $G_\infty$, $A_\infty$, $B_\infty$, and $T_\infty$ that are still unknowns at this stage. These terms include the first reflection on each element: $R_0$, $G_0$, $A_0$, $B_0$, $T_0$. For road, garden, and walls, $R_0$, $G_0$, $A_0$ and $B_0$ simply depend on the incident solar radiation on the surface and the albedo. From the equations 14, 15, 16, 17 and 18, we can deduce the reflected part occuring at the $n+1^{th}$ absorption as the complementary term of the $n+1^{th}$ solar radiation reception for only opaque elements

of the canyon, *road*, *garden*, *wall A*, and *wall B*, respectively. As examples, we obtain for road and wall A, respectively:

$$R_0 = \alpha_r \left( S_r^{\Downarrow} + S_r^{\downarrow} \right) \tag{C1}$$

$$R_{n+1} = R_0 + \alpha_r \left[ \Psi_{rw} \tau_{rw} W_n + c_{rt} \Psi_{rt} \delta_t T_n \right] \tag{C2}$$

$$A_0 = \alpha_w \left( S_w^{\Downarrow} + S_w^{\downarrow} \right) \tag{C3}$$

$$A_{n+1} = A_0 + \alpha_w \left[ \Psi_{ww} \tau_{ww} B_n + \Psi_{wr} \tau_{wr} \left( \delta_r R_n + \delta_g G_n \right) + c_{wt} \Psi_{wt} \delta_t T_n \right] \tag{C4}$$

The specific coefficients associated to the view factors related to the high vegetation in shortwave reflections calculations are defined as:

$$c_{rt} = 0 \tag{C5}$$

$$c_{wt} = \frac{1 - \Psi_{st}}{2 - \Psi_{st} - \Psi_{rt}} \tag{C6}$$

Only reflections from trees to sky or top part of walls are allowed. As explained in section 4.4, this mode of reflection by the leaves during the first reflection, which is by far the most energetic one, is more likely to occur than in an isotropic way. This assumption could be easily bypassed by fixing the previous coefficients to 1.

For the first tree canopy reflection, the part of received direct solar radiation is corrected by the transmitted flux (see Eq. 2):

$$T_0 = \alpha_t \left[ \left( S_t^{\Downarrow} - S_t^{\ggg} \right) + S_t^{\downarrow} \right] \tag{C7}$$

Some uncertainties remain about relevance of sky view or view factors which ones could formulate to represent reflective contributions from other surfaces at $n^{th}$ to the absorption or reflection by the tree layer at $n + 1^{th}$, as well as potential absorption of energy within the tree's crown. Consequently, the solar flux reflected by *tree* at $n + 1^{th}$ has been determined as the residual term by assuming that the $n^{th}$ solar reflection coming from each element (*road*, *garden*, *wall A*, and *wall B*) which is not

20  received by *road*, *garden*, *wall A*, *wall B* or which is not returned to *sky* at $n + 1$ is received by *tree*.

During each interreflection, a part of $n^{th}$ reflected flux which is potentially available for the $(n + 1)^{th}$ reflection or absorption is intercepted by high vegetation. This intercepted part related to the presence of foliage on the way of scattered rays towards each receiving element or sky are formulated as following:

$$I_s\,(n+1) \;=\; \Psi_{sr}(1-\tau_{sr})(\delta_r R_n + \delta_g G_n) + \Psi_{sw}\,(1-\tau_{sw})\,\frac{A_n + B_n}{2} \tag{C8}$$

$$I_r\,(n+1) \;=\; \Psi_{rw}\,(1-\tau_{rw})\,\frac{A_n + B_n}{2} \tag{C9}$$

$$I_g\,(n+1) \;=\; \Psi_{rw}\,(1-\tau_{rw})\,\frac{A_n + B_n}{2} \tag{C10}$$

$$I_{w_A}\,(n+1) \;=\; \Psi_{wr}(1-\tau_{wr})(\delta_r R_n + \delta_g G_n) + \Psi_{ww}\,(1-\tau_{ww})\,B_n \tag{C11}$$

$$I_{w_B}\,(n+1) \;=\; \Psi_{wr}(1-\tau_{wr})(\delta_r R_n + \delta_g G_n) + \Psi_{ww}\,(1-\tau_{ww})\,A_n \tag{C12}$$

Finally, the solar flux which is intercepted by $tree$ at $(n+1)$ is expressed as the sum of interceptions on the way of receiving elements of the canyon or sky:

$$I_t\,(n+1) = \frac{1}{\delta_t}\left(I_s\,(n+1) \;+\; \delta_r I_r\,(n+1) \;+\; \delta_g I_g\,(n+1) \;+\; \frac{2h}{w}\,\frac{I_{w_A}\,(n+1) + I_{w_B}\,(n+1)}{2}\right) \tag{C13}$$

The solar energy which is reflected by tree at (n+1) is consequently:

$$T_{n+1} \;=\; \alpha_t\, I_t\,(n+1) \tag{C14}$$

The formulations can be simplified by gathering the equations for walls in a single expression for a mean wall according to:

$$W\,(n+1) = \frac{A\,(n+1) + B\,(n+1)}{2} \tag{C15}$$

As a result, after an infinite number of reflections, the equation system can be written:

$$R_\infty \;=\; R_0 + \alpha_r\big[\Psi_{rw}\tau_{rw}W_\infty + c_{rt}\Psi_{rt}\delta_t T_\infty\big] \tag{C16}$$

$$G_\infty \;=\; G_0 + \alpha_g\big[\Psi_{rw}\tau_{rw}W_\infty + c_{rt}\Psi_{rt}\delta_t T_\infty\big] \tag{C17}$$

[revised manuscript text omitted]

---

## Referee Report (RR1)

**Implementation of street trees within the solar radiative exchange parameterization of TEB in SURFEX v8.0 by Redon et al.**
*gmd-2016-157*

**Overview**

The article is improved in many ways and clearer. In general, the authors have done a good job at revising the paper in terms of the clarity of written language. However, there remain a few key points to address, some areas to improve clarity, and some puzzling results. Overall, there are some key assumptions in the TEB tree model that are not fully expressed in the article, and I am not fully convinced by the utility of comparing the new model with SOLENE, nor are the differences in model assumptions and design fully clear. Finally, Figure 8 suggests that street trees increase solar absorption by walls and roads – both unlikely - and I suggest a different approach to presenting the results. The results in Fig. 9 appear worrisome as well.

**Specific comments**

**Context in Abstract**. P1 L16-18: Since these numbers will presumably depend on latitude as well as season, the mean of solar radiation fluxes over the set of observations should also be reported here (or, conversely, these mean absolute differences can be reported as percentage differences) to give the reported differences context.

**Forward Scattering is neglected**. In my understanding, forward scattering is transmission through the actual leaves and radiation that is reflected downward, not simply radiation that travels downward (forward) by virtue of not hitting a leaf/foliage element. In this understanding, forward scattering and transmission are different processes. Please see the second paragraph of Sect. 3.3.1 in Krayenhoff et al. (2014) and associated references, as well as Campbell and Norman (1998). I suggest the authors simply indicate that they neglect forward scattering; and I suspect that they implicitly include its effects to an extent by choosing tree albedos of ~0.15-0.25, which are closer to average values for a forest (individual leaves scatter/reflect approximately 50% of incident broadband shortwave radiation, by comparison), thereby permitting higher transmission and absorption. If you agree, the fact that effects of forward scattering are also implicitly included is probably also worth stating. It is unclear if SOLENE includes forward scattering, and therefore whether testing against SOLENE provides a helpful evaluation of this approximation in TEB.

**Directionality of reflected radiation from trees in SOLENE**. P11L31-32: It is not clear how SOLENE deals with reflected radiation. Is it reflected in all directions (i.e., scattered?), or it is directionally reflected? My understanding is that TEB directionally scatters (i.e., reflected radiation from trees is only reflected upward). Does this affect the viability of the comparison?

**Value and meaning of tree albedo**. P12L24-25: Albedo of trees in SOLENE is 0.25, and it is 0.15 is TEB simulations (Table 2), or 0.25? Please clarify. Are the albedo values at the same scale in both models? I.e., in TEB the tree albedo is an integrated value for the whole vegetation layer; since SOLENE represents independent portions of the foliage, what does the foliage albedo mean in its case (see previous comment)?

**Mean absolute difference**. I suggest using the term MAD (=Mean Absolute Difference) instead of MAE (which I suggested in error), since this is a model difference, not an error. There are no errors calculated in the article as far as I see, only differences. Please correct throughout.

**Neighbourhoods with trees taller than buildings are common**. P6L32: This paragraph appears to have a strong European bias. I would estimate that perhaps the majority of urban neighbourhoods in North America, being low density, exhibit trees that are taller than buildings. This assumption therefore limits the range of neighbourhoods to which this new TEB tree model can be applied. It is important that this limitation is stated in this paragraph – perhaps it is a good candidate for 'future work' at the end of the article.

P8L4: "...the Beer Lambert law..."

P18L28: "confounded" is best replaced by "combined" or similar. Please correct throughout.

**Unclear model results**. Sect. 6.4.1 and Figure 8: Trees should reduce absorbed shortwave radiation on walls and especially road. If I understand correctly, the divergence from expectation is due to the garden vs. road fraction. Additionally, trees will generally reduce wall shortwave absorption, but we do not see this result in the figure. As such, I am not clear regarding the utility of Fig. 8; what do we learn from it? I wonder if illustration of a few key example scenarios (rather than combining all simulations) might better illustrate the different effects of trees when represented as "high" vs. "low" vegetation. Overall, the clarity of the writing in Sect. 6.4.1 requires improvement.

**Apparent model error**. Sect. 6.4.2 and Figure 9: These differences due to trees appear excessively large. I think something may be incorrect. If you look at Fig. 6c of Krayenhoff et al. (2014) you will see that overall albedo varies little as a function of tree foliage presence (or lack thereof) and tree foliage height – for one point in time. What do your results look like (what is the effect of trees) for one particular point in the middle of a summer day? Do these large differences result from the diurnal averaging or the model implementation? Moreover, I suspect overall albedo should be lower in the TEB reference simulations given input road and wall albedos, and canyon albedo should definitely decrease with increasing H/W. Therefore, there appears to be something incorrect in these TEB reference simulations as well. The last paragraph on P19 contains some errors of interpretation in my opinion: a shallower canyon (lower H/W) yields less radiation trapping, less absorption (per horizontal area), and higher albedo. Trees can have multiple effects; if their greater sky view factor and reflectivity are more important than their trapping of solar radiation deep in the canyon, they can reduce absorption, otherwise they can increase it. Moreover, it has been found that trees have a smaller effect as H/W increases (e.g. Coutts et al. 2016, Theoretical and Applied Climatology). Fig. 9 suggests the opposite. It

seems clear that there is a problem with these results. The clarity of the writing in Sect. 6.4.2 also requires improvement.

P22EqA6-A8: It is not clear what "hcw" represents, nor where these equations derive from. Please provide more details on both fronts.

**Integration limits for view factor calculation**. P23EqB1-4: The choices of integration limits here appear arbitrary. Are model results sensitive to the choice of these limits? More importantly, the middle of the walls (h/4 to 3h/4) is weighted more heavily in these view factor calculations. It seems critical to me that all parts of the walls be weighted equally in the view factor calculation. So, for example, the 3h/4 to h should be more important for wall-sky view factors than h/2 to 3/4h, if the latter is already used for wall-to-wall view factor calculation. Please address this in the model design, or defend your choice in the article.

---

## Author Response (AR2)

*Summary Of Changes 2*
**Implementation of street trees within the solar radiative exchange parameterization of TEB in SURFEX v8.0 by Redon et al.**
*gmd-2016-157*

Firstly, we would like to express our thankfulness and appreciation to the reviewers for their useful comments to improve the paper. We have addressed all the comments as explained below.

**Overview:**

Considering the useful comments of the Referees we proceed to some changes in the article in order to provide full explanations of the purpose of our experiment. Details about the comparison work and its relevance have been added. Some assumptions and the associated errors that they induced have been discussed as well as present limitations to the application of the TEB model. Additional perspectives for future works have been mentioned. The vegetation fractions in the reference simulations of TEB have been corrected. Geometrical assumptions about the transmission process of radiation fluxes during multiple reflections by tree crowns have been changed a little for wall-wall exchanges. Figures have been revised in order to improve the understanding of key results. All the figures and text have been adapted throughout.

**REFEREE 1 :**

*« Context in Abstract. P1 L16-18: Since these numbers will presumably depend on latitude as well as season, the mean of solar radiation fluxes over the set of observations should also be reported here (or, conversely, these mean absolute differences can be reported as percentage differences) to give the reported differences context. »*

We added the Percentage Differences to the global statistics about the tree layer during summertime, all vegetation layouts combined, in the abstract (P1 L15,18) and the conclusion (P21 L14,20).

*« Forward Scattering is neglected. In my understanding, forward scattering is transmission through the actual leaves and radiation that is reflected downward, not simply radiation that travels downward (forward) by virtue of not hitting a leaf/foliage element. In this understanding, forward scattering and transmission are different processes. Please see the second paragraph of Sect. 3.3.1 in Krayenhoff et al. (2014) and associated references, as well as Campbell and Norman (1998). I suggest the authors simply indicate that they neglect forward scattering; and I suspect that they implicitly include its effects to an extent by choosing tree albedos of ~0.15-0.25, which are closer to average values for a forest (individual leaves scatter/reflect approximately 50% of incident broadband shortwave radiation, by comparison), thereby permitting higher transmission and absorption. If you agree, the fact that effects of forward scattering are also implicitly included is probably also worth stating. It is unclear if SOLENE includes forward scattering, and therefore whether testing against SOLENE provides a helpful evaluation of this approximation in TEB. »*

Thank you for pointing out the lack of explanations about assumptions of the TEB model concerning the forward scattering. If we consider the transmission process *sensu stricto,* it does not include forward scattering (the radiation that is transmitted through the canopy by reflections from

leaves to leaves before be reflected downward and leave the canopy) but only the radiation that is passing through the canopy without hitting leaves. We omitted to fully mention the following assumptions on which is based the radiative transfer in TEB.

During the first reflection / absorption / transmission of the solar shortwave radiation, the canopy is assumed to be an horizontal partially-transparent surface, characterized by a cover fraction and an extinction coefficient. This vegetated surface is placed at the height of trees because, in reality, most of the sunlit leaves are bordering the top of the canopy. For a rectangular parallelepiped shaped canopy, the maximum of interception is expected at the mid-height of the crown when an homogeneous distribution of foliage elements is respected. This hypothesis implies that we consider the integration of overlapping leaves on Pi (180°) only (that means the upper part of the canopy, from the middle to the top of crown) to represent the attenuation process. The intercepted flux is divided in reflection or absorption processes after, respecting the optical properties of the leaves (albedo). In this system, no more radiation is available below the mid-height of the crown to be forward scattered. Because of the description of the crown as a surface, TEB computes no processes within the crown as interactions leaves-to-leaves and trapping effect inside the canopy. Nonetheless, the forward scattering is implicitly included in attenuation or transmission processes in the upper part of the canopy. The relative importance of attenuation or transmission processes are finally applied to the top of the canopy.

Here, the choice of an albedo of 0.25 is related to the will of controlling the effect of garden fraction on the road fraction in the reference cases (initial version of TEB). It is a value that is acceptable for the two covers. Fixing the same value for these two fractions allows us to analyze the effect of the position of the trees (as a ground-based surface or an explicit canopy within the canyon) and the transparency properties of trees (opaque or partially-transparent cover) between the reference or the implemented simulations without the confusing effect of divergent radiative properties (different albedos).

In SOLENE model, while the forward scattering in the crown is also neglected and the same processes (reflection / absorption / transmission) are represented, the simulations of the received solar radiation fluxes in the two models remain comparable.

See P10 L11-14, P12 L16-17, P21 L 23-24

*« Directionality of reflected radiation from trees in SOLENE. P11L31-32: It is not clear how SOLENE deals with reflected radiation. Is it reflected in all directions (i.e., scattered?), or it is directionally reflected? My understanding is that TEB directionally scatters (i.e., reflected radiation from trees is only reflected upward). Does this affect the viability of the comparison? »*

SOLENE deals with reflections using the radiosity method, that means omnidirectional reflections. TEB also computes isotropic reflections that are just constrained upwards from the height of the middle of the crown on Pi (180°). In conclusion, the process of reflection is comparable between the two models.

See P11 L22-24, P12 L7-8, P14 L11, P18 L18-19.

*« Value and meaning of tree albedo. P12L24-25: Albedo of trees in SOLENE is 0.25, and it is 0.15 is TEB simulations (Table 2), or 0.25? Please clarify. Are the albedo values at the same scale in both models? I.e., in TEB the tree albedo is an integrated value for the whole vegetation layer; since SOLENE represents independent portions of the foliage, what does the foliage albedo mean in*

Thank you for pointing out an error in the table 2: the albedo of the high vegetation is 0.25 in both models. In the initial version of TEB, the albedo of the garden fraction (integrated fraction of low and high vegetation) is fixed to 0.25 while we can fix a specific albedo at the high vegetation only in the implemented version. In SOLENE model, the albedo is uniformly applied at the whole vegetated surface of rectangular parallelepipeds. In other words, all the meshes which belong to the surface of vegetation blocks are characterized by the same albedo (=0.25). A transmission factor of 0.5 is also applied to each meshes of vegetation blocks in SOLENE.

See Table 2 and P12 L14-15, P13 L2-3, P13 L23-24

*« Mean absolute difference. I suggest using the term MAD (=Mean Absolute Difference) instead of MAE (which I suggested in error), since this is a model difference, not an error. There are no errors calculated in the article as far as I see, only differences. Please correct throughout. »*

We agree with the referee that the term 'difference' is more suitable than 'error' here and we corrected throughout.

*« Neighbourhoods with trees taller than buildings are common. P6L32: This paragraph appears to have a strong European bias. I would estimate that perhaps the majority of urban neighbourhoods in North America, being low density, exhibit trees that are taller than buildings. This assumption therefore limits the range of neighbourhoods to which this new TEB tree model can be applied. It is important that this limitation is stated in this paragraph – perhaps it is a good candidate for 'future work' at the end of the article. »*

We agree with the referee that the assumption in TEB about the restriction of urban trees to the canyon space, which can not be taller than buildings, constitutes a limitation on the range of neighborhoods to which TEB can be applied. The errors induced by this limitation need to be estimated and future works could remedy this lack by allowing trees taller than buildings when it is necessary.

See P6 L32, P7 L1-11, P22 L11-12.

*« P8L4: "...the Beer Lambert law..." »*

We corrected this typing error.

*« P18L28: "confounded" is best replaced by "combined" or similar. Please correct throughout. »*

We agree with the referee that the term 'combined' is more suitable than 'confounded' here and corrected throughout.

*« Unclear model results. Sect. 6.4.1 and Figure 8: Trees should reduce absorbed shortwave radiation on walls and especially road. If I understand correctly, the divergence from expectation is due to the garden vs. road fraction. Additionally, trees will generally reduce wall shortwave absorption, but we do not see this result in the figure. As such, I am not clear regarding the utility of Fig. 8; what do we learn from it? I wonder if illustration of a few key example scenarios (rather than combining all simulations) might better illustrate the different effects of trees when represented as "high" vs. "low" vegetation. Overall, the clarity of the writing in Sect. 6.4.1 requires*

*improvement. »*

Section 6.4.1 : As suggested by the Referee 1, we changed the figure 8 in order to demonstrate more clearly the divergent dealing with trees between the initial and the implemented versions of TEB. We took the example of the A configured canopy (for which the cover fraction of the high vegetation in the canyon is 0.90) within a h/w=1 canyon, oriented North-South, during a summer day. Globally, we can see that considering the fluxes per squared meters of surface, the artificial surfaces absorb less energy when they are shaded by the high vegetation, while the daily dynamics are similar. For the beginning and the end of the range of lighting hours, trees trap more energy after the implementation of a real canopy within the canyon instead of cases where the foliage is placed at ground. It is explained by the fact that the more the trees are high in the canyon, the more they can receive solar radiation and avoid the shading of buildings. Around the solar noon, the absorption by the trees is substantially corrected by the effect of the partial-transparency of the canopy in regards to the initial version where all the urban vegetation surfaces were opaque.

Comparing the fluxes before and after their weighting based on the canyon surface, the road absorption by squared meters of canyon can extremely vary following the garden fraction at the ground in the initial version, so the contribution of the road to the total absorption of the canyon. Here, the road fraction is only 0.10. For the same version, the differences between before and after the weighting of absorbed fluxes is small because the tree fraction is high (=0.90). In this case, there is no differences for the walls according to the unit of the fluxes since the canyon aspect ratio is h/w=1. For smaller aspect ratios, the fluxes of walls based on the canyon surface will be lower. On the opposite, for aspect ratios exceeding 1, the fluxes of walls based on the canyon surface will be higher.

*« **Apparent model error. Sect. 6.4.2 and Figure 9:** These differences due to trees appear excessively large. I think something may be incorrect. If you look at Fig. 6c of Krayenhoff et al. (2014) you will see that overall albedo varies little as a function of tree foliage presence (or lack thereof) and tree foliage height – for one point in time. What do your results look like (what is the effect of trees) for one particular point in the middle of a summer day? Do these large differences result from the diurnal averaging or the model implementation? Moreover, I suspect overall albedo should be lower in the TEB reference simulations given input road and wall albedos, and canyon albedo should definitely decrease with increasing H/W. Therefore, there appears to be something incorrect in these TEB reference simulations as well. The last paragraph on P19 contains some errors of interpretation in my opinion: a shallower canyon (lower H/W) yields less radiation trapping, less absorption (per horizontal area), and higher albedo. Trees can have multiple effects; if their greater sky view factor and reflectivity are more important than their trapping of solar radiation deep in the canyon, they can reduce absorption, otherwise they can increase it. Moreover, it has been found that trees have a smaller effect as H/W increases (e.g. Coutts et al. 2016, Theoretical and Applied Climatology). Fig. 9 suggests the opposite. It seems clear that there is a problem with these results. The clarity of the writing in Sect. 6.4.2 also requires improvement. »*

Section 6.4.2: We found an error on vegetation fractions used in the initial version of TEB for the comparison of canyon albedos. We also changed the way to calculate the albedo, considering the solar noon albedo only, during a summer day, as suggested by the Referee 1. After this correction, the results are in accordance with the expected tendencies :
- the aspect ratio have a strong effect on the canyon albedo; the more the aspect ratio is high, the lower is the canyon albedo
- for canyons with h/w=0.5 aspect ratios, the canyon albedo of streets with an explicit tree canopy is lower than canyons with leaves represented at the ground

- for higher aspect ratios, the effect of trees is depending on the major process impacting the absorption of the canyon, increasing or reducing the total absorption of solar radiation by the canyon, but their relative impact is smaller if the thickness of the crown is not proportionally rescaled in accordance with the height of the buildings (thicker crown)

*« P22EqA6-A8: It is not clear what "hcw" represents, nor where these equations derive from. Please provide more details on both fronts. »*

The description of the symbol hcw is in Table 1. We added the symbol just after the associated reference in the text P23 L15 and explanations about the formulation of new view factors related to the high vegetation in Appendix A.

*« Integration limits for view factor calculation. P23EqB1-4: The choices of integration limits here appear arbitrary. Are model results sensitive to the choice of these limits? More importantly, the middle of the walls (h/4 to 3h/4) is weighted more heavily in these view factor calculations. It seems critical to me that all parts of the walls be weighted equally in the view factor calculation. So, for example, the 3h/4 to h should be more important for wall-sky view factors than h/2 to 3/4h, if the latter is already used for wall-to-wall view factor calculation. Please address this in the model design, or defend your choice in the article. »*

In order to ensure their geometric consistency, the choice of the limits 0-h, 0-h/2, h/4-3h/4 and h/2-h in the calculation of the mean transmissivity terms are based on sectors already scanned by associated view factors used in isotropic interactions, both from 1- the incoming diffuse solar radiation flux or 2- multiple reflections within the canyon.

For reception of incoming diffuse solar radiation, Psi_rs and Psi_ws are respectively multiplied by tau_rs and tau_ws (see Eq. 8,9). Terms related to RS (or ground – sky) interactions are defined on the entire height of the building (0-h). Terms related to WS (wall -sky) interactions are defined on the upper part of the canyon, from the middle of the building's height (h/2-h).

During multiple reflections, Psi_rw and Psi_ww are respectively multiplied by tau_rw and tau_ww (see Eq. C16, C18 in Appendix C). Terms related to RW (or ground – wall) interactions are defined on the lower part of the canyon, to the middle of the building's height (0-h/2). Terms related to WW (wall -wall) interactions are defined on the entire height of the building (0-h). We tested to restrict the sector used to calculate tau_ww to h/4-3h/4 as Lee and Park (2008) and because this term could be sensitive to the horizontal position of trees in the street, particularly their distance to walls. Since TEB does not take into account this parameter, we corrected the calculation of tau_ww as on the 0-h sector, consistently with the Psi_ww formulation.

Classical sky view factors from sky to road or from sky to wall have parts in common and overlapping each other (Appendix A). In this way, there is no reason to reject transmissivity terms based on overlapping sectors.

To test the sensitivity of results to this parameterization of the impact of trees on radiative transfers of isotropic radiation, we compared four configurations.

| Configuration / Area per interaction | 3 thirds | Lee and Park, 2008 | Former version of article | New version based on view factors |
|---|---|---|---|---|
| road- sky | 0 – h | 0 – h | 0 – h | 0 – h |
| wall – sky | 2h/3 – h | 3h/4 – h | h/2 – h | h/2 – h |
| road -wall | 0 – h/3 | 0 – h/4 | 0 – h/2 | 0 – h/2 |
| wall -wall | h/3 – 2h/3 | h/4 – 3h/4 | h/4 – 3h/4 | 0 – h |

We obtained the following results for the total shortwave absorption by surfaces, averaged by all the vegetation layouts, urban forms and street orientations during a summer day. Scores are calculated in the same way that in the article, and versus SOLENE simulations which simulates digital mock-ups with explicit representation of the tree crowns and allocate fluxes in a directional manner.

| Configuration /Scores | 3 thirds | Lee and Park, 2008 | Former version of article | New version based on view factors |
|---|---|---|---|---|
| MAD road | 17.62 | 17.89 | 16.26 | 16.31 |
| MAPD road | 26 | 27 | 24 | 24 |
| Bias road | +16.96 | 17.27 | +15.53 | +15.56 |
| MAD walls | 11.53 | 11.18 | 10.18 | 9.39 |
| MAPD walls | 17 | 17 | 15 | 14 |
| Bias walls | +11.03 | +10.66 | 9.22 | 8.15 |
| MAD trees | 25.42 | 25.75 | 22.14 | 21.12 |
| MAPD trees | 28 | 28 | 22 | 21 |
| Bias trees | -23.71 | -23.44 | -18.29 | -15.96 |

To our knowledge, there is no justification for cutting the canyon in the way of 3 thirds or on the Lee and Park, 2008 way. We need observations to assess which representation of attenuation / transmission is the more close to what is occurring in real cases but processes during multiple reflections can not be tracked easily. Nonetheless, we suggest a new cutting, consistent with our geometric assumptions and view factors used in TEB model.

**REFEREE 2 :**

**Page 2 Line 1** – "The high vegetation, especially the street trees…" does not make grammatical sense. Consider re-writing the sentence as "High vegetation, particularly street trees in densely built urban neighbourhoods, …"

Done

**Page 3 Line 16** – "More especially" is not the correct phrase for the start of a paragraph and doesn't really fit with the following sentence. Perhaps consider using "Many different approaches are currently applied…"

Done

**Page 3 Line 17** – "Among following single-layer models" does not make sense. Do you mean to say "Among single-layer models with integrated vegetation schemes, Lee and Park (2008) were the first…"?

Done

**Page 5 Line 28** – Think there is a miss-spelling of the word plane ("plan" is in the text). Do you mean "on a two dimensional plane which…"?

Done

**Page 7 Line 11** – Are the trees not technically "cuboids" as opposed to "parallelepipeds", which are typically formed of 6 parallelograms? In this paper you are only considering rectangular cross sections so maybe cuboid would be a more appropriate term.

Thank you for pointing out this error; a more accurate description of trees is needed. Trees are represented as 'rectangular parallelepipeds' because they are placed along the canyon in the axis of the street without discontinuities. The potential gaps between real crowns are taken into account in the high vegetation fraction. This term is a combination of the high vegetation fraction in the perpendicular plane of the street direction and the real covering of crowns lengthwise along the canyon. In this way, the 'rectangular cross section' remains a proper expression.

See P13 L13-18.

**Page 12 Line 11** – Should the word "declined" be "defined"?

Done

**Page 12 Line 16** – Consider again using the word "cuboid" as opposed to "parallelepiped"

See answer for Page 7 Line 11 question.

**Page 12 Line 16** – Sentence starting "The vegetation blocks…" does not make sense. Does it mean to say "The vegetation blocks are continuous lengthwise along the canyon."?

We changed the sentence in the following way: "These solid volumes are continuous lengthwise along the canyon. " in order to avoid the repetition with the previous sentence. See P12 L28.

**Page 13 Line 15** – This sentence doesn't make sense. Should it read along the lines of "In SOLENE, the incident radiation is roughly attenuated by 50% once crossing the mesh containing vegetation"?

We changed the sentence in the following way: "In SOLENE, the incident radiation is roughly attenuated by 50% once crossing a mesh which belongs to a vegetation envelope.". See P13 L26-27.

**Page 15 Line 20** – The link between the example of the daily cycle and the scatterplots is not clear. This could do with some additional sentence explaining some context as to why the scatterplots were created.

We added scatterplots in order to show the extent of discrepancies between the two models according to various meteorological conditions. We know that TEB provides lower performances when the incoming diffuse radiation part is high. Scatterplots are including results for the four season in temperate climate: winter, spring, summer and autumn. Winter is characterized by a high diffuse / direct incoming solar radiation ratio. Despite these different meteorological conditions, the scores remain acceptable. See P16 L1.

**Page 15 Line 30** – An additional "the" in this sentence is not required. Should it read "The two walls have different behaviours: the wall most exposed to the sun …"?

Done

**Page 16 Line 1** – This first few sentences don't make sense. You could consider replacing them with something along the lines of "In the calculation of diffuse solar radiation, TEB does not consider the two walls separately. Therefore the diffuse solar radiation flux is compared between the composite wall of TEB, and the average of the diffuse solar radiation fluxes received by separate walls of the SOLENE simulations (Fig. 4)."

Done

**Page 18 Line 1** – The word "Remind" does not make sense here? The word "Remembering" is more appropriate.

Done

**Page 19 Line 8** – Consider re-referencing figure 8 here so the reader knows which boxplot figure you are referring to. Perhaps include after the word "boxplots"?

Done

**Page 19 Line 26** – The start of the sentence "It is expected result.." does not make sense. Should it read "This is expected because…"?

Done

**Page 19 Line 30** – The sentence beginning "We observe…" does not make sense. Should it read something like, "We observed greater variability in the mean daily canyon albedo after the implementation of an explicit high vegetation stratum for an urban form with h/w = 1 and a high relative tree fraction."?

Done

**Page 20 Line 8** – Incorrect use of plural. Use "radiative transfer" instead of "radiative transfers".

Done

**Page 21 Line 6** – Should the word "extend" be "extent"?

Done

**Page 21 Line 7** – For this sentence to make sense use word "pair" instead of "pairs".

Done

**Page 21 Line 7** – Sentence should start "This study demonstrated …"

Done

**Figure Captions**
In the figure captions the units for meters should not be in italics.

Done

**Implementation of street trees within the solar radiative exchange parameterization of TEB in SURFEX v8.0**

Emilie Redon[a], Aude Lemonsu[a], Valéry Masson[a], Benjamin Morille[b], and Marjorie Musy[b]

[a]CNRM UMR 3589, Météo-France/CNRS, Toulouse, France, 42 avenue Gaspard Coriolis, 31057 Toulouse cedex 1, France
[b]Ecole nationale supérieure d'Architecture de Nantes, UMR 1563, quai François Mitterrand, 44262 Nantes cedex 2, France
*Correspondence to:* A. Lemonsu (aude.lemonsu@meteo.fr)

**Abstract.** The Town Energy Balance (TEB) model has been refined and improved in order to explicitly represent street trees and their impacts on radiative transfer: a new vegetated stratum on the vertical plane, which can shade the road, the walls and the low vegetation has been added. This modification led to more complex radiative calculations, but has been done with a concern to preserve a certain level of simplicity and to limit the number of new input parameters for TEB to the cover fraction of trees, the mean height of trunks and trees, their specific Leaf Area Index and albedo. Indeed, the model is designed to be run over whole cities, for which it can simulate the local climatic variability related to urban landscape heterogeneity at the neighborhood scale. This means that computing times must be acceptable, and that input urban data must be available or easy to define. This simplified characterization of high vegetation necessarily induces some uncertainties in terms of the solar radiative exchanges, as quantified by comparison of TEB with a high spatial-resolution solar enlightenment model (SOLENE). On the basis of an idealized geometry of urban canyon with various vegetation layouts, TEB is evaluated regarding the total shortwave radiation flux absorbed by the elements that compose the canyon. TEB simulations in summer gathered best scores for all configurations and surfaces considered, which is precisely the most relevant season to assess the cooling effect of deciduous trees under temperate climate. Mean Absolute Differences and biases of 6.03 and +3.50 W m$^{-2}$ for road, respectively, and of 3.38 and +2.80 W m$^{-2}$ for walls have been recorded in vegetationless canyons. In view of the important incident radiation flux, exceeding 1 000 W m$^{-2}$ at solar noon, the Mean Absolute Percentage Differences of 3 % for both surfaces remain moderate. Concerning the vegetated canyons, we noted a high variability of statistical scores depending on the vegetation layout. The greater uncertainties are found for the solar radiation fluxes received and absorbed by the high vegetation. The Mean Absolute Differences averaged over the vegetation configurations during summertime is 21.12 ± 13.39 W m$^{-2}$ or 20.92 ± 10.87 % of Mean Absolute Percentage Differences for the total shortwave absorption but these scores are associated to acceptable biases: −15.96 ± 15.93 W m$^{-2}$.

[revised manuscript text omitted]

More specifically for the shortwave radiation budget, three contributions are considered for a given element:

1. The direct solar radiation received before any reflections. This contribution depends on zenith angle since the incident direct radiation is unidirectional, street orientation, and canyon aspect ratio.

2. The diffuse solar radiation received before any reflections. This contribution depends on the sky view factor of the considered element since the diffuse radiation is assumed to be isotropic.

3. The total shortwave radiation received after multiple reflections within the canyon. After a first reflection on one of the elements of the canyon, initial contributions of direct and diffuse radiation are isotropic and are treated the same way. The part of radiation received by a given element then depends on the view factors of all the other elements and on their albedo that determine the reflected radiation part.

Although this paper focuses on resolution and evaluation of the shortwave radiation budget, it is worth to note that the validation of our shortwave radiation scheme contributes to verify our future longwave radiation scheme. Indeed, the same view factors used for the multiple reflections will be applied to the longwave radiation interactions within the canyon. The longwave exchanges are computed following  a linear approximation of the Stefan-Boltzmann law.  For numerical stability purpose, an implicit formulation is applied for longwave radiation budgets; it includes the surface temperatures at the previous numerical time step and at the current time step.

**3.3 Inclusion of a high vegetation stratum for solar radiation calculation**

To take into account the tree canopy in TEB, it is required to add a new vegetated stratum on the vertical plane, which can shade the road, the walls and the low vegetation. This modification led to more complex radiative calculations, but is done with a

30 concern to preserve a certain level of simplicity and to limit the number of new input parameters for TEB. This is motivated by the type of applications which are conducted with the TEB model, and more generally with the SURFEX land surface modeling platform (Masson et al., 2013) which TEB is part of. The system is frequently applied over domains of several hundred square kilometers with horizontal resolutions between few hundred meters to few kilometers and can be run for long time periods up to several years in case of climatic studies (e.g., Lemonsu et al., 2013; de Munck, 2013, in French). This means that computing times must be acceptable, and that urban input data must be available or easy to define.

5 The arrangement of tree canopy is here described using three parameters only (Fig. 1 and Table 1): its cover fraction ($\delta_t$), i.e. the proportion of canyon which is covered by the foliage stratum on the horizontal plane, as well as the mean height of trees ($h_t$), and the mean height of trunks ($h_{tk}$). In the current version of TEB (official SURFEX v8.0 version), urban trees are assumed to be less tall than surrounding buildings and systematically confined inside the canyon so that they cannot provide shade for roofs. This hypothesis is in accordance with  common urban planning specifications for
10 street trees management in Europe (in French, Municipality of Toulouse, 2008; City of Westminster, 2009; Barcelona City Council, 2011). These documents ensure a satisfactory juxtaposition of trees with urban structures for dwellers. Minimum distances between trunks or crowns and walls or balconies are strictly imposed to avoid problems such as excessive obstruction of crowns facing windows, disruption of underground services by roots or subsidence of buildings. These widespread practices are also applied in private gardens or suburban areas and tends to avoid shadow on roofs even if street trees can be taller than
15 buildings. Design with trees shading on roofs are statistically sparse and their impact on surface balance  is limited. At this stage, the application of TEB to low density neighborhoods is restricted if trees can significantly shade buildings. They are probably located in high latitudes where incoming fluxes are  lower than in mid or low latitudes and this potential bias would only occur in the early morning or late evening, when the zenith angle is large but the solar radiation flux far lower energetic than around noon.

For now, the shape of the foliage and the vertical distribution of leaves are not defined. The crowns of trees are considered as rectangular parallelepipeds (namely computed as a rectangular  cross-section in a two dimensional plane perpendicular to the street axis) with homogeneous foliage which is described by a Leaf Area Index ($LAI_t$) and an albedo ($\alpha_t$). It is how-ever possible to vary the LAI during the year, in order to simulate the seasonal cycle of deciduous trees. The high vegetation
25 cover fraction is a combination of the sum of widths of crowns in the plane perpendicular to the street axis but also takes into account the presence or not of gaps between crowns lengthwise along the canyon. Note that trunks are not taken into account in radiative calculations. The tree vegetation stratum is considered as a partially-transparent element for shortwave radiation. A part of the incident radiation received by trees is transmitted through the foliage. The part of radiation which is not transmit-ted is consequently reflected or absorbed, depending on albedo. These processes and the associated calculations are detailed
30 hereafter.

**4 Solar radiation absorption of vegetated street canyon surfaces**

In this part, equations related to the implementation of a tree layer into the TEB model are presented. In order to calculate these terms in TEB, the 4.1 and 4.2 following sections describe how direct and diffuse solar  radiation fluxes reach canyon surfaces. Then, absorption is obtained by separately resolving the first absorption of total shortwave radiation on each surface and the sum of absorbed shortwave radiation after infinite reflections within the canyon.

**4.1 Direct solar radiation received by each element**

5    The foliage of trees plays a role of obstruction and attenuation of incident direct solar radiation ($S^{\Downarrow}$) for the other elements of the urban canyon. Consequently, to determine the direct solar radiation received by each element of the canyon, we need to solve first the equations related to high vegetation.

The direct solar radiation potentially reaching the top of trees by geometrically taking into account the shading of build-
10    ings depends on buildings height ($h$), canyon aspect ratio ($h/w$), street orientation ($\theta_{can}$), zenith and azimuth angles ($\lambda$, $\theta_{sun}$), as well as tree height ($h_t$):

$$S_t^{\Downarrow} = S^{\Downarrow} max \left[ 0 \; ; \; 1 - \frac{h}{w} \left( \frac{h - h_t}{h} \right) tan(\lambda) \, sin \, |\theta_{sun} - \theta_{can}| \right] \tag{1}$$

As  previously explained, this radiation flux is partially transmitted through the foliage ($S_t^{\gg}$), whereas the remaining solar radiation is reflected ($S_t^{\Uparrow}$) or absorbed ($S_t^*$):

15    $$S_t^{\Downarrow} = S_t^{\gg} + S_t^{\Uparrow} + S_t^* \tag{2}$$

The proportion of direct solar radiation transmitted through the foliage is estimated by  the Beer Lambert law (Campbell and Norman, 1989) where the Leaf Area Index ($LAI_t$ expressed in $m^2$ of leaves per $m^2$ of ground) of tree canopy and an extinction coefficient ($k$) are involved. The extinction coefficient is fixed to 0.5, a default value corresponding to  an homogeneous distribution of leaves in terms of density and orientation (in other words, a spherical leaf angle distribution):

20    $$S_t^{\gg} = S_t^{\Downarrow} \exp^{(- \, k \, LAI_t)} \tag{3}$$

The reflected radiation part simply depends on the part of incident solar radiation untransmitted through the foliage and on the albedo of trees ($\alpha_t$):

$$S_t^{\Uparrow} = \alpha_t \, S_t^{\Downarrow} \left( 1 - \exp^{(- \, k \, LAI_t)} \right) \tag{4}$$

Finally, the incident direct solar radiation part absorbed by trees is neither transmitted nor reflected and calculated as the
25    residual term from Eq. 2:

$$S_t^* = (1 - \alpha_t) \, S_t^{\Downarrow} \left( 1 - \exp^{(- \, k \, LAI_t)} \right) \tag{5}$$

The direct solar radiation received by the ground (indiscriminately road or garden fraction) is deduced by correcting the incident solar radiation above canyon from the interception of radiation by high vegetation canopy (i.e., reflected and absorbed radiation weighted by high vegetation cover fraction referred to as $\delta_t$), and then from the shading effects of buildings (according to Lemonsu et al., 2012). The same equations are obtained for road ($S_r^\Downarrow$) and garden ($S_g^\Downarrow$):

$$S_r^\Downarrow = \left(S^\Downarrow - \delta_t \left(S_t^\Uparrow + S_t^*\right)\right) max\left[0\,;\,1 - \frac{h}{w}\,tan(\lambda)\,sin\,|\theta_{sun} - \theta_{can}|\right] \tag{6}$$

In this way, tree foliage is assumed to be uniformly distributed accross the canyon at the height of the trees ($h_t$), consistently with the Eq. 1.

The direct solar radiation which is not received either by high vegetation, or by road or garden is assigned to the sunlighted wall, whereas the opposite wall is in the shadow. By convention in TEB in the case of an oriented canyon, we define $wall\ A$ as the most sunlit wall and $wall\ B$ as the shaded one.

$$S_{w_A}^\Downarrow = \left(S^\Downarrow - S_r^\Downarrow - \delta_t \left(S_t^\Uparrow + S_t^*\right)\right) \frac{w}{h} \qquad\qquad S_{w_B}^\Downarrow = 0 \tag{7}$$

10 Note that shading effects of high vegetation on roofs are not represented, since urban trees are less tall than buildings by definition in the current version of TEB (SURFEX v8.0).

**4.2 Diffuse solar radiation received by each element**

The incoming diffuse solar radiation ($S^\downarrow$) is assumed to emit isotropically. Each urban surface of the canyon ($wall$, $road$, and $garden$) receives a part of diffuse solar radiation according to the sky view factor of the surface $\Psi_\star$ (see Appendix A) and the

15 mean radiative transmissivity between the sky and the given surface $\tau_{\star s}$ (see Appendix B). Note that the sky view factor of $wall$ is defined at mid-height of buildings; for ground-based surfaces $road$ and $garden$, a single sky view factor $\Psi_{rs} = \Psi_{gs}$ is defined at the center of the street (Masson, 2000; Lee and Park, 2008). The following equations are obtained for $road$ (same expression for $garden$) and for $wall$:

$$S_r^\downarrow = S^\downarrow\,\Psi_{rs}\,\tau_{rs} \tag{8}$$

$$S_w^\downarrow = S^\downarrow\,\Psi_{ws}\,\tau_{ws} \tag{9}$$

We admit that the residual flux of diffuse solar radiation which is not intercepted in the canyon by previous surfaces reaches the tree canopy:

$$S_t^\downarrow = \frac{S^\downarrow - \left(\delta_r\,S_r^\downarrow + \delta_g\,S_g^\downarrow + \frac{2h}{w}\,S_w^\downarrow\right)}{\delta_t} \tag{10}$$

25 This method presents two major advantages: (1) the diffuse solar radiation budget is always closed and (2) the computed diffuse solar radiation flux for the high vegetation is already corrected from the transmitted part, reaching the other surfaces.

The fluxes of each surface are here expressed according to the total ground-based surface of the canyon, with $\delta_r$ and $\delta_g$ the cover fractions of road and garden in the canyon ($\delta_r + \delta_g = 1$), respectively.

**4.3 First absorption of total shortwave radiation by each element**

The first absorption of total shortwave radiation $S^*(0)$, before any reflections, is only function of the total shortwave radiation received by the considered element and of its albedo ($\alpha_\star$). The same expression is obtained for walls and ground-based surfaces ($\star$ which stands for $r$, $g$, $w_A$ or $w_B$):

$$S^*_\star(0) = (1 - \alpha_\star)\left(S^{\Downarrow}_\star + S^{\downarrow}_\star\right) \tag{11}$$

For the tree canopy, the part of absorbed direct solar radiation is corrected by the transmitted flux:

$$S^*_t(0) = (1 - \alpha_t)\left[\left(S^{\Downarrow}_t - S^{\ggg}_t\right) + S^{\downarrow}_t\right] \tag{12}$$

$S^{\Downarrow}_t$ includes the transmitted flux $S^{\ggg}_t$ (see Eq. 2) contrary to $S^{\downarrow}_t$ which is calculated as a residual flux (Eq. 10), corrected from the transmitted flux.

**4.4 Sum of total shortwave radiation absorbed by each element**

Our goal is to compute the total shortwave radiation absorption for each element $S^*(\infty)$ by taking into account an infinite number of reflections between all elements composing the urban canyon. At each reflection, the isotropic radiation intercepted by a given element (1) after reflections on one of the other elements (2) is conditioned by the view factor of (2) from (1) referred to as $\Psi_{12}$ (see Appendix A), the mean radiative transmissivity $\tau_{12}$ (see Appendix B) and the absorption is then determined according to reflective properties of (1). Using a single view factor in TEB radiation calculations is obviously a limitation for accurately representing the various contributions of canyon's surfaces to high vegetation. Additionally, we have a poor knowledge of  the forward scattering or trapping effects occurring within the crowns. Because of the description of the canopy as a surface, TEB neglects these processes but the attenuation and transmission applied at the top of the canopy (see Eq. 3, 4, 5) arising from Eq. 1) could be adjusted in order to implicitly include these phenomena. To ensure a closing system, we define the total absorbed shortwave radiation by high vegetation as the remaining shortwave radiation, after accounting for absorption and reflection from the total incident solar radiation by all other elements of the canyon. This requires to calculate the part of shortwave radiation which leaves the canyon towards the sky. The terms $R_\infty$, $G_\infty$, $A_\infty$, $B_\infty$, and $T_\infty$ make reference to the sum of total shortwave radiation reflected by each surface, respectively, after an infinite number of reflections (see detailed resolution in Appendix C). Here is the expression of the total absorbed solar flux per surface:

$$S^*_s(\infty) = \Psi_{sr}\tau_{sr}(\delta_r R_\infty + \delta_g G_\infty) + \Psi_{sw}\tau_{sw}\frac{A_\infty + B_\infty}{2} + \Psi_{st}\delta_t T_\infty \tag{13}$$

$$S_r^*(\infty) = S_r^*(0) + (1 - \alpha_r)\left[\Psi_{rw}\tau_{rw}\frac{A_\infty + B_\infty}{2} + c_{rt}\Psi_{rt}\delta_t T_\infty\right] \tag{14}$$

$$S_g^*(\infty) = S_g^*(0) + (1 - \alpha_g)\left[\Psi_{rw}\tau_{rw}\frac{A_\infty + B_\infty}{2} + c_{rt}\Psi_{rt}\delta_t T_\infty\right] \tag{15}$$

$$S_{w_A}^*(\infty) = S_{w_A}^*(0) + (1 - \alpha_w)\left[\Psi_{wr}\tau_{wr}\left(\delta_r R_\infty + \delta_g G_\infty\right) + \Psi_{ww}\tau_{ww}B_\infty + c_{wt}\Psi_{wt}\delta_t T_\infty\right] \tag{16}$$

5  $$S_{w_B}^*(\infty) = S_{w_B}^*(0) + (1 - \alpha_w)\left[\Psi_{wr}\tau_{wr}\left(\delta_r R_\infty + \delta_g G_\infty\right) + \Psi_{ww}\tau_{ww}A_\infty + c_{wt}\Psi_{wt}\delta_t T_\infty\right] \tag{17}$$

$$S_t^*(\infty) = \frac{1}{\delta_t}\left[\left(S_t^\Downarrow -+ S_t^\ggg + S_t^\downarrow\right) - \left(S_s^*(\infty) + \delta_r S_r^*(\infty) + \delta_g S_g^*(\infty) + \frac{2h}{w}\frac{S_{w_A}^*(\infty) + S_{w_B}^*(\infty)}{2}\right)\right] \tag{18}$$

The view factors related to the high vegetation stratum are expressed in Appendix A. Specific coefficients are applied, in the

10  shortwave scheme only, to constrain the reflections from the high vegetation toward sky and top part of walls. In nature, the solar radiation is mainly redirected upwards by the receiving face of sunlit leaves in the top of crown during the first reflection. We suggest here to neglect the small amount of shortwave radiation which the tree stratum is supposed to reflect to the low part of the canyon during multiple reflections in favor of representing realistically the upward isotropic first reflection of solar radiation, which is by far the most energetic reflection. Solar reflections calculations are fully explained in Appendix C. As

15  previously mentioned, view factors used for the multiple reflections in the shortwave radiation scheme will be applied to the longwave radiation interactions within the canyon in future works.

**5  Comparative exercise with the SOLENE model**

[revised manuscript text omitted]

30 For each of these urban canyons, 13 different vegetation layouts are prescribed (Fig.3), as well as a control case without vegetation. The vegetation blocks are rectangular parallelepipeds representing the three-dimensional tree crowns, without trunk.  These solid volumes are continuous lengthwise along the canyon. Depending on the configurations, the trees can be organized in single or double rows. According to a cut plan through the canyon, the tree crowns can fill 30, 60, or 90 % of canyon width. For the three aspect ratios, height of trees is prescribed to 5 m or 7.5 m with trunks of 2.5 or 5 m, so that the thickness of tree canopy varies between 2.5 and 5 m. Additional configurations (referred to as $h/w = 2$ rescaled vegetation) are tested for which height of trees is prescribed to 10 m or 15 m with trunks of 5 or 10 m, depending on the thickness and the location of the crown (Cf. Fig. 2). In other words, vegetation layer is doubly thicker and higher than $h/w = 0.5$ and $h/w = 1$

5 cases for each vegetation configuration in order to rescale it for higher buildings and verify the effect of adapted vegetation layouts regarding the typology of the street. For all experiments, the Leaf Area Index of trees is prescribed to 1, and the albedo is prescribed to 0.25 for road, 0.30 for walls, and 0.25 for trees. All configurations are described in Fig. 1, 2, 3 and Table 2.

To treat the ensemble of configurations, 55 digital mock-ups (52 canyons with vegetation and 3 canyons without vegetation) have been built with the computer-aided-design (CAD) Salome V7_4_0 software. All mock-ups have been meshed by
10 the GMSH software which is a finite element mesh generator. We have applied here a non-uniform meshing with a characteristic length of only 1 m in order to refine the spatial discretization of vegetation blocks, whose smallest ones for some of the vegetation layouts do not exceed 2.4 m of width and 2.5 m of height.

15 Each canyon is projected following the four street orientations 0 °, 45 °, 90 °, and 135 ° (degrees from geographical north, in the counter trigonometric direction). A specific location (defined by latitude and longitude) must be prescribed for astronomic calculations. The city of Nantes (France) is chosen in an arbitrarily manner (46 ° N, 1 ° E). The solar radiation exchanges are then calculated for a single daily cycle, and under sunlight conditions covering the four seasons by selecting dates close to equinoxes and solstices of year 2010, i.e. 20[th] March, 21[st] June, 23[rd] September, and 23[rd] December of 2010. Note also that
20 the Perez model (see section 5.1) has been parameterized in order to generate perfectly clear cloudless skies.

**5.2.2   Canyon modeling in TEB**

In the same way, TEB is run for equivalent configurations to SOLENE configurations, respecting hypotheses, approaches, and spatial resolutions differences between the two models. For TEB simulations, the geometrical parameters describing the urban canyon form, as well as the height of trees and trunks are comparable to those of SOLENE simulations. But the different spatial
25 arrangements of trees simulated by SOLENE are simply prescribed as cover fractions in TEB. As a result, some configurations (e.g. B1 and B2 with different horizontal locations or B3 and B4 with different number of tree lines, shown in Fig.3) cannot be distinguished by the TEB approach, and are associated to the same cumulative cover fraction of tree canopy in TEB. In this

manner, interactions between tree lines are not taken into account in TEB, contrary to SOLENE where rays are attenuated at each time crossing a mesh belonging to a vegetated envelope. All geometric features of both sets of simulations with SOLENE

30  and TEB are summarized in Table 2. In SOLENE, the incident radiation  is roughly attenuated  by 50 %  once crossing a mesh  which belongs to a vegetation envelope. In terms of process modeling, the formulation of transmissivity of radiation through the foliage in TEB is here simplified for the evaluation stage. In order to be consistent with the SOLENE approach, the exponential attenuation expressing a maximum interception (including the Leaf Area Index) in Eq. 3, 4, 5 is replaced by 0.5. The same way, the formulation proposed in Appendix B (Eq. B1, B2, B3, B4) for modulating the radiation attenuation depending on the likely path of rays and based on the Leaf Area Density profile, is here substituted by the expression

5   $1 - 0.5(LAD/LAI)$, so that the maximum attenuation is 0.5 when all thickness of tree canopy is passed through. Note also that TEB is forced by the same conditions of incoming solar radiation than those calculated for the roofs in SOLENE. Using a unique forcing for each component (direct or diffuse) of the solar radiation from the SOLENE simulations, TEB forcings do not take into account the non-uniform distribution of incoming diffuse solar radiation. From this imprecision result differences from 1 to 4 % between the fluxes of the two walls, depending on their orientation, for studied aspect ratios during summertime

10  (sensitive analysis not shown).

**5.2.3   Comparison method**

Finally, 880 solar radiation simulations are performed with both TEB and SOLENE models. For each of them, hourly outputs are stored. They include the direct and diffuse solar radiation received by the separated elements ($road$, $walls$, and $tree$) before multiple reflections, as well as the total shortwave radiation absorbed by the separate elements after multiple reflec-

15  tions. The main objective of the comparative exercise is to evaluate the cover fraction approach of TEB against a model (SOLENE) resolving the urban radiation budget at fine scale and with trees explicitly represented by geometrical elements. For this purpose, gaps between the simulations of received direct or diffuse solar radiation fluxes by canyon surfaces have been investigated.  During multiple reflections, the radiation is assumed to be isotropic in  both TEB and

20  SOLENE  models. Reflections from the high vegetation in TEB are also omni-directional but just constrained upwards (in other words, they are based on $\pi$) in order to represent more realistically the first and, by far, the most energetic reflection (see details in section 4.4. and Appendix C). To compare the simulations of these two models, only the central part of the SOLENE's mock-up is used in order to avoid any boundary effects (see the scheme in Fig. 3). For each flux, the values calculated by SOLENE are summed

25  over all grid points that compose each element of the canyon (separately, the road, the two walls, and the trees). Finally, for both models, the fluxes are weighted to be expressed according to the total ground-based surface of the canyon, so that they can be compared to each other and compared to the incoming radiation. Note that tables with statistical scores presented hereafter have been fulfilled with the same procedure, i.e., the Mean Absolute  Difference (MAD in W m$^{-2}$), the Mean Absolute Percentage  Difference (MAPD in %) computed from the mean daily fluxes, and the mean bias (Bias in

5   $W\,m^{-2}$), according to following equations:

$$MAE\,MAD = \frac{\sum_{i=1}^{n_s}\sum_{j=1}^{n_o}\sum_{k=1}^{n_h}\sum_{l=1}^{n_t}|F_{SOL}(i,j,k,l)-F_{TEB}(i,j,k,l)|}{n_s \cdot n_o \cdot n_h \cdot n_t} \tag{19}$$

$$MAPE\,MAPD = 100 \cdot \frac{\sum_{i=1}^{n_s}\sum_{j=1}^{n_o}\sum_{k=1}^{n_h}|\frac{\overline{F_{SOL}(i,j,k)}-\overline{F_{TEB}(i,j,k)}}{\overline{F_{SOL}(i,j,k)}}|}{n_s \cdot n_o \cdot n_h} Bias = \frac{\sum_{i=1}^{n_s}\sum_{j=1}^{n_o}\sum_{k=1}^{n_h}\sum_{l=1}^{n_t}(F_{SOL}(i,j,k,l)-F_{TEB}(i,j,k,l))}{n_s \cdot n_o \cdot n_h \cdot n_t}$$

$($

$\tag{20}$

The indexes $i$, $j$, $k$, $l$ refer to as season, street orientation, aspect ratio of the street, and hour of the day, respectively, with $n_s = 4$, $n_o = 4$, $n_h = 3$, $n_t = 24$. We also run simulations of the $h/w = 2$ additional cases where canyons are greened  with

10   a rescaled vegetation (see section 5.2.1 and Fig. 2).

**6   Results**

**6.1   General analysis and seasonal effects**

Table 3 presents the statistical scores computed for the shortwave radiation absorbed by the different elements of the canyon for both cases, with and without vegetation. Since multiple reflections are treated differently between TEB and SOLENE,

15   these results are used to show the magnitude of variability of  differences related to the presence of vegetation in the canyon or the considered season. Results are presented here by gathering the experiments performed with different street orientations and different vegetation layouts, but by distinguishing the seasons. In the light of the mean biases, the TEB model tends to systematically overestimate the absorption by road and walls compared to SOLENE (up to +16 and +  8.5 $W\,m^{-2}$ for road and walls, respectively), and to underestimate the absorption by trees (up to  -16 W

20   $m^{-2}$), whether the configuration or the season. Considering that the temperate climate is characterized by four distinct seasons with contrasting sunshine, air temperature and humidity conditions, seasonal analysis was undertaken. Analysis of the results for each season separately indicates that relative  differences (MAPD) are particularly high for wintertime simulations for road and trees, due to a very low incoming solar radiation at that period. Nonetheless, the associated  MAD and biases are acceptable (less than 3.5 and $\pm$ 3 W $m^{-2}$, respectively). Summer is the season which provides

25   the best results in terms of  MAPD when the canyon is tree-filled (less than 25 %) or not (3 %). This season is also the most relevant to be examined here because our first concern is to improve the simulation of the potential cooling effect of street trees in a urban environment, submitted to a strong Urban Heat Island at this period. The evaluation is focused on

effects of deciduous trees which are typical and widely present in cities under temperate climate. Such trees are leafless during winter, so that they have a negligible impact on thermal comfort and energy demand at this season. That is why we focus on the summertime example hereafter, to assess the TEB performances for simulating the solar radiative exchanges in idealized canyons, vegetated or not.

**6.2 Case of urban canyons without vegetation**

5 The radiative calculations in TEB are first evaluated for the cases without vegetation. Several comparisons with observations of radiation fluxes at neighborhood-scale have been performed (Masson et al., 2002; Lemonsu et al., 2004, 2010; Pigeon et al., 2008). They have shown a good capacity of the model in computing the upward short- and long- wave radiation at the top of the urban canopy, in real case configurations. But these evaluation exercises have not allowed us to analyze separately the radiative contributions of various elements that compose the urban environment. Such a model-model comparison in a controlled
10 framework is ideal to deeply investigate and evaluate the TEB radiation parameterizations.

Both direct and diffuse solar radiation received by the road and the separate walls before any reflections, as well as the total shortwave radiation absorbed by these surfaces, are studied. An example of daily cycle is presented in Fig. 4 for the case $h/w = 1$ and the four street orientations, for summer season. The scatterplots integrate all the hourly fluxes simulated for the
15 three aspects ratios, the four street orientations, and the four seasons (Fig. 5). As expected, they demonstrate a strong positive linear relationship between fluxes calculated by SOLENE and TEB ($R^2 \geq 0.99$ except for diffuse solar radiation absorbed by roads: $R^2 = 0.979$) despite various meteorological conditions.

The comparison between SOLENE and TEB simulations for the direct solar radiation received by road and walls before any
20 reflections highlights very good results. TEB is able to reproduce the geometrical effects of the canyon on radiation penetration according to the time of the day, as well as the street orientation. For NE-SW and NW-SE oriented streets, TEB simulates correctly the dissymmetry of fluxes between the two walls, as well as the temporal shift in peak of radiation received by the road in comparison with the N-S oriented street. For the E-W street case, the direct radiation received by the road is marked by a plateau effect between 8 am and 7 pm. The two walls have different behaviors: the wall  most exposed to the sun receives
25 the maximum direct radiation at solar noon, whereas the most shaded wall receives direct radiation only early in the morning and late in the afternoon. The scores confirm the good performances of TEB:  MAD are 4.39 and 2.49 W m$^{-2}$, and biases are -0.28 and +0.40 W m$^{-2}$ for road and walls, respectively (Table 4). They are associated with low  MAPD of only 1 % for both road and walls.

30   In the calculation of diffuse solar radiation, TEB does not  consider the two walls  separately. Therefore, the diffuse solar radiation flux  is compared between the composite wall of TEB, and the average of the diffuse solar radiation fluxes received by  separate walls of the SOLENE  simulations (Fig. 4). By considering an average surface, TEB underestimates the diffuse solar radiation received by walls in the morning and the afternoon, and it overestimates it at solar noon. On the opposite, it overestimates the

5  diffuse solar radiation received by the road in the morning and the afternoon, and it underestimates it at solar noon. In this case,  MAD and biases remain weak (less than 3.5 and $\pm 1$ W m$^{-2}$, respectively) because the involved fluxes are not very high (depending on the season, the diffuse component is only 15-25 % of the total incident solar radiation), but  MAPD are slightly higher than for direct solar radiation, reaching 7 and 5 % for road and walls, respectively (Table 5). However, these discrepancies have no impact on performances of TEB neither at daily scale on fluxes of walls nor on instantaneous cumulated

10  canyon fluxes. Moreover, the dissymmetry of received diffuse solar radiation fluxes is no longer observed in E-W oriented cases.

Finally, the total shortwave radiation absorbed by road and walls is well estimated by TEB despite the simplified hypotheses of the model and the use of a unique sky view factor by surface:  MAD and biases are 6.03 and +3.50 W m$^{-2}$ for road, respectively, and 3.38 and +2.80 W m$^{-2}$ for walls (Table 6). In view of the important incident radiation flux, exceeding

15  1 000 W m$^{-2}$ at solar noon, the  MAPD of 3 % for both surfaces  remain moderate.

**6.3  Case of urban canyons with vegetation**

The same evaluations are conducted for vegetated canyons. The statistical scores are computed as previously (see Eq. 19, 5.2.3,

20) but by accounting for the 13 vegetation layouts. We also computed additional cases of vegetation layouts for the configurations referred to as "$h/w = 2$ rescaled vegetation" (see section 5.2.1. and Fig. 2 for further explanations). As an example,

20  a comparison of hourly fluxes is presented in Fig. 6, for the layout A and an aspect ratio $h/w = 1$, and for the summer daily cycle. This configuration is one of the most simple and comparable layouts between the two models: trees are 7.5 m high (i.e. almost the same height as buildings), tree crowns have a 5 m thickness, they are centered in the middle of the canyon and cover 90 % of the canyon width on the horizontal plane.

25  The daily evolution of direct solar radiation received by the different elements of the canyon can be compared to the case without vegetation (Fig. 4). The same patterns are obtained, whether for walls or road and the four street orientations, with an attenuation due to the presence of trees. Here, a significant part of direct incoming solar radiation is intercepted by the foliage.

Due to its partial transparency properties, the foliage allows at least half of radiation fluxes to pass through (see section 5.2.2.). As a result, the urban surfaces (walls and road) receive less incoming direct radiation but are never totally obstructed by trees. These processes are correctly simulated by TEB with some limitations: due to the expression of direct solar radiation intercepted by high vegetation at the top of the crown which is treated as an horizontal surface in TEB (Eq. 1), the fluxes reaching the trees in TEB are globally underestimated compared to the SOLENE fluxes that include contributions on the vertical faces of the crown envelope (Fig. 7 a., all seasons and configurations). Consequently, the solar radiation which is not intercepted by

5 the tree layer in TEB simulations is assigned to the road and the bottom of walls. This leads for the configuration A to MAD of 9.27, 3.52 and 6.22 W m$^{-2}$ for road, walls, and trees, respectively, and  MAPD of 14, 5, and 4 % only (Table 4). The thicker the crown is, the greater is the  difference associated to the tree (D1 vs B1 or D2 vs B2), particularly when the tree rows are away from the walls (B1 vs B2, D1 vs D2 or D3 vs D4), contrary to a continuous tree layer occupying almost the entire width of the canyon (A, C1, C2).

As expressed in Eq. 8 and 9, the diffuse solar radiation fluxes received by road and walls (walls are managed together as an average surface), depends on the sky view factor of the given surface and on an attenuation coefficient of the incoming radiation through the foliage. Regarding all seasons and configurations combined, the received solar radiation by ground-based surfaces is also overestimated while the diffuse solar radiation flux reaching tree crowns is underestimated in

15 the TEB simulations (Fig. 7 b.).  Remembering that the diffuse solar radiation received by the trees is calculated in TEB as the residual part of the incoming diffuse solar radiation flux which was not received by road and walls (see Eq. 10). Contrasted results are obtained for the vertical surfaces whether the vegetation layouts, in particular between double-row or centered trees (B3 vs B4). As previously discussed in section 5.4, these defects are related to the use of a single sky view factor for each surface, which is computed at mid-height of buildings for walls and in the middle of the street for road. In addition, the

20 isotropic nature of the diffuse solar radiation leads to an exacerbation of the underestimation of the received flux by the high vegetation, particularly in cases where the side surface of vegetation envelopes is larger: for example, thick crowns with longer sides and vegetation layouts including two rows where the number of sides is doubled (A vs B3, C1 vs C3). Considering the extremely vegetated canyon A,  MAPD are 15, 9 and 20 % for road, walls, and tree, respectively, corresponding to MAD of 1.91, 1.51 and 3.51 W m$^{-2}$ (Table 5). Globally, the estimation of the  diffuse solar radiation received

25 by the road and walls are acceptable regarding the  MAPD which are $\leq 30\%$. However, the relative  differences associated to the tree layer are contrasted (from 12 % to 49 %), depending on the characteristics of the vegetation layout.

For the same configuration, comparing TEB results to the SOLENE simulations as reference, the total shortwave radiation

absorbed by the different elements of the canyon is simulated with correct daily dynamics (Fig. 6). Despite similar tempo-

30    ral behavior of the fluxes, their magnitudes can greatly diverge, especially for the road and tree surfaces with  some

MAPD greater than 30 % and some biases greater than $\pm 25$ W m$^{-2}$ (Table 6). The global scatterplot (Fig. 7 c., all seasons and

configurations) confirms the trends previously found and the absorbed flux by the trees is underestimated. The predictability

based on SOLENE results is also weaker ($R^2 = 0.954$). These poor performances of TEB regarding the statistical

scores have to be interpreted with caution. This work aims at evaluating the cost in term of performances of TEB to simulate

5    a correct allocation of radiative fluxes for each facet of the canyon in spite of a simple geometric approach based on cover

fractions and unique a view factor per interaction. For example, the tree fraction is computed as a cumulative fraction

of all crowns in the street in TEB. Thus, the interactions between tree lines are not allowed, contrary to in SOLENE. However,

 the consequences of some other divergent assumptions can alter TEB scores by the comparison

of the absorbed fluxes even if neither one or the other model is the truth (as would have been observations) and SOLENE can

10    not be considered as the reference simulation for these points. Indeed, the way to calculate the transmissivity during multiple

reflections is dissimilar between the two models (see section 5.2.3). TEB code computes infinite isotropic reflections using a

unique view factor for each facet of the canyon and mean radiative transmissivity terms.  The formulation of the

mean radiative transmissivity terms  is based on strong hypotheses on the considering the studied interaction

 potential attenuation of rays: they express the ratio between the sector of canyon scanned by the view factor

15    associated to the studied interaction and the thickness of crown contained in this canyon sector (see Appendix B). In addition,

reflections from the high vegetation are constrained upwards in TEB (see section 4.4 and Appendix C for full explanations).

On the contrary, SOLENE radiative scheme computes  isotropic multiple reflections

within the canyon for each triangular mesh, using the radiosity method. The vegetation envelopes are here strictly semitransparent because the penetration of light through the foliage can not be modulated by the Leaf Area Density. Whatever the

20    foliage thickness which is crossed, as soon as a ray reaches a cell of vegetation envelope in SOLENE, this radiation is attenuated

by half.

Further works (not shown) have investigated the sensitivity of TEB results and performances according to the characteristics of the different vegetation layouts. They do not demonstrate clear and systematic patterns when studying the impact of:

25    (1) tree horizontal coverage (or the tree fraction); (2) tree canopy height compared to the building height; (3) tree location -

centered or on side - in the canyon on the  MAD and difference percentages recorded. It could be explained by

the interaction between opposite effects regarding the vegetation layouts characteristics.

**6.4 Benefits of TEB developments**

In this section, simulations provided by the initial version of TEB on one hand and by the implemented version on the other hand are compared. For this purpose, we use in the new version equations related to the transmission of radiation fluxes from the high vegetation following the expression presented at Eq. 3, adapted when it required to the considered Leaf Area Density (see Appendix B for further details).

**6.4.1 Analysis of the  behavior of absorbed shortwave radiation  per canyon surface**

The shortwave radiation received and absorbed by the walls and the road can be strongly affected by the presence of tree vegetation. The comparison between the initial version of TEB which deals with vegetation at ground level, and the new version which explicitly includes an additional tree stratum, shows differences. ~~Before weighting the fluxes by the appropriate cover fractions, the shortwave radiation absorbed by the ground-based surfaces are largely overestimated in the initial version of TEB (+ 77for road and +66for the garden in the case Ain summer ). These defects highlight the lack of representation of the interception of rays before reaching the road related to a reduced absorption of the vegetation by its transparency properties. On the opposite, we observe an underestimation of the absorbed flux regarding the two walls: -41for wall A and -53for wall B. It is explained because during multiple reflections, the tree layer (which is thick in case A) facing walls accentuates the absorption of vertical surfaces in the implemented version. Nonetheless, weighting the fluxes at the canyon scale affects greatly the road and garden fractions. In the reference cases, the high vegetation is treated as a ground-based vegetation fraction which is included in the garden fraction ($\delta_g$). Consequently, the road fraction, which is always defined as $1 - \delta_g$, is varying according to the vegetation layout. In this way, compared to the current version, a larger absorbed shortwave radiation flux in W m$^{-2}$ of road may lead to a lower flux expressed in W m$^{-2}$ of canyon when the garden fraction is high . During the summer and all configurations confounded (Fig. 8), we observe a mean underestimation of the shortwave radiation absorption of -44for the road and -10for the walls, compared to the implemented version. This can have a significant impact on thermal comfort conditions for pedestrians as well as on energy consumption of buildingsfor air-conditioning usage. Moreover,reference case presents a strong dispersion in the simulation of solar fluxes absorbed by the vegetation, depending on the configurations (aspect ratio of canyon, street orientation and vegetation layout). In some cases especially, this flux is greater than in the new version of TEB. This effect is related to the fact that the reference version does not treat the vegetation as a semi-transparent object and does not take into account its transmissivity properties~~considered surface, artificial surfaces absorb

less energy when they are shaded by an explicit tree canopy, while the daily dynamics are similar. At the beginning and the end of the range of lighting hours, trees absorb more energy after the implementation of a real canopy within the canyon instead of initial simulations where the foliage was placed at the ground level. It is explained by the fact that the more the trees are high in the canyon, the more they can receive solar radiation and avoid the shading of buildings. However, around the solar noon, the absorption by the trees is substantially reduced by its transmissivity properties in regards to the initial version where all the urban vegetation surfaces were opaque. As a result, the vegetation absorbs all the solar radiation which is received and not reflected. In the present case, this part is 75 % of the incident solar radiation since $\alpha_t = 0.25$. In the new version, 50 % of the received solar radiation is transmitted through the foliage and 75 % of the remaining flux is absorbed by trees, i.e. only 37.5 % of the total shortwave radiation received by trees.

Comparing the fluxes before and after their weighting based on their canyon fraction, the road absorption by squared meters of canyon can extremely vary following the garden fraction at the ground in the initial version. Indeed, in the reference cases, the high vegetation is treated as a ground-based vegetation fraction which is included in the garden fraction ($\delta_g$). This limitation implies that the garden fraction takes the value of the greater fraction between low and high vegetation. Consequently, the road fraction, which is always defined as $1 - \delta_g$, is varying according to the  vegetation layout. For example, here, the road fraction is only 0.10. For this experiment, the differences between before and after the weighting of absorbed fluxes by trees is small because the tree fraction is high (=0.90). In this case, there is no differences for the walls according to the unit of the fluxes since the canyon aspect ratio is h/w=1. For smaller aspect ratios, the fluxes of walls based on the canyon surface will be lower than those calculated by squared meters of wall. On the opposite, for aspect ratios exceeding 1, the fluxes of walls based on the canyon surface will be higher. The new developments have corrected a systematic overestimation bias in the reference version. This can have a significant impact on the simulation of thermal comfort conditions for pedestrians as well as on energy consumption of buildings for air-conditioning usage.

**6.4.2 Analysis of an integreted  canyon albedo**

The  albedo of the canyon ($\alpha_{can}$) is calculated as the ratio between the outgoing shortwave radiation (which is deduced from the difference between the total incoming radiation $S_{can}^{\Downarrow} + S_{can}^{\downarrow}$ and the absorbed radiation $S_{can}^{*}$ ) and the total

25  incoming shortwave radiation:

$$\alpha_{can} = \frac{\left(S_{can}^{\Downarrow} + S_{can}^{\downarrow}\right) - S_{can}^{*}}{\left(S_{can}^{\Downarrow} + S_{can}^{\downarrow}\right)} \tag{21}$$

For each simulation, these fluxes are  used at the solar noon in order to compute   an instantaneous canyon albedo. The results in summertime are presented as boxplots (Fig. 9) that gather  all the vegetation layouts if any, but that distinguish the canyon aspect ratios. Only North-South canyons are represented here. It is a synthetic indicator of TEB enhancement at the scale of the entire canyon, which is crucial when SURFEX is run coupled with an atmospheric model such as the research MESO-NH model (Lafore et al., 1997) or the operational model named AROME (Seity et al., 2011) to provide climatic simulations.

The aspect ratio has a significant impact on the canyon albedo: the canyon albedo decreases with an increasing aspect ratio. Since the geometry and radiative properties of the vegetationless canyons and ground-based vegetation canyons in the reference version of TEB are comparable, they provide similar canyon albedos. It is explained by the absence of foliage in the vertical plane and identical albedos of road and garden ($\alpha_r = \alpha_g = 0.25$).
10   For canyons with $h/w = 0.5$ aspect ratios, the albedo of vegetated canyons with an explicit tree canopy is systematically lower than unobstructed canyons. Trees can more easily absorb the incoming solar radiation than ground-based
15    surfaces. For higher aspect ratios, the impact of trees is depending on the balance between
20  their greater sky view factor and reflectivity regarding artificial surfaces and their trapping effect on solar radiation, reducing or increasing the total solar energy absorption by the canyon, respectively. Note also that their relative impact is smaller when the thickness of the crown is small, proportionally to the height of the buildings. For rescaled vegetation cases in deep streets, we observe a greater variability of
25

 canyon albedos compared to the same urban form with thinner tree crowns placed in the lower part of the canyon.

**7 Conclusions**

In order to investigate some of the physical processes related to the presence of vegetation in urban environment, e.g. for microclimate, hydrology or building energy consumption issues, the modeling is definitely a necessary tool.

The TEB model has been refined and improved in order to explicitly represent street trees and their impacts on radiative transfer. The new parameterization is based on the simple hypotheses of TEB: (1) a little detailed geometry without specific spatial arrangement of ground-based surfaces and (2) a single view factor to each emitting and receiving surface applied for radiative calculations.

To take into account the tree canopy in TEB, it was however required to add a new vegetated stratum on the vertical plane, which can shade the road, the walls and the low vegetation. This modification led to more complex radiative calculations, but has been done with a concern to preserve a certain level of simplicity and to limit the number of new input parameters for TEB. It is important to emphasize that the model is designed to be run over whole cities, for which it can simulate the local climatic variability related to urban landscape heterogeneities at the neighborhood scale. This means that computing times must be acceptable, and that input urban data must be available or easy to define. Consequently, the high vegetation is here described using only five input parameters: cover fraction of trees, height of trees, height of trunks, Leaf Area Index and albedo.

This simplified characterization of high vegetation necessarily induces some uncertainties on solar radiative exchanges. We estimated it by carrying out a comparative exercise between TEB and a high spatial-resolution solar and lighting model (SOLENE). On the basis of an idealized geometry of urban canyon with various vegetation layouts, TEB is evaluated regarding the  total shortwave radiation flux absorbed by the elements that compose the canyon. TEB simulations in summer gathered best scores for all configurations and surfaces considered, which is precisely the most relevant season to assess the cooling effect of deciduous trees under temperate climate. Statistical scores have demonstrated a good capacity of TEB to solve the radiative balance of canyons without vegetation despite the use of a unique sky view factor for each facet of the urban scene.  Mean Absolute Differences and biases of 6.03 and +3.50 W m$^{-2}$ for road, respectively, and of 3.38 and  +2.80

W m$^{-2}$ for walls have been recorded  in vegetationless canyons. In view of the important incident radiation flux, exceeding 1 000 W m$^{-2}$ at solar noon, the Mean Absolute Percentage Differences of 3 % for both surfaces remain moderate. Additionally, it is necessary to put in perspective obtained scores with the fact that they include the error generated by the treatment of walls as a mean wall in TEB in comparison with separate walls of SOLENE considering the diffuse component of the solar radiation. This identified error is no longer existing at the canyon or daily scales. Concerning the vegetated canyons we noted a high variability of statistical scores depending on the vegetation layout. The greater uncertainties are found for the solar radiation fluxes received and absorbed by the high vegetation. The Mean Absolute  Difference averaged over the vegetation configurations during summertime is  21.12 ± 13.39 W m$^{-2}$  or 20.92 ±10.87 % of Mean Absolute Percentage Difference for the total shortwave absorption but these scores are associated to acceptable biases:  −15.96 ± 15.93 W m$^{-2}$ . The systematic underestimation of fluxes reaching the new tree stratum is explained by the cover fraction approach in TEB where sides of the crown are not represented. Interactions between potential tree lines or intra-canopy scattering and trapping are also neglected. Presented scores are including the effect of divergent approaches in the formulation of the transmission process through trees between TEB and SOLENE even though neither of the two models could be considered as a reference at this stage for this point.

The parameterization of shortwave radiation exchanges within the canyon is now more realistic: shading effects of trees on vertical and ground-based surfaces but also shading effects of buildings on trees are computed. This is achieved by adding a new specific cover fraction describing the horizontal extend of high vegetation. Infinite reflections within the canyon are also conditioned to the transmissivity term calculated per  pair of exchanging surfaces. This study  demonstrated the enhancement of new developments on the computed absorbed shortwave radiation fluxes within the canyon between the former reference version of TEB and the implemented version. In the current version, trees can intercept and absorb the direct solar radiation at the canopy level instead of from the ground. Consequently, the walls and ground are more shaded.  High and low vegetation fractions are now explicitly dissociated. The grass and bare soil fractions only contribute to the garden fraction. In this way, the road fraction, defined as $1 - \delta_g$, is independent of the  tree cover. This implies that the weighting of fluxes at the canyon scale   has become realistic. The aspect ratio has a significant impact on the canyon albedo: the canyon albedo decreases with an increasing aspect ratio. For canyons with $h/w = 0.5$ aspect ratios, the albedo of vegetated canyons with an explicit tree canopy is systematically lower than unobstructed canyons. Trees within the canyon can more easily absorb the incoming solar radiation than ground-based surfaces do. For higher aspect ratios, the impact

of trees is depending on the balance between their greater sky view factor and reflectivity regarding artificial surfaces and their trapping effect on solar radiation, reducing or increasing the total solar energy absorption by the canyon, respectively. Canyon albedo is also more vegetation-responsive for $h/w = 1$ urban forms or in cases with $h/w = 2$ canyons when the  tree crown thickness is scaled regarding the depth of the street.

The future developments will focus on the separate calculation of turbulent energy fluxes for ground-based and high vegetation. The aerodynamic effect of trees on air flow within the canyon should also be parameterized. The adaptation of TEB to trees taller than buildings will broaden the range of potential neighborhoods to which its tree model could be applied. Based on this more sophisticated version of TEB, new impact studies could be conducted and greening adaptation strategies could be evaluated more precisely.

**8   Code availability**

The TEB code is available in open source via the surface modeling platform called SURFEX, downloadable at http://www.cnrm-game-meteo.fr/surfex/. This Open-SURFEX will be updated at relatively low frequency (each 3 to 6 months) and developments presented here are not included in the last version yet. If you need more frequent updates, or if you need what is not in Open-SURFEX (DrHOOK, FA/LFI formats, GAUSSIAN grid), we invite you to follow the procedure to get a SVN account and to access real-time modifications of the code (see instructions at the previous link).

**Appendix A:  Sky view and view factors**

Sky view factors for road, garden and wall (Eq. A1, A2) as well as the view factors between elements remain unchanged in relation to initial version of the radiative calculations in TEB (described in Masson (2000) and Lemonsu et al. (2012)):

$$\Psi_{rs} \;=\; \Psi_{sr} = \sqrt{\left(\frac{h}{w}\right)^2 + 1} - \frac{h}{w} \tag{A1}$$

$$\Psi_{wr} \;=\; \Psi_{ws} = \frac{\frac{1}{2}\left(\frac{h}{w} + 1 - \sqrt{\left(\frac{h}{w}\right)^2 + 1}\right)}{\frac{h}{w}} \tag{A2}$$

$$\Psi_{sw} \;=\; 1 - \Psi_{sr} \tag{A3}$$

20 $$\Psi_{ww} \;=\; 1 - 2\Psi_{ws} = \frac{\sqrt{\left(\frac{h}{w}\right)^2 + 1} - 1}{\frac{h}{w}} \tag{A4}$$

$$\Psi_{rw} \;=\; 1 - \Psi_{rs} = 1 - \left(\sqrt{\left(\frac{h}{w}\right)^2 + 1} - \frac{h}{w}\right) \tag{A5}$$

For the tree canopy, the sky  view factor and view factors from road and walls are computed in the middle of the canyon and at mid-height of crown ÷($h_{cw}$):

$$\Psi_{st} \;=\; \sqrt{\left(\frac{h}{w}\cdot\frac{h - h_{cw}}{h}\right)^2 + 1} - \left(\frac{h}{w}\cdot\frac{h - h_{cw}}{h}\right) \tag{A6}$$

5 $$\Psi_{rt} \;=\; \sqrt{\left(\frac{h}{w}\cdot\frac{h_{cw}}{h}\right)^2 + 1} - \left(\frac{h}{w}\cdot\frac{h_{cw}}{h}\right) \tag{A7}$$

$$\Psi_{wt} \;=\; 1 - \frac{1}{2}\left(\Psi_{st} + \Psi_{rt}\right) \tag{A8}$$

$\Psi_{st}$ and $\Psi_{rt}$ are calculated in the same way as for $\Psi_{sr}$ in the middle of the canyon but they are adjusted to the $[h_{cw} - h]$ area and the $[0 - h_{cw}]$ area, respectively. $\Psi_{wt}$ is computed as the complementary term of $\Psi_{st}$ and $\Psi_{rt}$ on $2\pi$ and for one wall (as

10 wall as $\Psi_{ws}$ and $\Psi_{ww}$). We normalized $\Psi_{wt}$ to express it on one $\pi$ only.

**Appendix B: Mean radiative transmissivity of canopy tree canopy**

The multiple reflections of solar radiation inside the canyon (as detailed in Masson (2000) and Lemonsu et al. (2012)) are now affected by the presence of trees whose foliage intercepts, reflects and absorbs a part of the energy. The transmissivity of radiation through the foliage of tree canopy is variable according to the way the rays cross the foliage and the distance they travel. According to the location of tree crowns inside the canyon and the dominant orientation of radiation (e.g. emission from sky to ground or from wall to ground do not reach the foliage the same way), these rays can cross all the foliage thickness or only a small portion. The vertical distribution of leaves in the tree crowns has also an impact on transmissivity.

Different transmissivity functions (referred to as $\tau_{12}$ for exchanges between element 1 and element 2) are calculated depending of the surfaces involved in radiative exchanges. One distinguishes four cases of radiation exchanges, by hypothesizing that transmissivity functions are symmetric, i.e., exchanges from element 1 to element 2 is equivalent to the reverse way: between ground-based surfaces and sky ($\tau_{rs} = \tau_{sg}$) or wall ($\tau_{rw} = \tau_{gw}$), between wall and sky ($\tau_{ws}$), between wall and wall ($\tau_{ww}$). For each case we admit, according to Lee and Park (2008), that majority of radiation exchanges occurs in a specific  area of the canyon, for which the Leaf Area Density ($LAD_t$ expressed in $m^2$ of leaf area per $m^3$ of volume) is calculated:

$$\tau_{rs} = 1 - \delta_t \left[ 1 - \exp^{\left( -k \int_0^h LAD_t \, dz \right)} \right] \tag{B1}$$

$$\tau_{rw} = 1 - \delta_t \left[ 1 - \exp^{\left( -k \int_0^{\frac{h}{2}} LAD_t \, dz \right)} \right] \tag{B2}$$

$$\tau_{ws} = 1 - \delta_t \left[ 1 - \exp^{\left( -k \int_{\frac{h}{2}}^h LAD_t \, dz \right)} \right] \tag{B3}$$

$$\tau_{ww} = 1 - \delta_t \left[ 1 - \exp^{\left( -k \int_{\frac{h}{4}}^{\frac{3h}{4}} LAD_t \, dz \right) \left( -k \int_0^h LAD_t \, dz \right)} \right] \tag{B4}$$

The limits of integrals involved in the calculation of the transmissivity functions have been consistently defined with canyon sectors scanned by the associated view factors. For example, Eq. 8 express how the diffuse incoming solar radiation reaches the road from the sky. This formulation is typically based on the sky view factor of the road $\Psi_{rs}$ which scans an area covering the entire height of the building. We added a transmissivity function $\tau_{rs}$ describing the ratio between the thickness of the tree crown

within the canyon and the vertical extent of the area of interest, here $[0 - h]$. Note that these expressions are in accordance with the one applied in Eq. 3 in which $LAD_t$ is integrated on the entire thickness of foliage so that it is equivalent to $LAI_t$. As mentioned above, since the tree crown is described as a rectangular parallelepiped with a regular distribution of leaves, a uniform vertical profile of LAD is here applied.

**15   Appendix C: Total shortwave radiation absorption by solving infinite reflections**

For solar radiation calculations, the TEB model takes into account an infinite number of reflections between all elements composing the urban canyon. At each reflection, the isotropic radiation intercepted by a given element (1) after reflections on one of the other elements (2) is conditioned by the view factor of (2) from (1) referred to as $\Psi_{12}$ (see Appendix A), the mean radiative transmissivity $\tau_{12}$ (see Appendix B) and the reflection is then determined according to reflective properties of (1). As 20   seen in the section 4.4, the total shortwave radiation absorbed by each elements of the canyon or redirected towards the sky is function of infinite reflections $R_\infty$, $G_\infty$, $A_\infty$, $B_\infty$, and $T_\infty$ that are still unknowns at this stage. These terms include the first reflection on each element: $R_0$, $G_0$, $A_0$, $B_0$, $T_0$. For road, garden, and walls, $R_0$, $G_0$, $A_0$ and $B_0$ simply depend on the incident solar radiation on the surface and the albedo. From the equations 14, 15, 16, 17 and 18, we can deduce the reflected part occuring at the $n+1^{th}$ absorption as the complementary term of the $n+1^{th}$ solar radiation reception for only opaque elements 25   of the canyon, $road$, $garden$, $wall\ A$, and $wall\ B$, respectively. As examples, we obtain for road and wall A, respectively:

$$R_0 = \alpha_r \left( S_r^{\Downarrow} + S_r^{\downarrow} \right) \tag{C1}$$

$$R_{n+1} = R_0 + \alpha_r \left[ \Psi_{rw}\tau_{rw}W_n + c_{rt}\Psi_{rt}\delta_t T_n \right] \tag{C2}$$

$$A_0 = \alpha_w \left( S_w^{\Downarrow} + S_w^{\downarrow} \right) \tag{C3}$$

$$A_{n+1} = A_0 + \alpha_w \left[ \Psi_{ww}\tau_{ww}B_n + \Psi_{wr}\tau_{wr}\left( \delta_r R_n + \delta_g G_n \right) + c_{wt}\Psi_{wt}\delta_t T_n \right] \tag{C4}$$

The specific coefficients associated to the view factors related to the high vegetation in shortwave reflections calculations are 5   defined as:

$$c_{rt} = 0 \tag{C5}$$

$$c_{wt} = \frac{1 - \Psi_{st}}{2 - \Psi_{st} - \Psi_{rt}} \tag{C6}$$

Only reflections from trees to sky or top part of walls are allowed. As explained in section 4.4, this mode of reflection by the leaves during the first reflection, which is by far the most energetic one, is more likely to occur than in an isotropic way. This 10   assumption could be easily bypassed by fixing the previous coefficients to 1.

For the first tree canopy reflection, the part of received direct solar radiation is corrected by the transmitted flux (see Eq. 2):

$$T_0 \;=\; \alpha_t \left[ \left( S_t^{\Downarrow} - S_t^{\gg} \right) + S_t^{\downarrow} \right] \tag{C7}$$

Some uncertainties remain about relevance of sky view or view factors which ones could formulate to represent reflective contributions from other surfaces at $n^{th}$ to the absorption or reflection by the tree layer at $n+1^{th}$, as well as potential absorption of energy within the tree's crown. Consequently, the solar flux reflected by $tree$ at $n+1^{th}$ has been determined as the residual term by assuming that the $n^{th}$ solar reflection coming from each element ($road$, $garden$, $wall\ A$, and $wall\ B$) which is not received by $road$, $garden$, $wall\ A$, $wall\ B$ or which is not returned to $sky$ at $n+1$ is received by $tree$.

During each interreflection, a part of $n^{th}$ reflected flux which is potentially available for the $(n+1)^{th}$ reflection or absorption is intercepted by high vegetation. This intercepted part related to the presence of foliage on the way of scattered rays towards each receiving element or sky are formulated as following:

$$I_s\,(n+1) \;=\; \Psi_{sr}(1-\tau_{sr})(\delta_r R_n + \delta_g G_n) + \Psi_{sw}\,(1-\tau_{sw})\,\frac{A_n + B_n}{2} \tag{C8}$$

$$I_r\,(n+1) \;=\; \Psi_{rw}\,(1-\tau_{rw})\,\frac{A_n + B_n}{2} \tag{C9}$$

$$I_g\,(n+1) \;=\; \Psi_{rw}\,(1-\tau_{rw})\,\frac{A_n + B_n}{2} \tag{C10}$$

$$I_{w_A}\,(n+1) \;=\; \Psi_{wr}(1-\tau_{wr})(\delta_r R_n + \delta_g G_n) + \Psi_{ww}\,(1-\tau_{ww})\,B_n \tag{C11}$$

$$I_{w_B}\,(n+1) \;=\; \Psi_{wr}(1-\tau_{wr})(\delta_r R_n + \delta_g G_n) + \Psi_{ww}\,(1-\tau_{ww})\,A_n \tag{C12}$$

Finally, the solar flux which is intercepted by $tree$ at $(n+1)$ is expressed as the sum of interceptions on the way of receiving elements of the canyon or sky:

$$I_t\,(n+1) = \frac{1}{\delta_t}\left( I_s\,(n+1) + \delta_r I_r\,(n+1) + \delta_g I_g\,(n+1) + \frac{2h}{w}\,\frac{I_{w_A}\,(n+1)+I_{w_B}\,(n+1)}{2} \right) \tag{C13}$$

[revised manuscript text omitted]

$$

*Acknowledgements.* We acknowledge Laurent Malys for his technical support on SOLENE first runs. Thanks are given to the anonymous reviewers for their helpful comments.

**References**

[revised manuscript text omitted]